# Neuromorphic detection and cooling of microparticles in arrays

Yugang Ren [1] ✉, Benjamin Siegel [2], Ronghao Yin[1], Qiongyuan Wu[1], Jonathan Pritchett[1], Muddassar Rashid[1] & James Millen [1,3] ✉

Micro-objects levitated in a vacuum are an exciting platform for precision sensing due to their low dissipation motion and the potential for control at the quantum level. Arrays of such sensors would offer increased sensitivity, directionality, and in the quantum regime the potential to exploit correlation and entanglement. We use neuromorphic detection via a single event based camera to record the motion of an array of levitated microspheres. We present a scalable method for arbitrary multiparticle tracking and control by implementing real-time feedback to simultaneously cool the motion of three uncoupled objects, a demonstration of neuromorphic sensing for real-time control at the microscale.

Modern technology relies on mechanical sensors, from accelerometers in mobile devices[1] to wearable health monitors[2]. As sensors are miniaturised, their surface-to-volume ratio increases and they dissipate more energy via their thermal contacts and through surface strain[3], limiting their performance. By levitating nano- or microparticles under ultra-high vacuum conditions, using optical, electrical or magnetic fields[4,5], one creates a mechanical oscillator with remarkably low dissipation[6,7]. Force sensitivities of yoctonewtons[8,9] and torque sensitivities at the $10^{-27}$ N m Hz$^{-1/2}$ level[10] have been achieved, with levitating sensors achieving quality factors in excess of $10^{10}$ [6], motivating researchers to use these systems to search for dark matter[7,11,12] and gravitational waves[13,14].

The control of levitated particles allows the exploration of a wide range of fundamental science[4,5], and the demonstration of cooling to the ground state of an optical potential[15–20] opens the door to macroscopic quantum physics[21–24]. An emerging frontier in this field is the study of arrays of particles, which in the quantum regime would allow the generation of entanglement[25] and tests of quantum gravity[26,27]. Interactions have been observed between pairs of levitated nanoparticles in optical[28–31], electrodynamic (Paul)[32] and magnetic[33] traps. Detecting and controlling multiple particles in vacuum has so far involved either single particle control with sympathetic cooling[31,32,34] or small arrays of optical traps[35,36]. Some applications will require the control of arrays of tens, or even thousands, of levitated particles[11].

We use neuromorphic imaging for the control of arbitrary particle arrays across a wide field-of-view. Neuromorphic sensors are highly efficient detectors that mimic neurobiological information gathering[37–41]. Dynamic vision sensors (DVS) are neuromorphic sensors that mimic the retinal response[42], detecting changes across a threshold on each pixel in an array asynchronously to produce a stream of events[43], ideally suited for object tracking[44]. Together with event-driven processing algorithms[45–47], DVS can achieve microsecond temporal resolution, sub-millisecond latency and high dynamic range detection (> 120 dB) with minimal data output at low power consumption[38,41,48]. Therefore, DVS are highly suited to high-speed and real-time applications requiring low power in environments with uncontrolled light levels, such as in robotics[49], autonomous driving[50] and space flight[51], as well as finding applications in microscopy[52,53] and astronomy[54]. In this work, we use an event-based camera (EBC) with integrated DVS to monitor the motion of an array of levitated particles with a bandwidth high enough to demonstrate real-time simultaneous feedback control of multiple uncoupled particles in an array.

We implement cold damping feedback[55] to cool the motion of the levitated particles[56,57], a technique with demonstrated quantum ground state cooling capabilities[16–18]. We cool a single microsphere to a temperature of a few Kelvin and a single degree of freedom of multiple particles. In this work, the number of objects we can simultaneously control is only limited by the number of output channels in our

[1]Department of Physics, King's College London, London, UK. [2]Wright Laboratory, Department of Physics, Yale University, New Haven, CT, USA. [3]London Centre for Nanotechnology, Department of Physics, King's College London, London, UK. ✉e-mail: yugang.ren@kcl.ac.uk; james.millen@kcl.ac.uk

feedback electronics. This single-device method for cooling and controlling particles in arrays is readily scalable due to the low data output of neuromorphic detection. Arrays of cooled micro-sensors will lead to enhanced signal-to-noise sensing through sensor fusion[58–60], enable force gradient sensing[25], and provide a larger interaction area without increasing the mass of the sensor[11]. Due to the low power consumption of neuromorphic detectors, our presented methods are ideal for integration into chip-scale technology[61].

## Results

### Neuromorphic imaging of levitated particle arrays

We levitate arrays of charged 5 μm diameter silica microspheres in a linear Paul trap under vacuum conditions[62–65] (see 'Methods') and record their motion using an EBC[66], see Fig. 1a. The charged particles form a stable array due to the Coulomb repulsion between them. Our particular Paul trap geometry (Fig. 1b) and the particles' distribution of charge, means that our particle arrays are non-uniform.

The EBC uses a neuromorphic DVS, which is an array of independent pixels featuring contrast detectors that output an event[38] in response to light levels on the pixel crossing a user-defined threshold. Pixels which do not experience the required level-change output no signal, removing the data-redundancy present in conventional cameras[38], while allowing the full sensor to be used at all times, sometimes referred to as a dynamic region-of-interest. The EBC hardware bundles asynchronous events into equal-length frames, and uses filters to identify objects within its field-of-view[44], after which a proprietary generic tracking algorithm (GTA) tracks the motion of each object independently. We have previously demonstrated single-particle object tracking with an SNR above 35 dB, and for a more detailed analysis of EBC performance in the context of levitated microparticles, see ref. 66.

The EBC allows the tracking of multiple objects with a high bandwidth and a linear scaling in data output with the number of tracked objects, as compared to a rapid increase in data volume with increased region-of-interest in a conventional camera. In our system, with fixed magnification, tracking a single particle at 1 kHz using the entire field-of-

view (3.75 mm²) of the EBC uses ~100 kB s⁻¹ as compared to 64,800 kB s⁻¹ using a standard CMOS camera (Thorlabs CS165MU/M) at the same frame rate and field-of-view. This means that the EBC can track many hundreds of particles before the data volume becomes comparable to standard camera technology. By not having to restrict the region-of-interest, the EBC can track objects dispersed over several hundred micrometres whilst retaining high spatial resolution[66]. The low data volume leads to a correspondingly low power consumption, see Supplemental Materials S5 for more details.

In Fig. 1c, we show an example of a single neuromorphic sensor being used to detect and track 10 levitated particles simultaneously. The EBC identifies each object and tracks it in 2D (illustrated by the coloured boxes), assigning each one a stable identification number allowing us to process the position data of each particle independently. The linear Paul trap defines the coordinate system $\{x, y, z\}$ for the levitated particles, Fig. 1b. The image on the EBC has a coordinate system $\{y', z'\}$, Fig. 1a. The $\{x, y\}$ axes are projected at 45° onto the $y'$-axis, and the $z'-$ and $z$-axes are parallel, see Supplementary Materials S1. This projection allows us to detect all three axes of motion of the levitated particles.

In Fig. 2a, we reconstruct the motion and relative position of four levitated particles obtained from the 1 kHz tracking algorithm of the EBC, which can be accessed in real-time. Particles levitated by a Paul trap undergo harmonic motion. In Fig. 2b, we generate the power spectral density (PSD) from the motion of each particle in the array of four to analyse their motion in frequency space. Each particle has a different charge-to-mass ratio (see Supplementary Materials S3) and is levitated in a different part of the confining field, meaning each particle has different frequency modes of oscillation, which are well separated under vacuum conditions. Below, we use this fact to independently cool the motion of multiple particles.

The four particles are aligned along the $x$-axis. Motion of the charged particles in this direction leads to coupling between them via the Coulomb interaction[31–33,67]. This is evident via the collective modes $x_C$ seen in Fig. 2b. By controlling the separation between the particles, we can control the coupling strength, see Supplementary Materials S2 for further details. For the multi-particle cooling presented below, we work in a regime where the coupling between the modes is too small to measure, and perform cooling along the $z$-direction.

The signal-to-noise for the different particles varies due to non-uniform illumination and varying coupling to electronic noise. It ranges from 3 to 30, with exact values given in Supplementary Materials S3. All data in this work is taken at gas pressures of $2.0–4.5 \times 10^{-2}$ mbar unless otherwise stated.

### Single particle cold damping using neuromorphic imaging

There are many reasons why it is desirable to control the energy of a levitated particle. Although the sensitivity of a levitated sensor does not increase through cooling[68], rapid damping of the motion increases the stability and measurement bandwidth of the system. Reduction of the particle energy to the ground-state of the levitating potential[15,17,19] opens up a toolbox of quantum control[5] and sensitivity enhancement[69,70] mechanisms.

Cold damping is a feedback method whereby a force proportional to the velocity of an oscillator opposes its motion. Depending on the phase of the feedback force relative to the motion, this method damps (cools) or amplifies (heats) the oscillator. When cooling, the input and output noise of the feedback electronics limit the ultimate temperature.

We process the position data from the EBC using a field programmable gate array (FPGA) to generate a feedback signal proportional to velocity, with variable gain and phase. This signal is filtered and applied to a control electrode, see Fig. 1a.

Figure 3a shows the PSD of a particle's measured motion along the $z$-axis as it is cooled via cold damping. The shape of the PSD from the

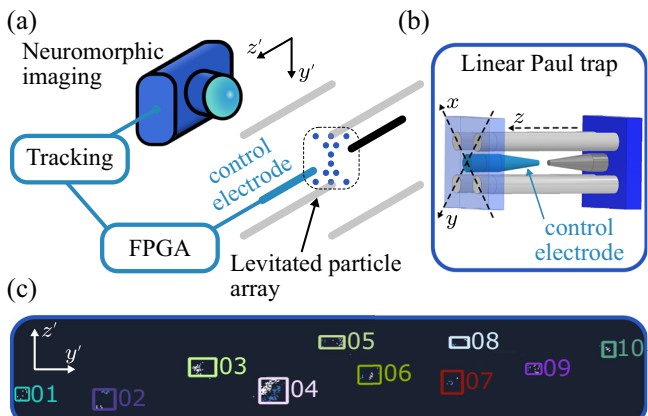

(a) Neuromorphic imaging — Tracking — FPGA — Levitated particle array

(b) Linear Paul trap — control electrode

(c) 01 02 03 04 05 06 07 08 09 10

**Fig. 1 | Neuromorphic detection of levitated particles. a** Schematic of the experimental setup: neuromorphic imaging via an event-based camera (EBC) tracks the positions of particles levitated in an array by a linear Paul trap (four grey electrodes, black and blue endcap electrodes). An FPGA (Field Programmable Gate Array) system processes this data to generate a feedback signal, which is applied to a control electrode (blue) near the particle array. **b** A schematic of the linear Paul trap, including the coordinate axes $\{x, y, z\}$ for the levitated particles, in contrast to the imaging coordinates $\{y', z'\}$ in (**a**). **c** An EBC image of an array of 10 particles. The EBC identifies each particle and tracks its motion (within the marked coloured box), assigning each particle an individual object ID (coloured numbers) which tags the streamed data. Each object is represented by only a few pixels (white for increasing intensity, blue for decreasing intensity), making the volume of streamed data very low.

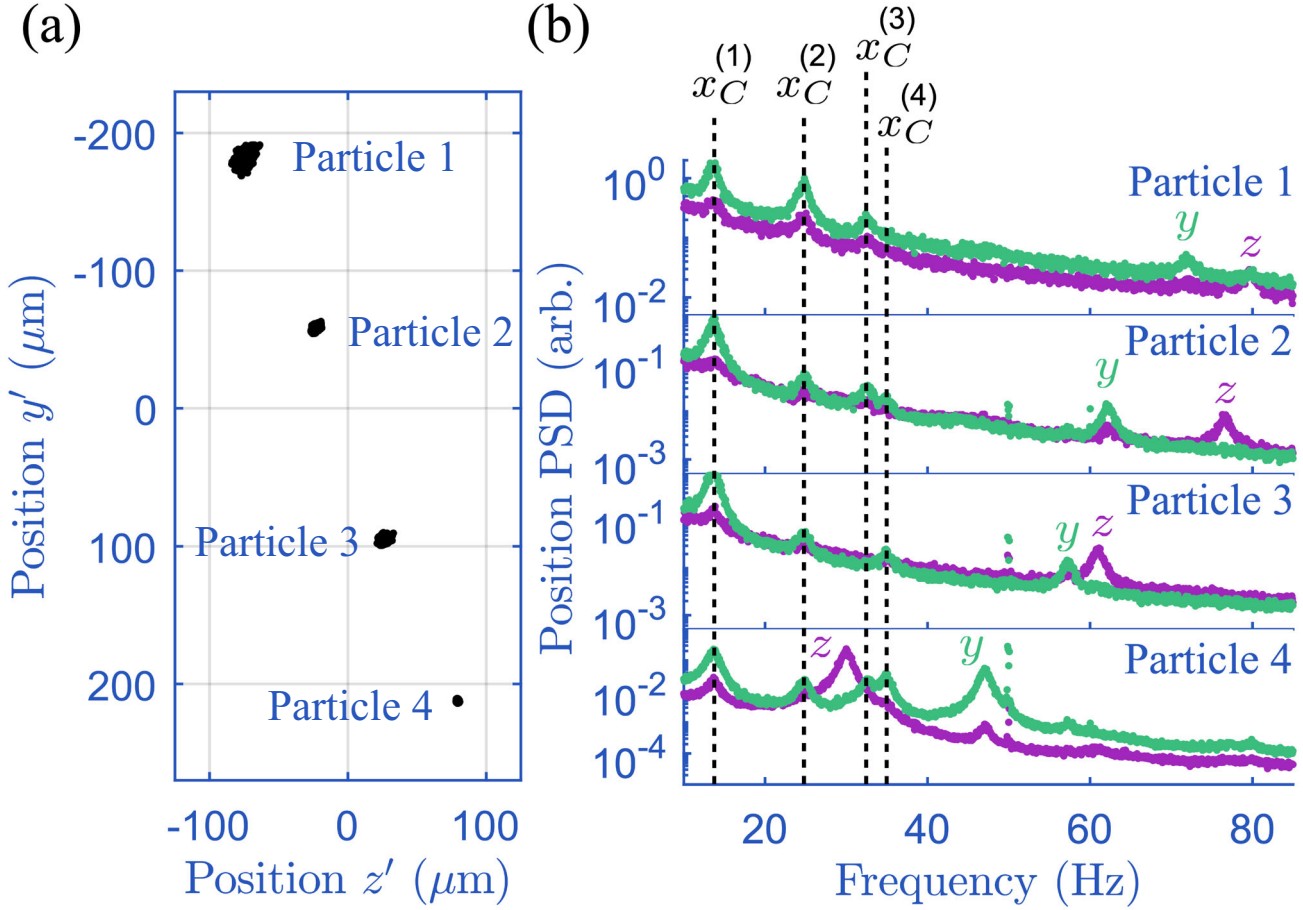

**Fig. 2 | Real-time tracking of multiple levitated particles. a** Reconstructed 2D position of four levitated microparticles tracked in real-time at a 1 kHz framerate. The amplitude of motion depends on the charge of each particle and where it sits within the levitating potential. Particles 2 and 4 are closest to and furthest from the Paul trap centre (the centre of the coordinate axes), respectively. **b** Position power spectral densities (PSDs) reconstructed from the 2D position tracking of the four particles in (**a**) (green trace: $y'$-direction, purple trace: $z'$-direction). Each particle is independently tracked. We observe interactions between particles in the array: $x_C^{(1)}$, $x_C^{(2)}$, $x_C^{(3)}$ and $x_C^{(4)}$ are four collective modes of all the four particles in the $x$-direction. We also see the individual bare modes in the $y$- and $z$-directions. For information on identifying modes, see Supplementary Materials S2 and S3.

in-loop detector is given by refs. 55,57:

$$S_{IL}(\omega) = \frac{2k_B T_0 \Gamma_0/m}{(\omega^2 - \omega_z^2)^2 + \Gamma_t^2\omega^2}$$
$$+ \frac{(\omega^2 - \omega_z^2)^2 + \Gamma_0^2\omega^2}{(\omega^2 - \omega_z^2)^2 + \Gamma_t^2\omega^2} S_{nn}, \quad (1)$$

where $\omega = 2\pi \times f$, $m$ is the particle's mass, $\omega_z$ is the mode frequency, $T_0$ is the bath temperature and $\Gamma_t = \Gamma_0 + \Gamma_{fb}(\phi)$ is the total momentum damping rate on the particle's motion, with $\Gamma_0$ being the calculable momentum damping rate due to the pressure of the surrounding gas[4] and $\Gamma_{fb}(\phi) = \Gamma_{fb}\cos(\phi + \phi_0)$ being the additional damping controlled by the feedback gain, which is feedback-phase $\phi$ dependent, with $\phi_0$ the uncontrollable phase delay caused by electronics and data processing. The term $S_{nn}$ is the feedback circuit noise that is modelled as white noise. When $\Gamma_{fb}$ is large, the feedback can introduce extra noise and lead to heating, as discussed in refs. 57,61,71. The parameters $T_0, \Gamma_0$ and $\Gamma_{fb}(\phi)$ can be extracted from a measured PSD by fitting Eq. (1) to the data. Due to voltage noise from the amplifiers driving our Paul trap, the equilibrium temperature of our particles without cooling ranges from $T_0 = 400–1500$ K, depending on their charge and spatial

location in the trap; hence, we express temperatures as a ratio. The equilibrium temperatures for all particles in this paper are given in Supplementary Material S6.

According to the equipartition theorem, the temperature of a levitated particle experiencing cold damping is[31,57,72]:

$$T_{CoM} = \frac{\Gamma_0 T_0}{\Gamma_0 + \Gamma_{fb}\cos(\phi + \phi_0)}$$
$$+ \frac{m\omega_z^2\Gamma_{fb}^2\cos(\phi + \phi_0)^2}{2k_B(\Gamma_0 + \Gamma_{fb}\cos(\phi + \phi_0))} S_{nn}. \quad (2)$$

Experimentally, the temperature of each mode $T_{CoM}$ can be extracted from Eq. (2) after fitting the measured PSD over the corresponding resonance peak with Eq. (1)[57]. In Fig. 3b, we show the effect of increasing $\Gamma_{fb}$ on the temperature of a single mode of a single particle.

The cooling depends on the phase between the feedback signal, which is proportional to velocity, and the detected motion of the particle. Eq. (2) is fit to experimental data as $\phi$ is varied in Fig. 3c, with $\Gamma_{fb}$ and $\phi_0$ as free parameters. The fitted value of $\phi_0$ is $370° \pm 5°$, noting that one period of phase delay does not significantly affect cooling for an underdamped oscillator[73], see Supplementary Materials S4. The value of $\Gamma_{fb}$ obtained by fitting the data in Fig. 3c

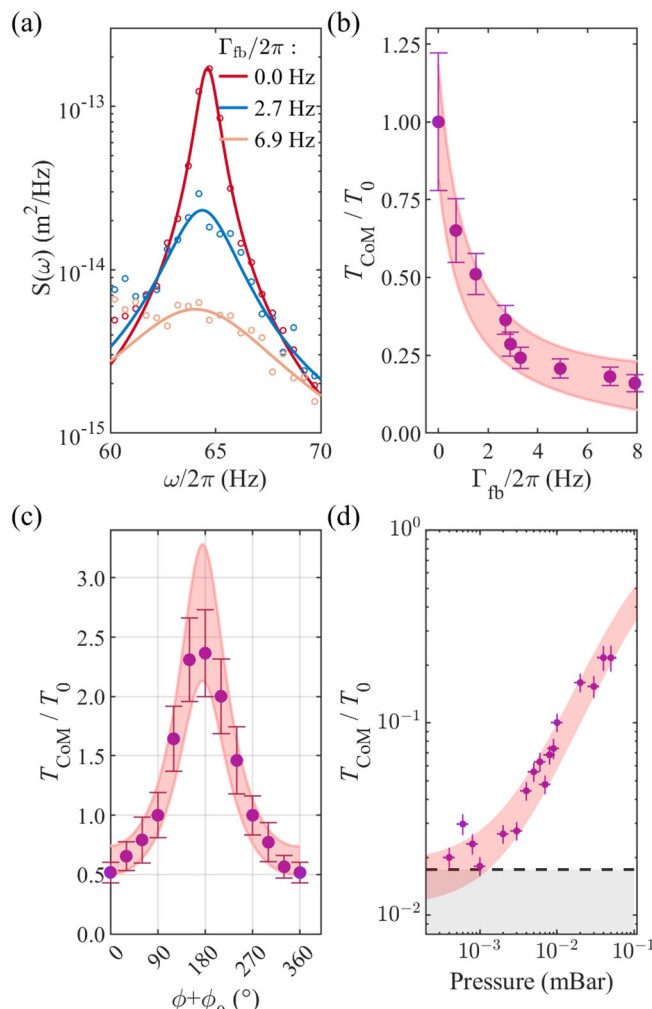

**Fig. 3 | Single particle cooling. a** A selection of single particle PSDs of the measured motion along the *z*-axis with different feedback gains ($\Gamma_{fb}/2\pi$), fit with Eq. (1). **b** Extracted centre of mass (CoM) temperature $T_{CoM}$ relative to initial temperature $T_0$ over a wider range of feedback gains than presented in (**a**). **c** Effect of the feedback loop phase delay on the temperature of the particle. **d** Variation in single particle temperature with fixed feedback gain and phase, as the background gas pressure decreases. When the pressure reaches $10^{-3}$ mbar we cool to the noise floor of our system, as indicated by the grey shaded region, corresponding to $T_{CoM} = (6.8 \pm 0.7)$ K. **b**–**d** include the model in Eq. (2) with the parameters extracted by fitting Eq. (1) to the PSDs of the measured particle motion—the pink shaded areas represent the uncertainty in these parameters. All experimental error bars in the figure are derived by taking 15 repeated experiments at each set of parameters to calculate a mean and standard deviation.

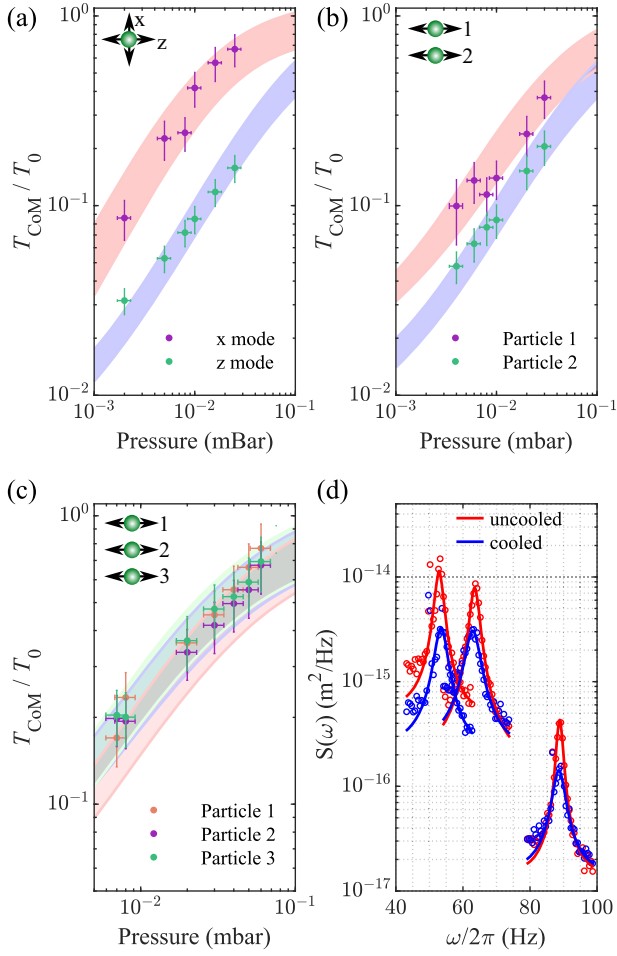

**Fig. 4 | Simultaneous multi-mode and multi-particle cooling. a** Cooling of two orthogonal modes of a single particle's motion along the *x*- and *z*-directions. **b** Cooling of two levitated particles' motion along the *z*-direction. **c** Cooling of three levitated particles' motion along the *z*-direction. **a**–**c** also include the model from Eq. (2) as shaded coloured regions, with the area representing uncertainty in our experimental parameters. **d** The PSDs of the motion along the *z*-direction of the three particles in (**c**) before and after cooling (particle 1, 3, 2 in order of increasing frequency). Solid lines are fits to the model in Eq. (1), which are used to extract the parameters used in the models (shaded regions) in (**a**–**c**). All experimental error bars in the figure are derived by taking 15 repeated experiments at each set of parameters to calculate a mean and standard deviation.

with Eq. (2) agrees with the value obtained at the same feedback gain by fitting the data with Eq. (1), $\Gamma_{fb}/(2\pi) = (0.82 \pm 0.05)$ Hz, $(0.70 \pm 0.09)$ Hz, respectively.

Finally, in Fig. 3d, we show the variation in temperature with $\Gamma_0$ by reducing the gas pressure, with $\Gamma_{fb}$ and $\phi$ fixed at around 6 Hz and 0°, respectively. At a pressure of $10^{-3}$ mbar, we reach the noise floor of our system, indicated by the grey region, at a temperature corresponding to $T_{CoM} = (6.8 \pm 0.4)$ K, representing 17 dB of cooling. The optimal $\Gamma_{fb}$ can be derived from Eq. (2), see Supplementary Materials S7. To further improve cooling, one can improve particle illumination and imaging, decrease the noise in the levitation electronics, and replace the GTA of the EBC with an optimised tracking algorithm. Object tracking has the potential to track levitated microparticles at the shot-noise limit[74].

## Simultaneous cooling of microparticles in an array

Our neuromorphic imaging system tracks the motion of every object it identifies. We are able to process this information and make a feedback loop for each degree of freedom that is detected. Each feedback loop consists of a dedicated FPGA and a set of analogue filters, and we are limited in our experiment to three loops in total. We stress that this is not a limitation of detection or processing power, simply the number of FPGA outputs available to us. For each degree of freedom the phase and gain of each feedback loop must be optimised, and filters must be set accordingly.

In Fig. 4a, we cool two orthogonal degrees of freedom (the *x*- and *z*-oscillation modes) of a single particle. We are sensitive to all three degrees of freedom due to the angle our imaging system makes to the principal axes of the trap, see Supplementary Methods S1. The geometry of our Paul trap allows the control of all degrees of freedom with a single electrode.

We optimise and fix the feedback parameters, then lower the background pressure to reduce $\Gamma_0$, hence lowering the temperature

$T_{\text{CoM}}$. The temperature is limited not by our noise floor, but by imperfect filtering, pumping energy from the feedback signal into the $y$-mode, since it is close in frequency to the $z$-mode. At low pressures, this causes the particle to become unstable, preventing us from further lowering the pressure. Cross-talk between particle modes would also limit the ultimate cooling temperature, which could be resolved by a better design of the feedback electrodes[75].

In Fig. 4b, we extend our cooling to the $z$-mode of two separate particles. As we lower the pressure, the temperature of each mode drops, reaching −10 dB and −15 dB of cooling. Since only high-pass filters are used when cooling in the $z$-direction (see 'Methods'), unfiltered noise from one particle is able to heat the uncooled modes of the other. At lower pressures, this causes particle instability and prevents further cooling.

In Fig. 4c, we cool the $z$-mode of three different particles, with the corresponding PSDs shown in Fig. 4d. To the best of our knowledge, this is the first demonstration of cooling more than two levitated particles. The issue of imperfect filtering is more pronounced when dealing with more particles, as there are more modes of the system overlapped with the unfiltered noise. We still achieve better than 7 dB of cooling. Filtering can be improved either by separating the particle modes and applying band-pass filtering, or through the use of phase-locked loops[76,77]. For the data in Fig. 4b–d, we adjust the particle spacing via the Paul trap voltages until the Coulomb interaction is weak enough such that there are no coupled modes, see Supplementary Materials S2. Collective modes in particle arrays can be cooled via sympathetic cooling in the limit that the feedback damping rate is smaller than the coupling strength between the modes[31,32].

The EBC used in this study has a sensor size of $640 \times 480$ pixels, with each particle image occupying $25 \times 25$ pixels (the coloured boxes in Fig. 1c) and having a motional amplitude of 4 pixels. As long as the centre of the particles are separated by approximately 60 pixels, the particles can be individually tracked. Hence, without changing our imaging system, we could simultaneously track up to 500 particles with this EBC. Considering the fact that object tracking allows for sub-pixel resolution[52,78], by changing the magnification of the imaging system, this EBC would be capable of simultaneously tracking at least 2000 levitated microparticles, with a correspondingly high data volume.

## Discussion

We have presented a scalable method for the detection and control of microparticles levitated in an array using a single neuromorphic detector. Neuromorphic imaging is ideally suited to this task, due to its natural affinity with detecting the motion of multiple objects[44,79] and low data-transfer rate[38]. The tracking speed in our work is limited by the proprietary tracking algorithm of the EBC. Commercial neuromorphic imaging sensors, such as the one used in this study, transfer data from the sensor to the camera hardware at GHz rates[48]. The development of custom algorithms has enabled object tracking at 30 kHz by working with the asynchronous data streamed from a DVS using only 4 MB of RAM on a standard 2.9 GHz Dual Core CPU[52]. For a fixed frame rate, the data volume is fixed regardless of the particle motion frequency. If the frame rate is increased, data are transferred more rapidly from the camera and the data volume and bandwidth increase. By pushing above 100 kHz, neuromorphic sensors would be suitable for feedback cooling optically levitated particles to the quantum ground state of motion[17], considering the shot-noise limited potential of object tracking[74,78]. This would require custom tracking algorithms and interfacing the sensor directly with FPGA or neuromorphic processing electronics[38,80], which would also enable the read-out and control of object alignment and rotation[79,81].

We believe that the particle control method presented in this work could be extended to an array of order 100 microparticles (see Supplementary Materials S8). Multichannel FPGA systems with high-

quality digital filters are a common tool in research labs. Paul traps are stable at low pressures[82], where the motional frequencies of levitated particles have sub-Hz linewidths[83]. The naturally varying charge-to-mass ratio of charged microparticles, along with application of electric field gradients, will enable the spectral separation of motional modes, making possible single-particle control and cooling even for large arrays. The neuromorphic detection and cooling presented here is independent of the levitation method, as long as there is optical illumination, meaning it is suitable for small dielectrics in optical traps, charged absorptive materials in Paul traps (such as organic material)[57], and magnetically levitated objects.

Since the motion of levitated sensors is well understood, simple machine learning could be used to optimise all of the feedback parameters[57] in an array, and optimal tracking algorithms used, which have been shown to enable quantum-level control[17]. When combined with the low power consumption of neuromorphic imaging technology (less than 30 mW per tracked particle, see Supplementary Materials S5), and great progress in chip-scale particle levitation[61], integrated devices containing arrays of quantum sensors are closer to being a reality.

## Methods
### Experimental setup
Figure 5 shows the experimental setup surrounding our linear Paul trap. The trap consists of four parallel cylindrical trapping electrodes forming a square, with two coaxial cylindrical endcap electrodes, one of which we call the control electrode, which is used for feedback control. The distance from the trap centre to the surface of the 1 mm-diameter trapping electrodes is $r = 1.15$ mm, and one opposing-pair is driven with 360 V amplitude at 1 kHz. Of the other pair of the four electrodes, the lower one has a constant voltage of 3 V applied to minimise single-particle micromotion. The two endcaps are 300 μm in diameter and separated by 800 μm, with a slight misalignment along the $z'$-axis, which causes particles to be trapped along a diagonal in the $y' - z'$ plane. The proximity of the electrodes to the centre of the trap means that a voltage applied to either one will create a field with a significant component in all three axes, enabling 3D control with a single electrode.

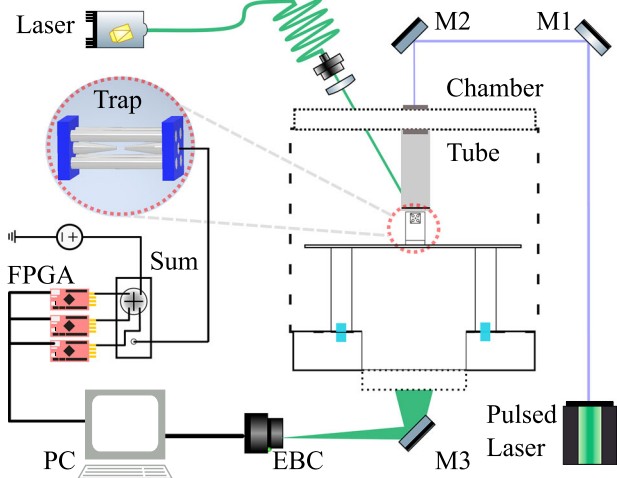

**Fig. 5 | Experimental setup for levitating, detecting and controlling arrays of microparticles.** A pulsed laser is used to launch particles into the trap via LIAD[54]. Particles are illuminated with a CW laser from above, and imaged onto an EBC from below. The EBC software runs on a PC, and the tracking algorithm outputs data to a series of FPGAs, which process the data to produce a feedback signal for each degree of freedom of each particle. Each signal is then filtered with analogue filters (not shown), and then all signals are summed together and drive the control electrode.

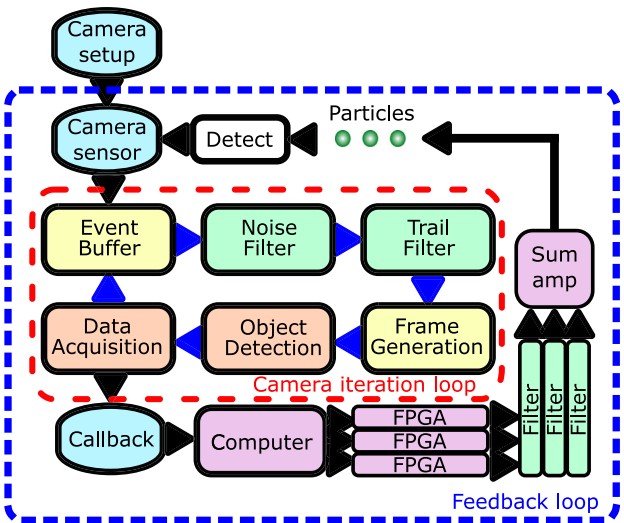

**Fig. 6 | Data pipeline for feedback control based on neuromorphic imaging.** The EBC takes asynchronous data from the neuromorphic sensor, bundles it into events, accumulates the data over time to generate frames, and then detects and identifies objects. This information is passed to a computer, which uses a generic tracking algorithm (GTA) to track the motion of each object in 2D. The tracking data for each object is split into two 1D data streams using simple Python code, and then each data stream is sent to a separate FPGA. The FPGAs each calculate the velocity from the position data, add a variable gain and phase shift, and generate voltage outputs. These are separately filtered using analogue filters, and the signals are combined with a summing amplifier, the output of which is sent to the control electrode to cool the particles.

Laser-induced acoustic desorption (LIAD)[84] is used to launch microparticles into the trap at a pressure of 1 mbar, and then the pressure is lowered to carry out the experiments presented in this manuscript. We typically trap particles of positive charge, ranging from $2 \times 10^3$ e to $2 \times 10^4$ e.

To image the particles, 18 mW of laser light of wavelength 520 nm is weakly focused onto the array. The scattered light is imaged onto an EBC (Prophesee EVK1 -Gen3.1 VGA (camera sensor: Prophesee PPS3MVCD, 640 × 480 pixels)) using a long working distance microscope. The camera is precisely calibrated using a method outlined in detail in ref. 66. When dealing with multiple particles, calibration is performed via displacing a translation stage on which our imaging system is mounted by a known amount.

### Data processing

The data pipeline is shown in Fig. 6. The operation of the EBC is described in detail in ref. 66. The GTA of the EBC outputs 2D position data for each object it detects. The EBC is communicated with via a Python script, which separates this 2D information into two 1D data streams for each object. The script passes each data stream to one of three FPGA systems (Red Pitaya STEMlab 125-14) to output the position of each particle. The FPGA clock is synchronised to the clock of the EBC to ensure timing consistency. Code on the FPGA computes the velocity from the position data and adds a variable gain and phase to the signal to generate the feedback signal. There is a latency of 10 ms in this pipeline, see Supplementary Materials S4. EBCs are available with FPGA systems on the camera hardware, which will significantly reduce this latency.

Each feedback signal is filtered with an analogue filter to isolate each frequency component of motion: a high-pass filter for the $f_z$ signal (Wavefonix 3320 HPF 24 dB per Octave) and a low-pass filter for the $f_x$ signal (Wavefonix 2140 LPF 24 dB per Octave). The filtered feedback signals are combined with a summing amplifier and sent to the control electrode.

## Data availability
The data supporting this article are openly available from the King's College London research data repository, KORDS, at https://doi.org/10.18742/28069136.

## Code availability
The code used in the present work is available from the corresponding authors upon request.

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

## Acknowledgements

The authors would like to thank Prof David Moore for useful discussions, Dr Ruvi Lecamwasam for assistance with writing FPGA code, and Dr John Dale for bringing neuromorphic imaging to our attention. This work is supported by STFC Grant ST/Y004914/1, EPSRC Grant EP/S004777/1 and ERC Starting Grant 803277.

## Author contributions

Y.R.—experimental design and build, data taking, data analysis. B.S.—experimental design. R.Y.—data taking. Q.W.—supporting simulations. J.P.—data analysis. M.R.—experimental design. J.M.—experimental design, data analysis.

## Competing interests

The authors declare no competing interests.
