## [Transparent Peer Review file · Nature Communications]

Neuromorphic detection and cooling of microparticles in arrays

Corresponding Author: Professor James Millen

Version 0:

Reviewer comments:

Reviewer #1

(Remarks to the Author)

The authors Y. Ren et al. of the manuscript Neuromorphic detection and cooling of microparticle arrays describe a novel technique to detect multiple particles simultaneously by using a single detector. This single detector is an event based camera that allows for scalable detection of many particles.

The authors employ this new technique to apply cold damping on several modes of one particle and single modes of several particles. The cooling itself does not match the state of the art, both in pressure regime and noise performance. The main result of the manuscript is the fact that several particles can be cooled, which is the first demonstration of a scalable technique that could allow the motion control of 100s of particles in high vacuum conditions.

The levitation field is currently moving towards many body physics in e.g. optical lattices or coulomb crystals. While the levitation of particle arrays by itself is already demanding, their motion control in vacuum relies on multiparticle detection and up to today no scalable technique is available. This makes the work of Y. Ren et al. very timely.

While the authors spend a lot of effort to discuss their cooling results, the discussion of the limitations and possibilities of this event based camera, their main contribution to the field, is a bit shallow. In my opinion, the authors could answer some key questions concentrating on the detection aspect:

1. What is the detection efficiency or signal-to-noise ratio in the current experiments? What is the main limitation?
2. What kind of improvements are needed to reach ground state cooling with event based cameras or is this out of reach?
3. With the employed event-based camera, how many particles can be controlled? Is there a trade-off between detection quality and particle number?
4. Ultimately, 3D detection will be needed. What are the possibilities and their consequences to implement 3D detection with the current setup?

Nevertheless, in my view this approach is promising and innovative and with some revision I believe the manuscript could merit publication in Nature Communication. Below the authors will find general and specific comments/suggestions, that I hope the authors will find helpful.

Quality of the data: The presented data, theory and conclusions on cooling is, except of a few points highlighted later, valid and robust. Nevertheless, data supporting the performance and limits of the new detection technique is not very abundant and would be valuable to add. This could also raise the significance of this work.

Appropriate referencing: The referencing can partially improved. A few times more original and more suitable references are not cited in favour of more current ones, replicating the original ones. Furthermore, references are sometimes not complete e.g. ground state cooling or not covering the referred topic e.g. quantum enhanced sensing. More precise comments can be found in the specific comments.

General comments:

- Some sentences are disconnected and causalities between statements are sometimes not super clear, e.g.

Detecting and controlling multiple particles in vacuum has so far involved single particle control with sympathetic cooling [23, 24, 26], and arrays of optical traps [27]. It will be a challenge to levitate and individually control arrays of tens, or even thousands of objects [9].

Why is it a challenge to control more particles with the referenced methods? This comment applies more strongly to the first part of the manuscript.

- The authors trap charged particles in a Paul trap that are coupled via Coulomb interaction.

How can the effects of sympathetic mode cooling be excluded?

- The authors mention that they apply additional filters to their feedback signal. It is not clear to me what the goal or figure of merit of this filtering is. Could the authors add this information to the manuscript?

- The authors refer to arrays of tens of particles. It is not clear to me if they mean an array of e.g. 3x3 or 10x10. Could the authors be more specific in the manuscript by giving the dimension of the hinted arrays?

- The detection method is inherently 2D and allows for 3D detection only at the price of misalignment. Can the authors discuss the limitations if their method is used for the ultimately required 3D detection.

- Can the authors discuss what the minimal distance on the detector sensor is before cross leakage between different particles occur and what this means for the maximum particle number? The signal of one particle is spread over how many pixels in this work?

- The detection bandwidth is high enough for Paul traps with lower mechanical eigenfrequencies.

What about optical traps? Can the authors state a value for the bandwidth in the manuscript?

- I believe there is not a single Paul trap reference. Could this be added?
- Suggestion: All equations are in terms of ω and plots are in terms of f . In my opinion it helps preventing misunderstandings if this is adjusted to one of the two cases. It is also not quite clear if quantitative values for Γ_{fb} are now $\Gamma_{fb} = 4\text{Hz}$ or $\Gamma_{fb} = 2\pi \times 4\text{Hz}$.
- A few typos can be found: Camers, Proprietary

Specific comments:

- In the abstract the authors claim that an array of particles increases the sensitivity. Why is that?

- I disagree with the following statement

... The performance of a mechanical sensor is inversely proportional to its volume due to dissipation through thermal contacts and surface strain ...

I claim that the acceleration or force sensitivity of a thermally limited harmonic oscillator is given by $S_{aa} \propto \sqrt{k_B T \Gamma / m}$ or $S_{FF} \propto \sqrt{k_B T \Gamma m}$ which is related to the mass and not the volume. Maybe the authors could clarify their statement to avoid misunderstanding.

- I claim the following statement is only true at ultrahigh vacuum

... By levitating nano- or micro-particles in optical, electrical or magnetic fields, one creates a mechanical oscillator with remarkably low dissipation ...

Could the authors clarify this?

- There exist older proposals that suggested macroscopic quantum physics with levitated particles than [16]

... opens the door to macroscopic quantum physics [16]. ...

as for example [Chang, Darrick E., et al. Proceedings of the National Academy of Sciences 107.3 (2010): 1005-1010] and [Oriol Romero-Isart et al 2010 New J. Phys. 12 033015].

- Are dynamics vision sensors and neuromorphic sensors the same?

- A word is missing

... Therefore, neuromorphic detection is highly suited to high-speed and real-time applications requiring low-power in environments with uncontrolled such ...

- In my opinion is the chosen reference for active feedback via cold damping unsuitable.

... We implement cold damping feedback to cool the motion of the levitated particles [40], ...

Instead the following should be cited [Tebbenjohanns, Felix, et al. "Cold damping of an optically levitated nanoparticle to microkelvin temperatures." Physical review letters 122.22 (2019): 223601], [Conangla, Gerard P., et al. "Optimal feedback cooling of a charged levitated nanoparticle with adaptive control." Physical review letters 122.22 (2019): 223602] and [M. Poggio, C. L. Degen, H. J. Mamin, and D. Rugar Phys. Rev. Lett. 99, 017201].

- I suggest to add the word of L. Magrini et al too.

... technique with demonstrated ground state cooling capabilities [41]. ...

- Could the authors add more quantitative information related to the following statement?

... This single-device method for cooling and controlling particle arrays is readily scalable due to the low data output of neuromorphic detection. ...

More precisely, how many particles can be tracked? And what is the limitation e.g. the sensor size or data flow?

- Could the authors elaborate on the following statement?

... Arrays of cooled microsensors will lead to enhanced sensitivity [42], ...

In my opinion, several coupled oscillators allow for the implementation of the mode localization method, but this method is not increasing the sensitivity of a thermally driven harmonic oscillator as such. Could the authors clarify their statement?

- In Fig 1a), what is the black electrode? The particles are depicted in a circle. Shouldn't the particles form a 1D chain?
- I find the relative panel size of Fig 1 unlucky, given that panel b) is the most important panel. Some boxes in Fig1b) show "two" particles. Why is that and what is the consequence?
- It is not clear of which variable the PSD is calculated:

... In Fig. 1(c) we generate the power spectral density (PSD) ...

- The relative position of the particles is unclear.

... for each particle in an array of four ...

Is this a 1x4 or 2x2 array?

- Could the authors elaborate more on this?

... Interactions between the particles are evident, as shown by the unlabelled peaks in Fig. 1(c). ...

I believe some peaks could also be higher harmonics of a certain mechanical eigenmode. I claim that interactions should lead to coupled oscillators with peaks at the sum and difference of the peaks, which I can not identify. If the authors have identified the origin of additional peaks, it would be helpful to add additional labels.

- Is the Paul trap symmetric in x and y? If so, could the authors elaborate why all x-modes are equal in frequency but the y-modes are different?
- I disagree with this statement:

... By reducing the position fluctuations interactions between the particles can be minimized. ...

I claim that the interactions can be stabilized due to fixed positions but this does not imply that interaction strengths are decreased.

- The authors should clarify that they mean active feedback methods.

... Reduction of the particle energy to the ground-state of the levitating potential [14, 46] ...

Otherwise, citations using cavity cooling are missing.

- References incomplete or misleading:

... opens up a toolbox of quantum control [47] and sensitivity enhancement [6]. ...

Depending on what the authors mean with quantum control, the references should be more towards the manipulation of mechanical oscillators in the quantum regime; or if the authors only refer to cooling, then the reference list should include all 1D cooling experiments too.

Concerning [6], I claim that the reference does not cover quantum enhanced sensitivity but instead is a classical force sensing experiment. An example of quantum enhanced sensing would be the exploitation of squeezed mechanical states.

- Original references on cold damping are missing:

... Cold damping is a feedback method ...

E.g. [Tebbenjohanns, Felix, et al. "Cold damping of an optically levitated nanoparticle to microkelvin temperatures." Physical review letters 122.22 (2019): 223601], [Conangla, Gerard P., et al. "Optimal feedback cooling of a charged levitated nanoparticle with adaptive control." Physical review letters 122.22 (2019): 223602] and [M. Poggio, C. L. Degen, H. J. Mamin, and D. Rugar Phys. Rev. Lett. 99, 017201].

- I believe in Eq. (1) it should be at least hinted that there is a noise term being neglected, that heats the particle motion for large gains (a regime that is explored here too). For more details see Eq. (4) in Conangla et al., PRL 122, 223602 (2019). Especially, because the noise term appears then suddenly in Eq. (2).

- In my opinion there are earlier references for Equation (2), namely Conangla et al., PRL 122, 223602 (2019).

- ... In Fig. 2(b) we show the effect of increasing feedback gain on the temperature of a single mode of a single particle, and compare the data to the model in equation (2), with only S_{nn} as a free parameter. ...

I disagree with this statement that S_{nn} is the only free parameter since the same data has previously been used to extract T_{com} and Γ_t . It would be also interesting if the fitted value for S_{nn} coincides with the noise extracted from the detection with the EBC and how this compares to standard photodiodes.

- The authors are fitting Γ_{fb} twice.

... This model is fit to experimental data, Fig. 2(c), with Γ_{fb} and ϕ_0 as free parameters. ...

Do the values coincide?

- In Fig 3, why are two particles colder than 3 particles if the particle detection is independent?
- It is not quite clear to me what imperfect filtering shall mean. Is this due to the fact that the detection with the camera does not deliver independent detection signals, or because the filters are too steep such that unwanted phase shifts are occurring or is it because only one single electrode is used, which leads to cross talk in the electric fields?
- Can the authors elaborate more on this statement?

... Different particles can have different noise floors, since their coupling to voltage

noise depends on the particle charge. . . .

In my opinion, the noise floor is particle independent and is the noise in absence of the particle motion. While I agree that the amplitude of the motion might depend on the particle charge and therefore the SNR, I disagree that the noise floor should depend on the particle charge. Could the authors explain this in more detail?

- . . . The issue of cross-talk is more pronounced when there are more modes of the system, yet we still achieve better than -7 dB of cooling . . .

Can the authors specify at which point this cross-talk happens, e.g. in the detection, in the summation of the individual feedback signals, or elsewhere.

- Phase lock loops have been in use for a long time. Maybe reference [50] is not the most suitable one.

- If I understand correctly, the independent detection of many particles with a single device is the main result of the paper, which is independent of the levitation technique. I find the following sentence therefore misleading since the charge is not needed for detection:

. . . It's also worth mentioning that our method is suitable for any object which can carry a charge, unlike optical trapping with intense beam detection which is limited to low-absorption dielectrics. . . .

- In Fig. 3d), which particle is which in Fig 3c)? Does the final temperature match the expectation from the SNRs in Fig. 3d)?

Reviewer #2

(Remarks to the Author)

This work presents a method for detecting and controlling arrays of levitated particles using an event-based camera, referred to by the authors as a neuromorphic detector. However, the demonstration is limited to a small scale with relatively low frame rates. While the authors suggest that the approach could be scaled up, this claim remains unsubstantiated.

From a neuromorphic technology perspective, the report lacks crucial details and benchmarks. There is an overemphasis on the potential advantages, which are not demonstrated. If an event-based detector is combined with a GPU or any other digital ASIC, the question arises: what fundamentally differentiates this from conventional methods? Additionally, the specific hardware and neuromorphic components employed are not sufficiently detailed. For example, while an FPGA is used alongside the detector, the overall energy consumption of the system must be thoroughly evaluated before labelling it as a low-energy platform. Furthermore, the time complexity of the entire computational process is not computed and unclear where it stands.

It is also unclear what specific neuromorphic hardware or algorithms are applied, and why they would provide a superior result. How does the resolution of this measurement depend on the hardware accuracy? Without hardware details, these are difficult to gauge. The absence of detailing and benchmarking makes it difficult to determine whether this approach offers any improvement relative to any alternative technologies.

In short, without more comprehensive engineering details and transparency, the paper appears to focus on a niche application and relies heavily on technical jargon, potentially overstating its contributions.

Reviewer #3

(Remarks to the Author)

In this work, the team of James Millen offers a brand-new detection technique to monitor the motion of multi-particle systems in levitation, which relies on a neuromorphic camera detection. On a conventional Paul-trap platform, allowing for the levitation of charged particles, they first demonstrate that neuromorphic imaging performs efficient multi-particle motional detection associated to a relatively low data transfer. Afterwards, relying on this detection scheme, they implement cold damping on a single particle before cooling arrays of particles or several motional modes of a single object.

If I find that this paper definitively has some merits in terms of innovation and is scientifically robust, I am not one hundred percent in line with some of the claims that are made and it is not clear to me what is new here. Moreover, I find that some of the comparisons that are made with current techniques are not always totally fair. Therefore, I would like the authors to address the list of concerns/comments provided below before taking a decision on whether this paper should be accepted for publication in Nature Communications.

Major comments:

[1] For me, one of the major concerns is that it is not clear to me what is new here. If the neuromorphic detection is clearly brand-new, the cooling scheme corresponds to the well-established cold-damping approach that is here multiplexed in frequency. The neuromorphic signals are filtered in frequency and the cold damping is applied as usual on the remaining signal. How is that different from the cooling of multiple particles performed for instance in ref 27? Can the authors stress exactly what is and what is not new?

[2] My second major concerns regards the limitations of the technique and the claim that it can cool arrays of particles. Unless I missed something, if the neuromorphic detection can track large sets of particles with any mechanical frequency within its bandwidth, the cooling scheme can only address degrees of freedom that are well separated in frequency: one cools frequencies that very weakly overlap spectrally. Thus, the coupling between particles is very small. This can be observed in Fig 3d, in which the three cooled modes barely overlap. This is also stressed by the authors when they explain that cross talk between the modes degrades cooling and renders the particles unstable.

For me this inability to cool strongly coupled modes enters in contradiction with the claim of cooling “arrays of microparticles”. Indeed, arrays need their elements to talk to one another. In the introduction, the authors cite many papers in which arrays are made of mechanical resonators with nearby resonances (e.g., 21 or 42 to name a few). Moreover, all the different quantum applications like entanglement (ref 17) that are listed, rely on strong cross talk between the particles. Don't get me wrong here: I am not questioning the quality of the work; I am stressing that the claim of cooling arrays of microparticles should be adapted.

[3] Along the same line, the authors try to demonstrate that some degree of interaction exist between their particles (which it must, of course). For that, they claim that the “non-labelled” peaks in Fig. 1(c) are a signature of it. Here, I have some doubts as clearly some peaks correspond to harmonics of the fundamental resonances (e.g. x on the bottom panel). To me the presence of extra peaks does not stand for a proof of particles' interactions as it can simply be related to non-harmonical trapping within the confinement of each particle (that leads to the formation of any combination between fx, fy and fz). In short: of course the particles talk to one another but I am not sure that these extra peaks are a signature of it.

[4] Even though I like the neuromorphic approach, I find the comparison with other techniques a bit unfair. At first, the authors state themselves that it suffers from nearly millisecond latency, which I suspect is one of the reason why the authors used ultra-low mechanical resonances (below 100 Hz). Thus, the usage of neuromorphic detection for 100 KHz resonators and above (which form the core of the research activity in optical levitation) is far from obvious to me.

More importantly, the authors claims:

” Tracking of a single particle using the entire field-of-view of the EBC used $\sim 100 \text{ kB s}^{-1}$ as compared to $64,800 \text{ kB s}^{-1}$ using a standard CMOS camera with the same sensor size.“

I find this comparison a bit unfair as the field of view of a camera can be as well split in Regions Of Interest (ROIs), which makes the data required to monitor the motion of one or several particles totally manageable (as it is common practice in other fields of physics).

[5] Also, I find the “easily-scalable” selling point a bit too much. One degree of freedom requires one FPGA and multiplying FGPA's comes with a great cost in terms of complexity. No solution is perfect and if the authors want to put forward the scalability aspect, I would recommend that they perform a fair comparison with other techniques like the one described in ref 27.

I also add some minor comments:

[6] In the introduction, it would be appreciated to complete the citation of experiments having reach the ground state (ref 14, 15) by the work of F. Marin:

<https://doi.org/10.1103/PhysRevResearch.4.033051>

Also, alongside ref 27, the authors should mention the many-body cooling approach proposed by S. Rotter:

<https://doi.org/10.1103/PhysRevLett.130.083203>

[7] At last, on the neuromorphic approach, can the authors explain if the data transfer rate increase with larger mechanical frequencies (kHz instead of Hz)? Will the data bandwidth be affected? Also, can the neuromorphic approach detect the rotation of the particles? What is the optimal “misalignment” one should impose to get the best detection of translation along x, y and z?

Version 1:

Reviewer comments:

Reviewer #1

(Remarks to the Author)

(Please find attached)

Reviewer #2

(Remarks to the Author)

I have reviewed the responses to the referees, and my primary observation is that the authors need a stronger understanding of general computing terminology and, more specifically, neuromorphic computing.

It appears the authors are utilizing hardware without fully grasping how it operates. There are numerous neuromorphic hardware platforms and algorithms aimed at improving noise performance and reducing computational complexity. The response regarding computational complexity is particularly misaligned and highlights a lack of perspective on how computation is actually performed.

In short, this implementation doesn't really bring anything noteworthy to the table from the perspective of neuromorphic hardware. I recommend that the authors familiarize themselves with the following references to gain a better understanding

of computation:

1. <https://www.nature.com/articles/s41586-018-0180-5>
2. <https://pubs.aip.org/aip/apr/article/7/3/031301/997525/Analog-architectures-for-neural-network>
3. <https://www.nature.com/articles/s41586-024-07902-2>
4. <https://www.nature.com/articles/s41586-019-1677-2>

Reviewer #3

(Remarks to the Author)

I read with a lot of scrutiny the reply, which was provided by the authors.

To give a bit of context, I would like to stress that I do have a lot of respect for the work of James Millen. Moreover, I believe that I provided a descent and very positive review, which was meant to improve this article and I spent a lot of time casting it.

Sadly, after reading the reply, I do have the feeling that my comments have been treated lightly. In addition, I do not agree with some of the physical arguments that have been brought forward. More importantly, no data or any new scientific material have provided. So, with the present reply, I cannot accept this article for publication.

My refusal is based on scientific and (let say) "less-scientific" aspects that I list below:

"Less-scientific" comments:

- First, the "quality" of the reply was of a very low standard. The authors did not list the sentences that have been changed (apart from one). They seemed to expect that I would scroll the whole text for each of their reply in order to guess which sentence was changed and for what reason.

- No new data or any other kind of material has been added to the manuscript. The authors only provided text with unbacked claims and asked me to believe them. For instance, in my comment [3] of the former review, I raised a reasonable concern regarding the nature of the different peaks in the spectrum. This point is not minor at all. As a reply, the authors claimed that they implemented two techniques to confirm the nature of the peaks but without providing any data or figures to back it up. I quote:

"We can say with absolute confidence that the particles are interacting. We hope that the reviewer agrees that we do not need to add extra data to the manuscript [...]"

Maybe adding extra material to the manuscript is much (even though I doubt it). Yet, they should have provided those data to the supplements or the Methods, and more importantly, I should have access to them. They cannot ask me to believe something that I have not seen.

- As a last example, I asked in my comment [7] some questions that were meant to improve the paper and addressing them in the manuscript would have been a strong plus. Here, I would like to quote the first sentence of their answer to my comment:

"We thank the reviewer for these interesting questions! I feel this is for discussion rather than alterations to the manuscript. " After this introduction sentence, they wrote a few lines and did not make any change to the text. I spent a great deal of time reading and reviewing this paper for free. Thus, a small talk is not what I was looking for.

Scientific comments:

Here, I list some comments regarding the answers provided. Numbering stays the same as prior.

[1] The authors claim that they perform detection over the motion of multiple particles and that "cooling is not necessarily required". The way I understand this point is that tracking is the most important input of this method. And I agree that this is impressive! Yet, in the following answers, they use cooling as a major argument (even as part of their title). Thus, I have the feeling that the role of cooling is downplayed on purpose here to please me here before being put forward further down the line.

Also, I do not have the time to scroll the text to check all the blue sections and try to guess which ones are supposed to have been adapted. Please make a detailed list.

[2] Here, I don't agree with the different points brought forward by the authors.

They claimed that levitated particles can have micro-Hz linewidth and that they can be separated spectrally to be cooled. Just as a rule of thumb, if I assume that I collect a PSD in which two "peaks" are spaced by a few micro meters, the time trace will look like an almost perfect sinus with an extremely low envelope modulation. To acquire a PSD with two peaks one would need seconds of acquisition. To fully measure a micro-Hz linewidth you need 10^6 seconds. You can infer it faster, but recent work like the one from Tracy Northup's group require way more than a few ms to do so. Thus, I have reasonable

doubts that using hundreds of particles oscillating at 1 kHz and spaced by even a few Hz would work since everything must happen way faster than a single oscillation period.

Also, the argument that you could use a PLL to pick the individual narrow linewidth is also not convincing to me. In order to lock, a PLL needs at least five oscillations (if you are very good). If you have frequencies that are very close in frequency I really doubt that would work or that you could converge fast enough.

At last, I don't get the nuance when changing the claim "to cool particles arrays" into "cool particles in an array". The authors have to define exactly what they mean here.

[3] Here, as discussed above, the authors have to provide data of what they claim. They claim that they have implemented "two methods confirming the modes [they] observe". I am more than ready to believe them but not without data and evidence. And those data have to be incorporated in the Methods or into a Supplementary Material.

Same thing when they claim that they can "switch the interactions on-and-off". I cannot trust something that is not proved.

[4] The authors acknowledge that they do have a latency problem but that FPGAs will reduce it. Here, no quantification of how much is to be expected. Also, they claim that a latency in the cooling of a few cycles is harmless by citing a paper that stresses the fact that delays actually change all the thermodynamics of levitated objects [Debiossac et al.] (i.e., substantial changes in the dynamics). Thus, a clarification would be welcome here.

Regarding the ROI vs EBC comparison, I find the edits to the text still biased. I would agree with a statement saying that ROI's scale quadratically with the number of particles while EBC scales linearly. Yet, saying that the data rates are 100 kB/s versus 64 000 kB/s is still misleading.

[5] Regarding the scalability, as discussed in [2], I do not get how stacking FPGA will help cooling as I doubt that their internal PLL could lock with a resolution low enough.

[7] As discussed previously, my comments were meant to improve the paper and not for a small talk between myself and the authors. Thus, I would appreciate if they were to take them into account.

Version 2:

Reviewer comments: Reviewer #1

(Remarks to the Author)

(Please find attached)

Reviewer #3

(Remarks to the Author)

I read the response from the authors with scrutiny and I acknowledge the efforts that have been made. If I agree with many of the replies to my comments, I would like the authors to make a few clarifications before accepting the publication of their work.

Comment 2:

I am convinced (and I was already) that the particles can indeed couple. The new data are convincing in that regard and improve the quality of the paper without a doubt. Yet, as the authors stress it at some point in the text, the multiparticles modes that are cooled need to be uncoupled. Thus, I would appreciate if this important point is mentioned early in the text. For instance, can the author modify the sentence in the abstract:

"[...] control by implementing real-time feedback to cool the motion of three objects simultaneously [...]" into

"[...] control by implementing real-time feedback to cool the motion of three uncoupled objects simultaneously [...]"

Comment 5:

I believe there is a misunderstanding between myself and the authors. Their technique (at least at this stage) requires the modes to be uncoupled (see above). Thus, they should not overlap spectrally. I believe that we all agree that, if we note Δf the linewidth of the resonances, assuming that the authors can assemble resonances over the total spectral range B , the max number of modes that can be cooled scales as $B / \Delta f$. Thus, I do understand that taking narrower resonances increases the number of modes that can be cooled.

My point relates to the fact that Δf cannot be arbitrarily small. Let's assume that they have two modes that are ultra-narrow and very close in frequency. As suggested in the reply, I can admit (assuming no experimental drift) that they start with a routine that measure extremely precisely the resonance frequencies of those modes. The cooling of these two modes will be achieved using an electronic signal that encapsulate the carrier frequencies of both modes. Yet, this signal has a certain time duration, meaning that it will possess a non-negligible spectral width that limits the spectral resolution of the technique. Therefore, it seems to me that there should be a limit in the number of modes that can be cooled (like in any

technique). In that regard, is there a way to estimate (by a rule of thumbs) of many modes can be addressed? This would for me a straightforward way to back up the claim below:

“We believe that the particle control method presented in this work could be extended to an array of order 100 microparticles [...].”

, which at this stage remains speculative to me.

Version 3:

Reviewer comments:

Reviewer #1

(Remarks to the Author)

Dear editors and authors

The authors Y. Ren et al. of the manuscript Neuromorphic detection and cooling of microparticle arrays addressed all of my questions and answered them sufficiently. Despite some level of disagreement between the authors and myself as a referee about clarification of certain details in the manuscript to avoid misunderstandings, I still believe that this approach is promising and innovative and I finally recommend the publication in Nature Communications.

Review response: *Neuromorphic detection and cooling of microparticle arrays*, Y. Ren *et al.*

We would like to thank the referees for their patience. This work was severely delayed by our lead-author having a personal emergency which meant they were away from work for 3 months.

The authors would like to sincerely thank the referees for the effort put into their reports, giving us the opportunity to improve our manuscript. When preparing this work, we faced the challenge of using a commercial event-based camera with some proprietary functions, and wanted to avoid simply evaluating the particular model of camera we were using. In addition, our previous work on this topic [Ren *et al.*, *Appl. Phys. Letts.* **121**, 113506 (2022)] went into depth with evaluating the detection method. We believe that there was great value in presenting neuromorphic imaging as a *general* control method for particle arrays, and we thank the referees for assisting us in better finding the balance.

We hope the editor and referees note that significant changes we have made to the manuscript. Changes in the manuscript have been highlighted in blue, excepting typographical, referenceing or language-clarity points.

Response to Reviewer 1

We thank the reviewer for recognizing the value and robustness of our work, and for giving us the opportunity to improve the manuscript. We faced a challenge in quantifying the behaviour of event-based imaging, since we are using a specific device and did not want to simply benchmark one commercial product, but rather demonstrate the utility of this type of imaging. The way that the imaging works is unusual compared to other techniques, meaning that it isn't always easy to quantify performance against, for example, a photodiode. Finally, we utilize the tracking algorithm integrated with the camera, which is proprietary, so again not something we wanted to benchmark. We have more quantitatively assessed event-based imaging elsewhere [Ren *et al.*, *Appl. Phys. Letts.* **121**, 113506 (2022)].

Saying all this, the reviewer's questions have enabled us to convincingly make statements about the performance of our system, greatly strengthening the manuscript.

1. What is the detection efficiency or signal-to-noise ratio in the current experiments? What is the main limitation?

Event-based imaging is well known to offer a high-dynamic range, since instead of recording images, it registers events [e.g. G. Gallego *et al.*, *IEEE Trans. Pattern Anal. Mach. Intell.* **44**, 154–180 (2022)]. Our particular camera can offer a SNR (defined as the range between the low- and high-light cut-off) of up to 39 dB, and a dynamic range up to 120dB. This depends sensitively on a host of parameters, which we optimize for our application, and are camera dependent.

In the current experiment, our SNR is up to 30dB, and in previous work, where we optimized our system to detect a single particle, we achieved a SNR over 35dB [Ren *et al.* *Appl. Phys. Letts.* **121**, 113506 (2022)]. Here, by SNR, we mean the detection of the resonant motion of

the particle above the base-level noise, but this noise is not due to the camera, it is due to electronic noise on the electrodes which are suspending the particle. The “base level noise” of the camera is subtle, since it is either tracking an object, or it is not. The resolution with which we can track the object depends on the magnification of our system (which is lower than our previous work so that we can track multiple particles), and the parameters of the tracking algorithm, which is proprietary to the camera and unfortunately a “black box” in this case. In future work we intend to work with an FPGA based event-camera system to write our own tracking algorithm, which would allow us to benchmark its performance.

However, I would note that work by others have shown that camera based imaging with object tracking can reach the shot-noise limit [Werneck *et al.*, *Rev. Sci. Instrum* **95**, 073708 (2024)], and there is nothing about event-based imaging which would suggest this shouldn't be possible.

We have added the following to the manuscript:

“We have previously demonstrated single-particle object tracking with an SNR above 35 dB, and for a more detailed analysis of EBC performance in the context of levitated microparticles see [44].”

“Object tracking has the potential to track levitated microparticles at the shot-noise limit [Werneck *et al.*, *Rev. Sci. Instrum* **95**, 073708 (2024)]”

2. What kind of improvements are needed to reach ground state cooling with event based cameras or is this out of reach?

There are two aspects to this question – is it possible to cool to the ground state in our experiment, and is it possible to cool to the ground state using event-based imaging. Considering the former question, our noise-floor is very high (due to voltage noise on the supplies driving the electrodes which levitate the particles), around 0.5K. This could straightforwardly be improved, but for a 100Hz oscillator the ground state is below 0.1 micro-Kelvin, much colder than the state-of-the-art cooling in the literature of a few hundred micro-Kelvin. So, our system of a levitated charged microparticle in a Paul trap would be extremely hard to optimize to the level to achieve ground state cooling. When trying to cool to the ground state, the community typically uses optical levitation as the >100 times higher oscillation frequencies make achieving the ground state possible (as has been demonstrated), or explore magnetic levitation for its incredibly low noise (no ground state cooling here yet).

The next question is whether event-based imaging is suitable for ground state cooling. Currently, optically trapped nanoparticles with oscillation frequencies in excess of 100kHz have been cooled to the ground state. Commercial event-based camera *tracking* systems are limited to tracking particles at 1kHz, far too slow to enable ground-state cooling. However, the *sensors* (not the camera tracking algorithms) are capable of registering events at 1 GHz [e.g. G. Gallego *et al.*, *IEEE Trans. Pattern Anal. Mach. Intell.* **44**, 154–180 (2020)], and object tracking is well recognised as a low-noise detection technique [N. P. Bullier *et al.*, *Rev. Sci. Instrum.* **90**, 093201 (2019), Werneck *et al.*, *Rev. Sci. Instrum* **95**, 073708 (2024)].

Custom EBC tracking algorithms, working directly with the raw asynchronous data coming from the camera, have reached 30kHz [Z. Ni *et al.*, *Journal of Microscopy* 245, 236 (2012)]. In our lab, we have managed to push our event-based technology to run at 100kHz and track motion at ~ 10 kHz, we include an image shown in Figure 1 below for the referee (this is not data which we wish to include in this manuscript, and we have data for optically trapped particles for another manuscript in preparation). We believe it is feasible to push event-based tracking well above 100kHz (at which point interfacing with an FPGA control system would also be a challenge), but this is our belief and a matter of active research in our lab. At this point, since object tracking can be shot-noise limited [Werneck *et al.*, *Rev. Sci. Instrum* 95, 073708 (2024)], achieving ground state cooling should be possible.

In the manuscript, we have indicated that the sensors can register events at 1 GHz, and also the challenges following this, in the following way:

Figure 1: Event based imaging detecting motion at a harmonic of our Paul Trap micromotion frequency

“The tracking speed is limited by the proprietary tracking algorithm of the EBC used in this work, but the development of custom algorithms has enabled object tracking at 30\,kHz by working with the asynchronous data streamed from the DVS [Z. Ni *et al.*, *Journal of Microscopy* 245, 236 (2012)]. By pushing above 100\,kHz, neuromorphic sensors would be suitable for feedback cooling optically levitated

particles to the quantum ground state of motion [52], considering the shot-noise limited potential of object tracking [N. P. Bullier *et al.*, *Rev. Sci. Instrum.* 90, 093201 (2019), Werneck *et al.*, *Rev. Sci. Instrum* 95, 073708 (2024)]. This would require customizing an EBC via tracking algorithm development, and interfacing directly with control electronics.”... “Commercial neuromorphic imaging sensors, such as the one used in this study, transfer data from the sensor to the camera hardware at GHz rates [G. Gallego *et al.*, *IEEE Trans. Pattern Anal. Mach. Intell.* 44, 154–180 (2020)], and we believe there is a feasible roadmap to faster tracking (>100kHz).”

3. With the employed event-based camera, how many particles can be controlled? Is there a trade-off between detection quality and particle number?

In our experiment when considering feedback control, each particle degree-of-freedom requires an FPGA system of its own, which limited us to a maximum of 3 particles – however this is clearly not a limitation of the technique, only of the equipment in our lab.

I believe the question is more general. Our field-of-view is 640x480 pixels, with a spatial resolution of 1.24×10^4 dpi used in this experiment. The data volume due to tracking a single particle, as stated in the paper, is 100kB/s, and this increases linearly with number of particles. Therefore, we could track ~650 particles at 1kHz before we reached the same data bandwidth as a CMOS camera.

The image of a typical particle is 25 pixels square, as defined by the box that the tracking algorithm puts around each object (as can be seen in Fig 1b) and its motion is about 4 pixels within this box. The bounding box sets the separation of objects which the camera can distinguish, suggesting that without making further adjustments we could track around 500 particles. To track more, we could de-magnify: object tracking is cable to detecting sub-pixel motion [N. P. Bullier *et al.*, *Rev. Sci. Instrum.* **90**, 093201 (2019)], so being extremely conservative and saying that the object must move by 1 pixel, we could distinguish 2000 particles simultaneously (with high data volume). We are no-where near limited by the precision of the tracking algorithm, rather experimental noise as discussed above.

Of course at some point, the image would become so small that the camera would no longer consider it a moving object, but this is due to the nature of the tracking algorithm. I think at this stage we are confident in saying that the method we present would be suitable for tracking “over 100” particles, and going beyond this would require further study.

“This means that the EBC can track many hundreds of particles before the data volume becomes comparable to standard camera technology.”

“The EBC used in this study has a sensor size of 640 x 480 pixels, with each particle image occupying 25 x 25 pixels (the coloured boxes in Fig. 1(b)) and having a motional amplitude of 4 pixels. As long as the centre of the particles are separated by approximately 60 pixels, the particles can be individually tracked. Hence, without changing our imaging system we could simultaneously track of order 500 particles with this EBC. Considering the fact that object tracking allows for sub-pixel resolution [N. P. Bullier *et al.*, *Rev. Sci. Instrum.* **90**, 093201 (2019)], by changing the magnification of the imaging system this EBC would be capable of simultaneously tracking at least 2000 levitated microparticles with a correspondingly high data volume.”

4. Ultimately, 3D detection will be needed. What are the possibilities and their consequences to implement 3D detection with the current setup?

Whether or not 3D detection would be needed of course depends on the application. We get 3D detection by imaging at an angle to the principle axes of the Paul trap – as shown in the manuscript. This is a typical technique when cooling the centre-of-mass of atomic ions, for example [e.g. Itano, Bergquist, Bollinger & Windeland, *Phys. Scr.* **1995**, 106 (1995)]. This comes with a trade-off in signal-to-noise. There are a couple of options for more robust 3D detection – one could add a second camera at right-angles to the first, to image two planes of the particles, and the cameras come with suitable synchronization channels to enable parallel operation. The state-of-the-art in event-based imaging also involves depth perception (<https://www.prophesee.ai/event-camera-structured-light-evk-3d/>), though it would not be trivial to integrate this into our system, or one could directly apply the structured-light-scattering techniques involved [e.g. Huang, Zhang & Xiong, *Optics Express* **29**, 35864 (2021)].

We have added:

“We detect the motion in 3D due to a slight misalignment between the EBC and the axes of the Paul trap.”

Quality of the data: The presented data, theory and conclusions on cooling is, except of a few points highlighted later, valid and robust. Nevertheless, data supporting the performance and limits of the new detection technique is not very abundant and would be valuable to add. This could also raise the significance of this work.

Much of the work of characterizing the imaging technique was presented in our previous work [Ren *et al.*, *Appl. Phys. Letts.* **121** (2022)]. In this work we wanted to focus on neuromorphic detection as a method for control of microparticles, whereas our previous work presented neuromorphic detection as a method for imaging. However, the other referees raised a similar valid desire for more technical details, so they have been introduced throughout the manuscript, we refer you to the rest of our response.

Appropriate referencing: The referencing can partially improved. A few times more original and more suitable references are not cited in favour of more current ones, replicating the original ones.

We have added many more references throughout the manuscript now:

N. P. Bullier *et al.*, *Rev. Sci. Instrum.* **90**, 093201 (2019)
Werneck *et al.*, *Rev. Sci. Instrum* **95**, 073708 (2024)
Conangla, Rica & Quidant, *Nano Letters* **20**, 6018 (2020)
Liang *et al.*, *Fundamental Research* **3**, 57 (2023)

Event vision references:

Z. Ni *et al.*, *Journal of Microscopy* **245**, 236 (2012)
Z. Ni *et al.*, *Neural Computation* **27**, 925 (2015)

Paul trap references:

B. E. Kane, *Phys. Rev. B* **82**, 115441 (2010)
A. Kuhlicke, A. Schell, J. Zoll & O. Benson, *Appl. Phys. Lett.* **105**, 073101 (2014)
J. Millen *et al.*, *Phys. Rev. Lett.* **114**, 123602 (2015)
T. Delord, L. Nicolas, L. Schwab & G. Hetet, *New J. Phys.* **19**, 033031 (2017)

Sensor fusion references:

S. Palit, *Signal Processing* **61**, 199 (1997)
J. Z. Sasiadek, *Annual Reviews in Control* **26**, 203 (2002)
M. Nazarahari & H. Rouhani, *Information Fusion* **68**, 67 (2021)

Macroscopic quantum physics references:

P. F. Barker & MN Shneider, *Phys. Rev. A* **81**, 023826 (2010)
D. E. Chang *et al.*, *Proc. Natl. Acad. Sci. USA* **107**, 1005 (2010)
O. Romero-Isart *et al.*, *New J. Phys.* **12** 033015 (2010)

Cold damping references:

F. Tebbenjohanns *et al.*, *Physical Review Letters* **122**, 223601 (2019)
G. P. Conangla *et al.*, *Physical Review Letters* **122**, 223602 (2019)
M. Poggio *et al.*, *Phys. Rev. Lett.* **99**, 017201 (2007)

Collective mode references:

B. R. Slezak & B. D'Urso, *Applied Physics Letters* **114**, 244102 (2019)
V. Svak *et al.*, *Optica* **8**, 220 (2021)
T. W. Penny, A. Pontin & P. F. Barker, *Phys. Rev. Res.* **5**, 013070 (2012)
D. S. Bykov, L. Dania, F. Goschin & T. E. Northup, *Optica* **10**, 438 (2023)

Squeezing enhanced sensing references:

O. Cernotik, *Phys. Rev. Research* **2**, 013052 (2020)
D. Mason *et al.*, *Nature Physics* **15**, 745 (2019)

Phase Locked Loop references:

V. Jain *et al.*, *Phys. Rev. Lett.* **116**, 243601 (2016)
J. Vovrosh *et al.*, *JOSA B* **34**, 1421 (2017)

Some sentences are disconnected and causalities between statements are sometimes not super clear, e.g. Detecting and controlling multiple particles in vacuum has so far involved single particle control with sympathetic cooling [23, 24, 26], and arrays of optical traps [27]. It will be a challenge to levitate and individually control arrays of tens, or even thousands of objects [9].

We have undertaken a significant scan of the manuscript for language, and made many small changes, which we have not highlighted due to their number.

Why is it a challenge to control more particles with the referenced methods?
This comment applies more strongly to the first part of the manuscript.

Firstly, when considering CMOS camera-based imaging methods, I think we've made it clear why there is a challenge to scaling up, since the data volume becomes very large for such pixel arrays when trying to capture a large region of interest at high speed.

Considering the state-of-the-art in optical traps [27], progress in cold atomic physics has shown it's possible to construct very large arrays of optical tweezers, albeit at the use of hundreds of Watts of laser power [L. Pause *et al.*, *Optica* 11, (2024).]. When applied to optically trapped nanoparticles, this method relies on being able to spectrally separate the combined signals (as does our method when cooling), but we also note that our technique returns the motion of each particle individually and tagged with a unique ID regardless of the motion being overlapped spectrally – spectral separation is only required for cooling.

As shown in an earlier response, without any changes to our imaging system, we could independently detect the motion of >500 particles in our experiment. As long as they were spectrally separated (which is achieved by working in high-vacuum where the spectral linewidth of their motion is $\ll 1\text{Hz}$, and due to the particles' naturally varying charge), and with enough FPGA channels, we could cool all of them.

The reference to other methods facing a “challenge” has been removed, since we believe this work stands on its own as a complement to other cutting edge techniques.

“Detecting and controlling multiple particles in vacuum has so far involved either single particle control with sympathetic cooling [23,24,26] or small arrays of optical traps [27] Some applications will require the control of arrays of tens, or even thousands, of levitated particles [9].”

The authors trap charged particles in a Paul trap that are coupled via Coulomb interaction. How can the effects of sympathetic mode cooling be excluded?

We did not attempt to cool the normal modes of the coupled particle system, only the bare modes. For the 2 and 3 particle cooling data presented in the paper, the particles were trapped sufficiently far apart ($>30\mu\text{m}$) that their interactions were not evident upon spectral analysis. This is achieved since the application of a DC bias field shifts the equilibrium position of the Paul trap in a charge dependent way, and since each particle has a different charge it's trapped in a different position. Critically, we do not observe interactions in the z-direction, and this is the mode we use for multiparticle cooling.

We have clarified the role of interactions in the following way, where some of the included information is further clarified with later responses to this reviewer:

“Interactions between the particles are evident, as shown by the collective-mode peaks in Fig. 2(b). These correspond to collective centre-of-mass modes (x_C, y_C) and breathing modes ($y_B^{\{(1,2)\}}$) [27-29, 61], which we identify by resonantly exciting the particles and observing the amplified motion. We do not see coupling in the z-direction, which is the mode we use for multiparticle cooling below.”

The authors mention that they apply additional filters to their feedback signal. It is not clear to me what the goal or figure of merit of this filtering is. Could the authors add this information to the manuscript?

To cool each mode of our 3D oscillators, one must feedback a signal proportional to the velocity of each mode with gain. Even for cooling a single mode of a single particle, one requires filtering, otherwise you also feedback technical noise at frequencies away from the mode frequency, which ultimately non-resonantly heats the particle. For this reason, a bandpass filter is required for each mode to be cooled.

It isn't trivial to build tight bandpass filters in this low frequency range (in particular, we find digital filters do not perform well). Instead, we had available some analogue high-pass filters, which we used to separate the z-mode from the x- and y-modes. This was imperfect, since often the y-mode was quite close to the z-mode. The exception was Fig. 4(a), where a single low-pass filter was used to separate the x-mode from the y- and z-modes.

The best route in the future would be to use phase-locked-loops to lock-on to each particle frequency (negating the use of filters, and also allowing us to cool motional frequencies that are very close together), and this can be built into FPGA systems.

The filters were already mentioned in the Methods section. The exact limiting factor for Fig. 4(a) was already in the manuscript. For Fig. 4(b) we added: “Since only high-pass filters are used when cooling in the z-direction (see Methods), unfiltered noise from one particle is able to heat the uncooled modes of the other. At lower pressures this causes particle instability and prevents further cooling”. For Fig. 4(c) we clarified by removing the use of the phrase “cross-talk” (see below).

The authors refer to arrays of tens of particles. It is not clear to me if they mean an array of e.g. 3x3 or 10x10. Could the authors be more specific in the manuscript by giving the dimension of the hinted arrays?

This is an extremely interesting comment. Our linear Paul traps have a design which produces a field with a complex shape, which we use for multi-dimensional control amongst other things. The consequence is that our arrays are not a simple shape in 3D, and the shape can be altered a lot by changing the parameters of the trap. It is certainly possible to produce regular arrays by using different trap geometries, as is commonly done with atomic ions.

More relevant, though, is that our particles have a broad distribution of charge (and a less significant distribution of mass), meaning that they won't form a regular array. If every particle had exactly the same charge then they would form a regular array, but it would be hard to cool them since they would all have the same oscillation frequencies (though each particle could still be independently tracked). A small DC field could lift this degeneracy, at the expense of the array becoming slightly non-uniform. In high vacuum, the oscillators can have linewidths below a micro Hz ([Dania *et al.*, *Phys. Rev. Lett.* **132**, 133602 (2024)]), so only a small irregularity would be required to cool a more regular array.

The manuscript has been updated with the statement below, a more accurate Fig 1a), and a simpler to understand Fig 1b):

“Our particular Paul trap geometry, and the particles' distribution of charge, means that our particle arrays are non-uniform.”

The detection method is inherently 2D and allows for 3D detection only at the price of misalignment. Can the authors discuss the limitations if their method is used for the ultimately required 3D detection.

This was addressed above.

Can the authors discuss what the minimal distance on the detector sensor is before cross leakage between different particles occur and what this means for the maximum particle number? The signal of one particle is spread over how many pixels in this work?

Each particle image occupies about 25x25 pixels and moves about 4 pixels. When the centres get closer than 60 pixels the tracking algorithms becomes unreliable, though we note there are many parameters in the algorithm which can be optimized to try and overcome this.

This was addressed in an earlier comment.

The detection bandwidth is high enough for Paul traps with lower mechanical eigenfrequencies. What about optical traps? Can the authors state a value for the bandwidth in the manuscript?

The default settings of the camera's tracking algorithm fix the detection bandwidth to 500Hz (1kHz frame-rate). This is a proprietary setting of the particular camera we are using. We now have access to a different event-based imaging system, which we manage to run at 100kHz, detecting motion over 5kHz. We strongly believe that we can utilize this method for optically trapped particles, but it requires us to modify the hardware of the cameras and develop new tracking algorithms (this is in progress).

A more thorough discussion was included in the outlook, addressed earlier.

I believe there is not a single Paul trap reference. Could this be added?

Added, see the list of added references above.

Suggestion: All equations are in terms of ω and plots are in terms of f . In my opinion it helps preventing misunderstandings if this is adjusted to one of the two cases. It is also not quite clear if quantitative values for Γ_{fb} are now $\Gamma_{fb} = 4\text{Hz}$ or $\Gamma_{fb} = 2\pi \times 4\text{Hz}$.

All values are in rad/s. The figure axes have been updated to remove ambiguity.

A few typos can be found: Camers, Proprietary

The typos are revised in the manuscript.

In the abstract the authors claim that an array of particles increases the sensitivity. Why is that?

We thank the author for this comment, which we agree was not properly addressed. The signals from multiple sensors can be combined in a process known as *sensor fusion* to reduce measurement uncertainty, we cited a completely unsuitable reference and have updated that. There are some other specific advantages to using an array, such as increasing effective resonant detection bandwidth, and increasing the effective sensor area as cited early in the manuscript.

We have updated:

“Arrays of cooled micro-sensors will lead to enhanced signal-to-noise sensing through sensor fusion [52-54], force gradient sensing [21], and provide a larger interaction area without increasing the mass of the sensor [10].”

The references have been included in an earlier response.

I disagree with the following statement ...The performance of a mechanical sensor is inversely proportional to it's volume due to dissipation through thermal contacts and surface strain...

I claim that the acceleration or force sensitivity of a thermally limited harmonic oscillator is given by $S_{aa} \propto \sqrt{k_B T / \Gamma m}$ or $S_{FF} \propto \sqrt{k_B T / \Gamma m}$ which is related to the mass and not the volume. Maybe the authors could clarify their statement to avoid misunderstanding.

We agree this was unclear, and have clarified in the following way:

“As sensors are miniaturized, their surface-to-volume ratio will increase and the sensors become more susceptible to energy dissipation via their thermal contacts and through surface strain [3], limiting their performance.”

I claim the following statement is only true at ultrahigh vacuum . . . By levitating nano- or micro-particles in optical, electrical or magnetic fields, one creates a mechanical oscillator with remarkably low dissipation . . . Could the authors clarify this?

We clarify: “By levitating nano- or micro-particles under ultra-high vacuum conditions, using optical, electrical or magnetic fields [4.5], one creates a mechanical oscillator with remarkably low dissipation [6].”

There exist older proposals that suggested macroscopic quantum physics with levitated particles than [16].

Updated with the classic references, listed in comment above.

Are dynamics vision sensors and neuromorphic sensors the same?

We thank the reviewer for this question. A dynamic vision sensor (DVS) is a type of neuromorphic imaging device, and event based cameras (EBC) use DVS.

We have clarified: “Dynamic vision sensors (DVS) are neuromorphic sensors which mimic retinal response” and updated the rest of that paragraph to make clear that EBCs use DVS.

A word is missing ... Therefore, neuromorphic detection is highly suited to high-speed and real-time applications requiring low-power in environments with uncontrolled such ...

The word of “light levels” is added.

In my opinion is the chosen reference for active feedback via cold damping unsuitable. . . . We implement cold damping feedback to cool the motion of the levitated particles [40], . . .

References updated as requested (see above)

I suggest to add the word of L. Magrini et al too.

Added.

Could the authors add more quantitative information related to the following statement? . . . This single-device method for cooling and controlling particle arrays is readily scalable due to the low data output of neuromorphic detection. . . More precisely, how many particles can be tracked? And what is the limitation e.g. the sensor size or data flow?

This is thoroughly dealt with above.

Could the authors elaborate on the following statement? . . . Arrays of cooled microsensors will lead to enhanced sensitivity [42], . . . In my opinion, several coupled oscillators allow for the implementation of the mode localization method, but this method is not increasing the sensitivity of a thermally driven harmonic oscillator as such. Could the authors clarify their statement?

This is dealt with above, via the introduction of sensor fusion.

In Fig 1a), what is the black electrode? The particles are depicted in a circle. Shouldn't the particles form a 1D chain?

The black and blue electrodes are the endcap electrodes confining the particle predominantly along the z-axis. They are coloured differently since the control voltage (which cools the particles) is only on one of the endcap electrodes. This is an unusual configuration, where our endcap electrodes are in close proximity (<1mm) to the centre of the Paul trap – this means that the voltages applied to the endcaps produce a 3D field (which enables us to address all degrees of freedom of the particles with a single electrode). If the endcaps had a large separation then the referee is correct and one would get a horizontal line of particles. Our geometry is quite different.

The caption to the figure has been clarified, a sub-figure showing the Paul trap arrangement has been included, the figure has had a more representative arrangement of particles included, and there was a relevant change made earlier in this response.

I find the relative panel size of Fig 1 unlucky, given that panel b) is the most important panel. Some boxes in Fig1b) show "two" particles. Why is that and what is the consequence?

The resolution of the images one can export from the camera (which show the tracking boxes which we believe is important to show) is low, which is why we didn't increase the size. We believe it's important to show the actual images of the particles, rather than reconstructing the motion, so the reader gets a feel for the camera identifying objects. The subfigure (b) has been improved and clarified using a different set of data and by enhancing the colours, and Fig 1 has been split into two.

All boxes contain a single particle, the camera has two colours, indicating increasing and decreasing intensity, so the particles leave a "trail" as they move. This has been clarified in the caption.

It is not clear of which variable the PSD is calculated:

We have clarified that it's the position PSD in the manuscript.

The relative position of the particles is unclear. . . . for each particle in an array of four . . . Is this a 1x4 or 2x2 array?

It is a 1x4 array. This has been clarified by splitting Fig 1 into two figures, and explicitly including a new subfigure (Fig 2(a)) which shows the reconstructed motion of the particles and their relative position and spacing.

Could the authors elaborate more on this?. . . Interactions between the particles are evident, as shown by the unlabelled peaks in Fig. 1(c). . . I believe some peaks could also be higher harmonics of a certain mechanical eigenmode. I claim that interactions should lead to coupled oscillators with peaks at the sum and difference of the peaks, which I can not identify. If the authors have identified the origin of additional peaks, it would be helpful to add additional labels.

We thank the reviewer for this. We have significantly updated this part of the discussion, separated the PSD out into a new figure (Fig 2), labelled all of the peaks, cited more literature, and we confirm which mode is which by resonantly exciting the motion and seeing how the particles behave on the camera.

Is the Paul trap symmetric in x and y? If so, could the authors elaborate why all x-modes are equal in frequency but the y-modes are different?

No, the Paul trap is not perfectly symmetric due to manufacturing limitations. The unusual geometry of the trap (mentioned above) mean that small imperfections (e.g. the endcaps not being perfectly coaxial) lead to reasonable separation of x and y frequencies. This is also verified in simulation (SIMION). Further note that the behaviour depends on the charge to mass ratio, which varies particle-to-particle. We do not see the bare x mode, only the collective mode, as the coupling is strong in this direction. We see the bare y mode and collective y modes, and no collective mode in z. The manuscript has a much greater description of the Paul Trap geometry and the collective modes.

I disagree with this statement: . . . By reducing the position fluctuations interactions between the particles can be minimized. . . . I claim that the interactions can be stabilized due to fixed positions but this does not imply that interaction strengths are decreased.

We thank the referee for this point, and have removed this statement. By changing the interparticle spacing we can indeed minimize the interactions, so have added:

“For the data in Figs. 4(b-d) we adjust the particle spacing via the Paul trap voltages until we cannot see the type of interparticle interactions seen in Fig. 2(b).”

The authors should clarify that they mean active feedback methods.. . . Reduction of the particle energy to the ground-state of the levitating potential [14, 46] . . . Otherwise, citations using cavity cooling are missing.

These citations include cavity cooling and active feedback. Citations have been extensively updated throughout the manuscript.

References incomplete or misleading:. . . opens up a toolbox of quantum control [47] and sensitivity enhancement [6]. . . . Depending on what the authors mean with quantum control, the references should be more towards the manipulation of mechanical oscillators in the quantum regime; or if the authors only refer to cooling, then the reference list should include all 1D cooling experiments too.

Concerning [6], I claim that the reference does not cover quantum enhanced sensitivity but instead is a classical force sensing experiment. An example of quantum enhanced sensing would be the exploitation of squeezed mechanical states.

Original references on cold damping are missing:

. . . Cold damping is a feedback method . . .

E.g. [Tebbenjohanns, Felix, et al. "Cold damping of an optically levitated nanoparticle to microkelvin temperatures." Physical review letters 122.22 (2019): 223601], [Conangla, Gerard P., et al. "Optimal feedback cooling of a charged levitated nanoparticle with adaptive control." Physical review letters 122.22 (2019): 223602] and [M. Poggio, C. L. Degen, H. J. Mamin, and D. Rugar Phys. Rev. Lett. 99, 017201].

We have updated the references accordingly.

I believe in Eq. (1) it should be at least hinted that there is a noise term being neglected that heats the particle motion for large gains (a regime that is explored here too). For more details see Eq. (4) in Conangla et al., PRL 122, 223602 (2019). Especially, because the noise term appears then suddenly in Eq. (2).

For our parameters, and in particular values of feedback gain, Equation (1) reliably describes our dynamics (which we have verified), but of course the referee is correct. When we greatly increase our feedback gain we see noise squashing. The manuscript has been updated: "When Γ_{fb} is large the feedback can introduce extra noise which modifies equation (1) as discussed in \cite{Conangla2019, Melo2024}."

In my opinion there are earlier references for Equation (2), namely Conangla et al., PRL 122, 223602 (2019).

We have updated the reference.

. . . In Fig. 2(b) we show the effect of increasing feedback gain on the temperature of a single mode of a single particle, and compare the data to the model in equation (2), with only S_{nn} as a free parameter. . . . I disagree with this statement that S_{nn} is the only free parameter since the same data has previously been used to extract T_{com} and Γ_t . It would be also interesting if the fitted value for S_{nn} coincides with the noise extracted from the detection with the EBC and how this compares to standard photodiodes.

Agreed, we have modified our statement accordingly: "with S_{nn} as a free parameter and the other parameters extracted from the PSDs in Fig. 3(a)". Regarding the fitted value of S_{nn} from the variation in temperature with feedback gain, it is $(2 \pm 1) \times 10^{-14} \text{ m}^2/\text{Hz}$, which agrees with the calibrated value from the position PSDs generated from the EBC, $(1.6 \pm 0.8) \times 10^{-14} \text{ m}^2/\text{Hz}$

The authors are fitting Γ_{fb} twice. . . . This model is fit to experimental data, Fig. 2(c), with Γ_{fb} and ϕ_0 as free parameters. . . . Do the values coincide?

Yes, they agree within uncertainty.

In Fig 3, why are two particles colder than 3 particles if the particle detection is independent?

When performing multi-particle cooling, just using high-pass filters as discussed above, a significant spectrum of noise is fed-back into the system of particles, which significantly heats the collection of particles. When there are more particles, there is more feedback noise which is spectrally overlapped with the uncooled modes of the oscillators, which causes instability at a low pressures.

We have added: “The issue of imperfect filtering is more pronounced when dealing with more particles, as there are more modes of the system overlapped with the unfiltered noise.”

It is not quite clear to me what imperfect filtering shall mean. Is this due to the fact that the detection with the camera does not deliver independent detection signals, or because the filters are too steep such that unwanted phase shifts are occurring or is it because only one single electrode is used, which leads to cross talk in the electric fields?

This is extensively covered above.

Can the authors elaborate more on this statement? . . . Different particles can have different noise floors, since their coupling to voltage noise depends on the particle charge. . . . In my opinion, the noise floor is particle independent and is the noise in absence of the particle motion. While I agree that the amplitude of the motion might depend on the particle charge and therefore the SNR, I disagree that the noise floor should depend on the particle charge. Could the authors explain this in more detail?

We apologise for this abuse of terminology. This is not the noise floor of our detection, so noise floor was the wrong term to use. Rather the base-line noise we measure in our PSDs is set by the thermal energy of the particle, so as you say different particles will have different equilibrium temperatures – we explicitly say in the manuscript: “The signal-to-noise for the different particles varies due to non-uniform illumination and varying coupling to electronic noise.”

We remove that sentence from just before the conclusion, since it’s explicitly (and correctly) mentioned earlier in the manuscript.

. . . The issue of cross-talk is more pronounced when there are more modes of the system, yet we still achieve better than -7 dB of cooling . . . Can the authors specify at which point this cross-talk happens, e.g. in the detection, in the summation of the individual feedback signals, or elsewhere.

This is dealt with extensively above and the term “cross-talk” has been removed from the manuscript.

Phase lock loops have been in use for a long time. Maybe reference [50] is not the most suitable one.

References updated.

If I understand correctly, the independent detection of many particles with a single device is the main result of the paper, which is independent of the levitation technique. I find the following sentence therefore misleading since the charge is not needed for detection: . . . It’s also worth mentioning that our method is suitable for any object which can carry a charge, unlike optical trapping with intense beam detection which is limited to low-absorption dielectrics. . . .

Agreed, have changed this to “This method is independent of the levitation method, as long as there is optical illumination, meaning it is suitable for small dielectrics in optical traps, charged absorptive materials in Paul traps (such as organic material) [Conangla2020], and magnetically levitated objects. Commercial neuromorphic imaging sensors, such as the one used in this study, transfer data to the camera hardware at GHz rates [37], and we believe there is a feasible roadmap to faster tracking (>100kHz) with commercial systems.”

In Fig. 3d), which particle is which in Fig 3c)? Does the final temperature match the expectation from the SNRs in Fig. 3d)?

Fig 4(d) (updated label) shows Particles 1,3,2 moving from left to right, we have updated the caption accordingly. We do not approach the thermal noise level of these modes due to the stability issues discussed extensively above. The temperatures in Fig 4(c) are extracted from the PSDs in Fig 4(d).

Response to Reviewer 2

This work presents a method for detecting and controlling arrays of levitated particles using an event-based camera, referred to by the authors as a neuromorphic detector. However, the demonstration is limited to a small scale with relatively low frame rates.

While the authors suggest that the approach could be scaled up, this claim remains unsubstantiated.

There are two points to address here: number of particles in the experiment, and the low frame-rate.

Regarding the number of particles, we track 10 particles independently, show reconstructed trajectories of 4 (we could have done 10, but displaying 10 PSDs felt excessive), and cooled three. This has not been demonstrated in the community before – 9 particles have been trapped in an array [J. Yan *et al.*, *Photon. Res.* 11, 600-608 (2023)], but their motion hasn't been independently tracked, and no more than 2 particles have been cooled. 10 is no hard limit, it was just the largest number we were able to load into our Paul trap and image with the optics available to us. We believe it's indisputable that we could cool more particles with more electronic hardware.

Here is a more quantitative discussion. Our field-of-view is 640x480 pixels, with a spatial resolution of 1.24×10^4 dpi used in this experiment. The data volume due to tracking a single particle, as stated in the paper, is 100kB/s, and this increases linearly with number of particles. Therefore, we could track ~650 particles at 1kHz before we reached the same data bandwidth as a CMOS camera. EBCs produce asynchronous data that is extremely light-weight (each pixel can only record one of two values, and empty pixels are not recorded) which is ideally suited for object tracking [Ni *et al.*, *Neural Comput.* 27, 925 (2015)].

The image of a typical particle was 25 pixels square, as defined by the box that the tracking algorithm puts around each object (as can be seen in Fig 1b) and its motion was about 4 pixels within this box. The bounding box roughly sets the separation of objects which the camera can distinguish, suggesting that using all of the parameters of our current work we could detect around 500 particles. To detect more, we could de-magnify: object tracking is cable to detecting sub-pixel motion [N. P. Bullier *et al.*, *Rev. Sci. Instrum.* 90, 093201 (2019)], so being extremely conservative and saying that the object must move by 1 pixel, we could distinguish 2000 particles simultaneously. We are no-where near limited by the precision of the tracking algorithm, rather experimental noise as discussed above.

Of course at some point, the image would become so small that the camera would no longer consider it a moving object, but this is due to the nature of the tracking algorithm. I think at this stage we are confident in saying that the method we present would be suitable for tracking “over 100” particles, and going beyond this would require further study.

The following changes have been made to the manuscript.

“This means that the EBC can track many hundreds of particles before the data volume becomes comparable to standard camera technology.”

“The EBC used in this study has a sensor size of 640 x 480 pixels, with each particle image occupying 25 x 25 pixels (the coloured boxes in Fig. 1(b)) and having a motional amplitude of 4 pixels. As long as the centre of the particles are separated by approximately 60 pixels, the particles can be individually tracked. Hence, without changing our imaging system we could simultaneously track of order 500 particles with this EBC. Considering the fact that object tracking allows for sub-pixel resolution [N. P. Bullier *et al.*, *Rev. Sci. Instrum.* **90**, 093201 (2019)], by changing the magnification of the imaging system this EBC would be capable of simultaneously tracking at least 2000 levitated microparticles.”

Figure 2 Event based imaging detecting motion at a harmonic of our Paul Trap micromotion frequency

Werneck *et al.*, *Rev. Sci. Instrum* **95**, 073708 (2024)]. In our lab, we have managed to push our event-based technology to run at 100kHz and track motion at ~ 5 kHz, we include an image shown in Figure 2 above for the referee (this is not data which we wish to include in this manuscript, and we have data for optically trapped particles for another manuscript in preparation). Working directly with the asynchronous data-stream from a DVS has enabled researchers to track at 30kHz [Ni *et al.*, *Journal of Microscopy* **245**, 236 (2012)]. We believe it is feasible to push event-based tracking well above 100kHz (at which point interfacing with an FPGA control system would also be a challenge), but this is our belief and a matter of active research in our lab. At this point, since object tracking can be shot-noise limited [Werneck *et al.*, *Rev. Sci. Instrum* **95**, 073708 (2024)], achieving ground state cooling should be possible.

In the manuscript, we have indicated that the sensors can register events at 1 GHz, and also the challenges following this, in the following way:

“By pushing above 100kHz, neuromorphic sensors would be suitable for feedback cooling of optically levitated particles to the quantum ground state of motion [46], considering the shot-

Regarding the speed of our system. We are currently limited by the proprietary tracking algorithm that is bundled with the camera, and cannot be bypassed. However, the sensors (not the camera tracking algorithms) are capable of registering events at 1 GHz [e.g. G. Gallego *et al.*, *IEEE Trans. Pattern Anal. Mach. Intell.* **44**, 154–180 (2020)], and object tracking is well recognised as a low-noise detection technique [N. P. Bullier *et al.*, *Rev. Sci. Instrum.* **90**, 093201 (2019),

noise limited potential of object tracking [N. P. Bullier *et al.*, *Rev. Sci. Instrum.* **90**, 093201 (2019), Werneck *et al.*, *Rev. Sci. Instrum* **95**, 073708 (2024)]. This would require customizing an EBC via tracking algorithm development, and interfacing directly with control electronics.”... “Commercial neuromorphic imaging sensors, such as the one used in this study, transfer data from the sensor to the camera hardware at GHz rates [G. Gallego *et al.*, *IEEE Trans. Pattern Anal. Mach. Intell.* **44**, 154–180 (2020)], and we believe there is a feasible roadmap to faster tracking (>100kHz) with commercial systems upon developing our own tracking algorithms [Ni *et al.*, *Journal of Microscopy* **245**, 236 (2012)].”

For us, scaling up only required more FPGA output channels, which is not a hard limit. This is also discussed in the manuscript.

From a neuromorphic technology perspective, the report lacks crucial details and benchmarks.

We thank the reviewer for this comment, as it was an issue we found it challenging to balance. In previous work, we benchmarked the performance of event-based imaging for tracking a single levitated particle [Ren *et al.* *Appl. Phys. Letts.* **121**, 113506 (2022)], and we did not want to repeat too much of this. In addition, we did not want to simply benchmark a company’s tracking algorithm, for us it was important to demonstrate what was *generally* possible with this technology.

Event-based imaging is well known to offer a high-dynamic range, since instead of recording images, it registers events [e.g. G. Gallego *et al.*, *IEEE Trans. Pattern Anal. Mach. Intell.* **44**, 154–180 (2022)]. Our particular camera can offer a SNR (defined as the range between the low- and high-light cut-off) of up to 39 dB, and a dynamic range up to 120dB. This depends sensitively on a host of parameters, which we have to optimize for our application, and are camera dependent.

In the current experiment, our SNR is up to 30dB, and in previous work, where we optimized our system to detect a single particle, we achieved an SNR over 35dB [Ren *et al.* *Appl. Phys. Letts.* **121**, 113506 (2022)]. Here, by SNR, we mean the detection of the resonant motion of the particle above the base-level noise, but this noise is not due to the camera, it is due to electronic noise on the electrodes which are suspending the particle. There is no obvious “base level noise” of the camera, since it is either tracking an object, or it is not (it’s not simply related to dark count). The resolution with which we can track the object depends on the magnification of our system (which is lower than our previous work so that we can track multiple particles), and the parameters of the tracking algorithm, which is proprietary to the camera and unfortunately a “black box” in this case. In future work we intend to work with an FPGA based event-camera system to write our own tracking algorithm, which would allow us to benchmark its performance.

However, I would note that work by others have shown that camera based imaging with object tracking can reach the shot-noise limit [Werneck *et al.*, *Rev. Sci. Instrum* **95**, 073708 (2024)], and there is nothing about event-based imaging which would suggest this shouldn’t be possible.

We have added the following to the manuscript, along with the discussion in response to your previous question:

“We have previously demonstrated single-particle object tracking with an SNR above 35 dB, and for a more detailed analysis of EBC performance in the context of levitated microparticles see [44].”

“Object tracking has the potential to track levitated microparticles at the shot-noise limit [Werneck *et al.*, *Rev. Sci. Instrum* **95**, 073708 (2024)]”

There is an overemphasis on the potential advantages, which are not demonstrated. If an event-based detector is combined with a GPU or any other digital ASIC, the question arises: what fundamentally differentiates this from conventional methods?

We believe the reviewer is asking what differentiates event-based imaging from traditional imaging with intensive image processing? Apart from the incredible power requirements of high-speed tracking with conventional technology (and we do list low power consumption as a benefit in the manuscript), we believe it is clear that one could not simultaneously track hundreds of objects over hundreds of microns with sub-micron resolution and real-time read-out with a traditional system. While we only demonstrate tracking of 10 objects, as argued above we could track many hundreds before we hit the data volume of a traditional camera using its *full field of view*. An EBC is fundamentally different from CMOS technology, since each pixel can only take 2 values (representing an increase / decrease in intensity across a threshold) – there are no zero pixels, there is no 12 bit integer representing intensity. This dramatically reduces data volume, and also is ideally suited to object tracking.

We already demonstrate the use of an EBC for multiparticle cooling, and no camera-based method has achieved this before.

Additionally, the specific hardware and neuromorphic components employed are not sufficiently detailed. For example, while an FPGA is used alongside the detector, the overall energy consumption of the system must be thoroughly evaluated before labelling it as a low-energy platform.

We write “Due to the low-power consumption of neuromorphic detectors our presented methods are ideal for integration into chip-scale technology [43].” In this sentence, we are trying to say that neuromorphic detectors are typically low-power devices and therefore they could be ideal for integration into a hypothetical chip-scale device; we are not saying the system (FPGA + neuromorphic detector) has been optimized for low power performance in this case.

We use Prophesee PPS3MVCD sensors in our experiments, as detailed in the Methods section. From the sensor manual, we get the sensor static power consumption is 26 mW with a dynamic power consumption 3 nW/event. In experiments, the event rate per particle is

about 500 kevt/s, so the detectors have about 27.6 mW energy consumption in our experiments (this has been added to the manuscript). Besides, the maximum readout of detectors is 50 Mevt/s, which means the maximum energy consumption is 176 mW, which is still very low compared to standard cameras (for example, a standard CMOS camera from Thorlabs uses about 1W, and a high-speed iX Cameras i-SPEED 230 with 2500 fps uses 17 W). This is despite low power consumption not being a focus of this current work.

Furthermore, the time complexity of the entire computational process is not computed and unclear where it stands.

We thank the reviewer for this interesting point. We measured the latency of the entire data pipeline, by exposing a levitated particle to an impulse at $t=0$ and then monitoring the response in the data streamed from the last step in our pipeline (after the FPGA). Below, we present the result averaged over 40 realisations. The response is within approximately 10ms, as indicated by the red line:

Lets see if this makes sense. Our data-pipeline is sub-optimal, since data is transferred via a PC to the FPGA (as discussed in Methods) – in the future we aim to use an FPGA wired directly onto the camera hardware. However, in our current system:

- Latency from the changes in light intensity at each pixel to event output: typical 200 us from the detector manual.
- Data Transfer to Computer: The camera uses USB 3.0 at a maximum data-rate of 4 GBit/s – this will never be the limiting case because it's much larger than the data transfer at maximum event-rate. USB 3.0 has a latency of about 30 microseconds.
- The EBC software then tracks the objects – this step is very hard to evaluate, since the process is proprietary.
- Python code processes the tracking data for it to be sent to the FPGA, this is simple and can be considered negligible.
- The data is transferred via Ethernet and a network switch to our collection of FPGAs, which has a latency of about 300 microseconds.

- Data Processing on FPGA: Each FPGA systems computes velocity, applies gain and a phase shift, and outputs a feedback signal. Since our FPGA runs at 124 Msps, it is reasonable to say the latency is negligible compared to the data transfer delays.

When controlling harmonic oscillators in vacuum, a time-lag of several oscillation periods has negligible effect when cooling (e.g. Debiossac, Grass, Alonso, Lutz & Kiesel, *Nature Communications* 11 1360, (2020)), so a delay of 10 milliseconds has little consequence for a 100Hz oscillator.

We again stress, that this is a very inefficient data pipeline, used for a proof-of-principle. The optimum solution is for on-board processing of the data on the camera itself, which is entirely feasible with newer, more complex models, where the limit will be the sensor latency (200us) and communication between the camera hardware and a multi-channel DAC.

The Methods section has been updated accordingly.

It is also unclear what specific neuromorphic hardware or algorithms are applied, and why they would provide a superior result.

We are using camera Prophesee EVK1 -Gen3.1 VGA with neuromorphic sensor: Prophesee PPS3MVCD, 640×480 pixels and a proprietary generic tracking algorithm (GTA) provided by the company, which is illustrated in the Methods part of our manuscript. We use this camera/system for its off-the-shelf usability. This paper is not intended as a study in which particular hardware would be optimal, just a proof of principle that such hardware is useful.

The sensor has 120 dB dynamic range and detection based on object tracking has been shown to allow sub-pixel resolution and to be shot-noise limited (see updated references in the manuscript). We have a direct comparison of this neuromorphic detection to the detection of a quadrant photodiode (QPD) to see that the two methods have a similar SNR at the motion frequency, which is conducted in our previous work [Y. Ren *et al.*, *Appl. Phys. Lett.* 121, 111101 (2022)].

How does the resolution of this measurement depend on the hardware accuracy?

We are unsure exactly what measurement the reviewer refers to, but there are many parameters in the hardware and software which must be optimized to get the best possible resolution. Detection based on object tracking has been shown to allow sub-pixel resolution, and to be shot-noise limited, which would apply for our method with optimized sensor parameter settings (higher speed, lower background activity, higher contrast sensitivity threshold, etc.). We feel the place for best improvement is the tracking algorithm, and we are now working with a system where one can write one's own tracking algorithm. Until that time, we are limited to the proprietary algorithm bundled with the hardware.

The discussion around these topics has been strengthened in the manuscript.

Without hardware details, these are difficult to gauge. The absence of detailing and benchmarking makes it difficult to determine whether this approach offers any improvement relative to any alternative technologies.

We believe we have dealt with this above, and the demonstration of multiple object tracking with low data volume and real-time control is clearly an advance and novel addition to our field.

Reviewer 3

We thank the reviewer for their thoughtful review, and hope that they enjoy reading our significantly updated manuscript, we found your questions pertinent, and hope we addressed them to your satisfaction.

[1] For me, one of the major concerns is that it is not clear to me what is new here. If the neuromorphic detection is clearly brand-new, the cooling scheme corresponds to the well-established cold-damping approach that is here multiplexed in frequency. The neuromorphic signals are filtered in frequency and the cold damping is applied as usual on the remaining signal. How is that different from the cooling of multiple particles performed for instance in ref 27? Can the authors stress exactly what is and what is not new?

We thank the referee for this reasonable comment, and believe we have managed to strengthen the manuscript significantly in this regard. We have removed any claim that our method is “better” than e.g. ref 27, whilst highlighting the novelty and different areas of applicability.

The clear advantages are as follows: using event-based imaging produces a very low data volume, which makes it possible to achieve real time control of large arrays of particles over a wide field of view. We now include a discussion in the manuscript, whereby we calculate that without changing anything it would still be possible to track 500 particles simultaneously, with a roadmap for going further. The largest number of levitated particles in the literature that have been *imaged* is 9, and the largest number that have been tracked is 2. We track 10, and this is only limited by our particular design of Paul trap and illumination system.

Compared to optically trapped and measured particles, each particle is *independently* tracked, with a unique identification in the data-stream, meaning that even with hundreds of particles, you would know which is doing what. Cooling is not necessarily required e.g. for sensing, and we’ve updated our discussion on multi-particle sensing.

In comparison to the optically levitated array, which relies on generating an array of tightly focussed optical tweezers for readout, our method is incredibly general. It only requires that objects are illuminated (and the dynamic range of the EBC is very high, >120dB). This means our method works for absorbing particles in Paul traps, or magnetically levitated particles in cryogenic environments as well as optically levitated particles.

But again, we believe our method stands on its own as an important and novel contribution – a general method for tracking arrays of levitated microparticles, cutting-edge multi-particle control and a novel demonstration of real-time control with an event-based camera. This kind of high-speed real-time feedback control is novel in neuromorphic sensing.

The manuscript has been extensively updated to reflect this, see sections highlighted in blue in the manuscript.

[2] My second major concern regards the limitations of the technique and the claim that it can cool arrays of particles. Unless I missed something, if the neuromorphic detection can track large sets of particles with any mechanical frequency within its bandwidth, the cooling scheme can only address degrees of freedom that are well separated in frequency: one cools frequencies that very weakly overlap spectrally. Thus, the coupling between particles is very small. This can be observed in Fig 3d, in which the three cooled modes barely overlap. This is also stressed by the authors when they explain that cross talk between the modes degrades cooling and renders the particles unstable.

For me this inability to cool strongly coupled modes enters in contradiction with the claim of cooling “arrays of microparticles”. Indeed, arrays need their elements to talk to one another. In the introduction, the authors cite many papers in which arrays are made of mechanical resonators with nearby resonances (e.g., 21 or 42 to name a few). Moreover, all the different quantum applications like entanglement (ref 17) that are listed, rely on strong cross talk between the particles. Don’t get me wrong here: I am not questioning the quality of the work; I am stressing that the claim of cooling arrays of microparticles should be adapted.

Firstly, we address the issue of simultaneously cooling many particles. It has been demonstrated that particles levitated in ultra-high vacuum have linewidths significantly below micro-Hz. In this way, we are confident that it would always be possible to individually cool particles, whose frequencies would be different either through varying charge-to-mass ratio in a Paul trap, application of a bias voltage to particles charged in any way, or by tuning individual optical trap powers in an optical-tweezer array. Our work was greatly limited by the electronics available to us, but standard phase-locked loops (PLLs) would certainly be able to pick out individual narrow-linewidth oscillations. Cooling the modes of individual particles in an array will cool collective modes via sympathetic cooling (e.g. TW Penny, A Pontin, and PF Barker. *Physical Review Research* 5 013070 (2023), Dmitry S Bykov *et al.*, *Optica* 10, 438 (2023)).

We removed the phrase “cross-talk” from the manuscript, as this was poor terminology. Our cooling is limited since we only use simple *high-pass* filters (the performance of our digital filters in this frequency range was poor, so we had to use analogue filters). This means that a

large amount of un-filtered noise is fed-back into the system, heating un-cooled modes. We have clarified this in the manuscript. PLLs would mitigate this almost entirely.

In this work we have cooled particles *in an array*, you are correct that we have not cooled particle arrays, though this is only because of the limit in our lab on number of FPGA channels, as just discussed it would be possible to fully cool the array. This has been clarified and the title of the manuscript has been updated.

[3] Along the same line, the authors try to demonstrate that some degree of interaction exist between their particles (which it must, of course). For that, they claim that the “non-labelled” peaks in Fig. 1(c) are a signature of it. Here, I have some doubts as clearly some peaks correspond to harmonics of the fundamental resonances (e.g. x on the bottom panel). To me the presence of extra peaks does not stand for a proof of particles’ interactions as it can simply be related to non-harmonical trapping within the confinement of each particle (that leads to the formation of any combination between f_x , f_y and f_z). In short: of course the particles talk to one another but I am not sure that these extra peaks are a signature of it.

We thank the reviewer for highlighting this unclear aspect of our manuscript. **We have now labelled every peak in our spectra**, and even separated the sub-figure into a new figure for clarity. We have two methods of confirming the modes we observe. One is that we resonantly drive the Paul trap using a sinusoidal voltage at the frequency each peak in the spectrum individually. We can then look on the camera to see the response of the particles. Secondly, we can use a Cross-Spectral Density (CSD) analysis, which picks out only the correlated peaks from the spectra.

By these means, we are confident that we observe a 4-particle centre-of-mass mode in both the x - and y - directions, and two y -direction breathing modes consisting of 3 particles each. The manuscript has been updated to reflect this.

We have spent a fair amount of time understanding the behaviour of just two particles and their interactions. By controlling the interparticle spacing (which is straightforward with a DC field since each particle has a different charge-to-mass ratio), we can “switch” the interactions on-and-off.

We can say with absolute confidence that the particles are interacting. We hope that the reviewer agrees that we do not need to add extra data to the manuscript – cooling multiple particles is involved enough without also describing the complex interactions between many charged objects. We hope to do more work in the future on multi-particle interactions in large arrays. The manuscript has been significantly updated to reflect the above discussion.

[4] Even though I like the neuromorphic approach, I find the comparison with other techniques a bit unfair. At first, the authors state themselves that it suffers from nearly millisecond latency, which I suspect is one of the reason why the authors used ultra-low mechanical resonances (below 100 Hz). Thus, the usage of neuromorphic detection for 100 KHz resonators and above (which form the core of the research activity in optical levitation) is far from obvious to me.

More importantly, the authors claims:

” Tracking of a single particle using the entire field-of-view of the EBC used $\sim 100 \text{ kB s}^{-1}$ as compared to $64,800 \text{ kB s}^{-1}$ using a standard CMOS camera with the same sensor size. “I find this comparison a bit unfair as the field of view of a camera can be as well split in Regions Of Interest (ROIs), which makes the data required to monitor the motion of one or several particles totally manageable (as it is common practice in other fields of physics).

Regarding the first point, we agree and have now softened our comparison to other methods, as discussed above. However, we do believe there is a clear path to object tracking above 100kHz. The referee correctly points out that our latency would become the issue for higher frequency oscillators, which is of course true. However, we believe it is possible to interface the camera sensor directly with an on-board FPGA (such devices exist, we in fact have one but it is not trivial to programme) which would hugely reduce the latency, and we also note that several periods of delay in a feedback loop does not significantly harm the cooling of a low-dissipation oscillator [Debiossacet al., *Nat. Commun.* 11, 1360 (2020)].

Commercial event-based camera systems are limited to tracking particles at 1kHz, far too slow to enable ground-state cooling. However, the sensors (not the camera tracking algorithms) are capable of registering events at 1 GHz [e.g. G. Gallego *et al.*, *IEEE Trans. Pattern Anal. Mach. Intell.* 44, 154–180 (2020)], and object tracking is well recognised as a

Figure 3 Event based imaging detecting motion at a harmonic of our Paul Trap micromotion frequency

low-noise detection technique [N. P. Bullier *et al.*, *Rev. Sci. Instrum.* 90, 093201 (2019), Werneck et al., *Rev. Sci. Instrum.* 95, 073708 (2024)]. In our lab, we have managed to push our event-based technology to run at 100kHz and track motion at ~ 10 kHz, we include an image for the referee in the Figure 3 below (this is not data which we wish to include in this manuscript, and we have data for optically trapped particles for another manuscript in preparation). We believe it is feasible to push event-based

tracking well above 100kHz (at which point interfacing with an FPGA control system would also be a challenge), but this is our belief and a matter of active research in our lab. At this point, since object tracking can be shot-noise limited [Werneck *et al.*, *Rev. Sci. Instrum* **95**, 073708 (2024)], achieving ground state cooling should be possible.

In the manuscript, we have indicated that the sensors can register events at 1 GHz, and also the challenges following this, in the following ways:

“By pushing above 100 kHz, neuromorphic sensors would be suitable for feedback cooling optically levitated particles to the quantum ground state of motion [52], considering the shot-noise limited potential of object tracking [69, 72]. This would require customizing an EBC via tracking algorithm development, and interfacing directly with control electronics.”

“Commercial neuromorphic imaging sensors, such as the one used in this study, transfer data from the sensor to the camera hardware at GHz rates [41], and we believe there is a feasible roadmap to faster tracking (>100kHz) with commercial systems upon developing bespoke tracking algorithms [45].”

Regarding the ROI point of view, the EBC always uses its full field-of-view then object tracking. When reducing the ROI of a CMOS camera of course all of the objects you track must fit within that ROI, and you record every pixel in the ROI, regardless of whether it contains useful information or not. As you increase the ROI you quadratically increase the data volume, whereas tracking more particles on an EBC only linearly increases the data volume, from a low threshold as well. We agree that for tracking a single object, a CMOS with restricted ROI and the use of a GPU for real-time tracking is doable, and we cite work doing just that. But tracking objects dispersed over 100s of microns which retaining high spatial resolution would be a great challenge. In our previous work on using an EBC to track a single object, we could track motion over 100s micrometers which retaining 30nm/sqrt(Hz) position sensitivity.

As noted, we have softened our comparisons, and now also explicitly included the last point we made into the discussion. We also note that the power consumption of the EBC is far lower than a CMOS (approximately 23mW per detected particle compared to >1W for a standard CMOS and >10W for a high-speed camera) – a more explicit discussion of this has also been added to the manuscript.

[5] Also, I find the “easily-scalable” selling point a bit too much. One degree of freedom requires one FPGA and multiplying FGPA’s comes with a great cost in terms of complexity. No solution is perfect and if the authors want to put forward the scalability aspect, I would recommend that they perform a fair comparison with other techniques like the one described in ref 27.

One degree of freedom requires *one FPGA output*, not an entire FPGA, that is just what we were limited to in our lab. It is not particularly expensive to buy systems with many more channels (we now have a 64 channel FPGA). It may also be possible to sum all of the tracking

data together within the FPGA, achieving multiparticle control with a single channel. This is for real-time control – no FPGAs are required if you want to make a record of the behaviour of large arrays of particles. However, we totally agree that we were not making a reasonable comparison to other techniques, and we hope that the reviewer prefers our updated manuscript.

[6] In the introduction, it would be appreciated to complete the citation of experiments having reach the ground state (ref 14, 15) by the work of F. Marin: <https://doi.org/10.1103/PhysRevResearch.4.033051>

Also, alongside ref 27, the authors should mention the many-body cooling approach proposed by S. Rotter: <https://doi.org/10.1103/PhysRevLett.130.083203>

References updated.

[7] At last, on the neuromorphic approach, can the authors explain if the data transfer rate increase with larger mechanical frequencies (kHz instead of Hz)? Will the data bandwidth be affected? Also, can the neuromorphic approach detect the rotation of the particles? What is the optimal “misalignment” one should impose to get the best detection of translation along x, y and z?

We thank the reviewer for these interesting questions! I feel this is for discussion rather than alterations to the manuscript. For a fixed frame rate (called an accumulation time with event-based imaging) the data volume is fixed regardless of the frequency, it's just like sampling at a fixed rate. Of course, if the accumulation time is decreased, then data is transferred more rapidly from the camera, and the data volume increases. As mentioned above, we have managed to now track above 5 kHz, which actually involved a frame-rate of 100 kHz... we just didn't have a higher frequency signal to detect! As you can imagine, we are aiming to test this on optically levitated particles. A slightly more thorough description of opportunities for higher speeds is in the manuscript.

Regarding the rotation of particles, this is a very interesting question. We believe the answer is yes, but not with enough confidence to include it in the manuscript. The camera produces a tracking box around an object, and this can be set to dynamically change in size as the object moves, in a way that can be read-out. We believe that the rotation of a non-spherical object would cause a scattering pattern that could be recorded in this way. Perhaps more reliably, the raw data from the camera certainly contains this information, so once one can write one's own tracking algorithm, it should be possible.

Regarding optimal “misalignment”, we are not too sure, we get a good enough signal to cool all modes as it is. For true 3D control I would imagine a second camera should be used, although event cameras have been used with structured light to achieve 3D imaging, but implementing this in a system of levitated particles may be challenging.

Review response II: Neuromorphic detection and cooling of microparticles in arrays, Y. Ren et al.

The authors sincerely thank the referees for their valuable comments, which enabled us to make significant changes to improve the manuscript. We are grateful for the referees' efforts in preparing their comprehensive reports.

We have made extensive changes to the manuscript. This has involved bringing-in two new authors who have assisted with simulations and data analysis. We have updated our figures and clarified our imaging system geometry, we have completely overhauled the theoretical analysis of cooling, broadened and clarified our discussions of future potential, and added extensive new Supplementary Materials.

In this response letter, we individually reply to reviewer's comments in details. We colour the reviewer's comments in red, our replies in black, and our changes to the manuscript in blue, excepting some typographical or language-clarity points. To increase the readability of the manuscript and concise writing style, we also improve our expressions and labelled the changes as blue in the manuscript.

Reviewer #1 (Remarks to the Author):

The authors Y. Ren et al. of the manuscript Neuromorphic detection and cooling of microparticle arrays addressed all of my questions and answered most of them sufficiently. I agree with the authors that limitations and possibilities are hard to quantify, but I also believe that this is required to make this work a high-impact contribution to the field and to underline the discussed prospects of high frequency detection and cooling to low phonon states, despite the setup being unable to do so.

I stand with my view that this approach is promising and innovative but unfortunately I also believe that more revision is required before the manuscript could merit publication in Nature Communication. As you will see my comments concern mainly the cooling via cold damping and not the detection method itself. Below the authors will find general and specific comments/suggestions, that I hope the authors will find helpful.

General comments:

As a matter of personal liking, I still believe that the readability of the manuscript could be improved and a more concise writing style could be applied.

We thank the reviewer for recognizing the approach being promising and innovative, and we appreciate the effort taken to raise valuable questions that has helped us significantly improve the work. We have made revisions according to the reviewer's latest comments. We hope these revisions are sufficient to merit publication, but we are also open to any further comments, if needed, to make this work better. We have extensively proof-read the manuscript for readability.

Comment 1:

The following statement references only ground-state cooling with a cavity.

... and the demonstration of cooling to the ground state of an optical potential [15–17] opens. . .

As generally as the sentence is phrased, I find the list of references incomplete and, given that the authors are applying measurement-based feedback, also misleading. I would suggest the authors either cite also experiments for ground state cooling with measurement based feedback, or concentrate solely on measurement based feedback and phrase the sentence accordingly.

Thank you for pointing this out. We've added the following citations of ground state cooling with measurement-based feedback:

F. Tebbenjohanns, *et al.*, *Nature* 595.7867, 378-382 (2021).

L. Magrini, *et al.*, *Nature* 595.7867, 373-377 (2021).

M. Kamba, *et al.*, *Optics Express* 30.15, 26716-26727 (2022).

Also, we add one citation with cavity cooling to ground state:

A. Ranfagni, *et al.*, *Phys. Rev. Res.* 4.3, 033051 (2022).

The text was originally written:

“The control of levitated particles allows the exploration of a wide range of fundamental science [4, 5], and the demonstration of cooling to the ground state of an optical potential [15–17] opens the door to macroscopic quantum physics [18–21].”

Now it reads:

“The control of levitated particles allows the exploration of a wide range of fundamental science [4, 5], and the demonstration of cooling to the ground state of an optical potential [15–20] opens the door to macroscopic quantum physics [21–24].”

Comment 2:

. . . a truly scalable method . . .

Unfortunately, I do not understand what “truly scalable” means in comparison to scalable. Could the authors please clarify this to me?

With the “truly scalable” expression, we wanted to emphasize that the complexity of our method is only minimally increased as one adds more particles, and that it is a route to *detecting* the motion of a very large number of particles (certainly >100) with no added experimental effort. Currently the number of particles we cool is predominantly limited by the output channels of our FPGA system. We agree that “truly scalable” has no other meaning, and sounds like a criticism of other techniques (which is not our intention) and we now just use “scalable” in our text.

The updated sentence reads:

“We present a scalable method for arbitrary multiparticle tracking and control by implementing real-time feedback to cool the motion of three objects simultaneously...”

“We use neuromorphic detection in a scalable method for the control of arbitrary particle arrays across a wide field-of-view.”

Comment 3:

Concerning Fig 1a: The smaller inset of the Paul trap is identical to the larger illustration in the same figure. Instead of showing the same structure twice, I

would appreciate if the 3D aspect of the electrode layout were displayed more clearly. It is not clear to me that black and blue electrodes are end-cap electrodes.

We have replaced the smaller inset with a 3D aspect of the electrodes. To make the two endcap electrodes clear, we've labelled the blue electrode with "control electrode" in Fig. 1(a) and mentioned "black and blue endcap electrodes" in figure caption. The construction is expanded upon in the methods section.

Comment 4:

Concerning Fig.2a: Why are the particles placed on a diagonal line and not the central line (along z)?

Firstly, we make a clarification that there are two coordinate systems in our experiment, as shown in Fig. 1 (a)&(b). We add a caption to Fig. 1(b) as follows:

"(b) A schematic of the linear Paul trap, including the coordinate axes $\{x, y, z\}$ for the levitated particles, in contrast to the imaging coordinates $\{y', z'\}$."

Due to the proximity of the endcap electrodes to each other along the z- axis, the shape in which repelling particles arrange themselves in the x-y plane has a complex shape away from the centre, as roughly illustrated by the arrangement of the particles in Fig. 1(a). This is why we can address all three degrees-of-freedom of a particle with a single endcap electrode.

The endcap electrodes only have a diameter of $300\mu\text{m}$. Because of assembly imperfections, the endcaps are not perfectly coaxial, shown in Figure 1, and this means the particle form a diagonal line even close to the trap centre. We see particles are distributed mainly along camera y' -axis (Paul trap x-y plane) with a slight tilt along the z' -axis. We confirm this by loading many tens of particles into the trap and observing their arrangement in Fig. 1(c).

Figure 1: Real image of the experimental linear Paul trap system with a levitated particle in the trap. Due to of assembly imperfections of the two endcap electrodes, trapped particles are aligned diagonally in the camera $y' - z'$ plane.

Furthermore, our Paul trap has the voltages on the DC trapping electrodes set to maximize the stability of single-particle trapping, which involves having a small DC offset on one of the

trapping electrodes (now explained in the Methods). The offset shifts the array of particles in the x -direction. This is the reason that, in the case of the 4 particles experiment, particle 2 is the closest to the trap centre.

We have updated the manuscript in the following way:

“Our particular Paul trap geometry (Fig. 1(b)), and the particles’ distribution of charge, means that our particle arrays are non-uniform.”

“Particles 2 and 4 are closest to and furthest from the Paul trap centre.” in the caption to Fig. 2. “Of the other pair of the four electrodes, the lower one has a constant voltage of 3V applied to minimize single-particle micromotion” and “with a slight misalignment along the z' –axis which causes particles to be trapped along a diagonal in the $y' – z'$ plane” have been added to the Methods section.

Comment 5:

In my understanding, the particle furthest away from the Paul trap center should experience the strongest noise and strongest micromotion. In Fig 2a, the particles’ amplitudes suggest the opposite? Could the authors elaborate on this?

Several factors determine the amplitude of motion (temperature). In Fig. 2(a), Particle 2 is the closest to the trap centre and particle 4 is the furthest, and hence has the highest centre-of-mass temperature along the z - axis. However, the coupling to the noise depends on the particle’s charge, and Particle 1 has the highest charge (seen in the higher centre-of-mass frequencies), and this leads to enhanced motional amplitude. In Fig. 2(b), the four particles have different noise floors due to non-uniform illumination via a single laser beam.

We have added the following to the manuscript: “Particles 2 and 4 are the closest and furthest particle to trap centre respectively.”, and it’s also expanded upon in your next comment.

Comment 6:

Which particle is thermalized to 900K?

The particle in Fig. 3 (a) & (b) (without feedback cooling) is thermalized to 900K, and in Fig. 3(c) and (d) are thermalized to 1500K and 400K respectively. This is why we tend to express temperatures as ratios. The trapped particles are not always in equilibrium at the same temperature due to their varying charges and spatial locations. We have clarified this in the manuscript:

“Due to voltage noise from the amplifiers driving our Paul trap, the equilibrium temperature of our particles without cooling ranges from approximately $T_0 = 400$ -1500 K, depending on their charge and spatial location in the trap, hence we express temperatures as a ratio.”

Comment 7:

Fig 2a suggests that the particles thermalize to different temperatures since they display different amplitudes or can this be attributed to different charges?

The reviewer is correct that the particles in Fig.2(a) are thermalised to different temperatures. Indeed, the temperature of the particles varies because of both the varying charge and the fact the noise amplitude isn't uniform in space (see other work from our group which explores this arXiv:2501.03677). For a given noise amplitude, the higher the charge of the particle, the higher the equilibrium temperature it reaches (because it feels a stronger force). In addition, in this experiment, the illumination isn't uniform. A better illumination gives a better signal-to-noise (SNR, cf. Table 2 in comment 9, and in the Supplementary Material). Particle 1 has the highest charge and worst illumination, hence its temperature is the highest, and we have the worst SNR for this particle.

These points have been addressed in response to your comments above, and the additional Supplementary Materials.

Comment 8:

Have the authors measured charge to mass ratios of the particles?

We regularly measure the charge to mass ratio of single particles in the trap centre and find they typically have a positive charge ranging from $2 \times 10^3 e$ to $2 \times 10^4 e$. The mass is $(1.3 \pm 0.2) \times 10^{-13} \text{kg}$ (manufacturers data). The size and charge of the particles are provided in the manuscript (including the methods), though we don't explicitly state the charge-to-mass ratio. According to simulations based on the ion-optics software SIMION, we can characterize the geometry of our Paul trap, and then estimate the charge-to-mass ratios from the measured particle oscillation frequencies. We outline the process in detail in our manuscript <https://doi.org/10.1063/5.0106111>.

For the referee's interest, here are the charge to mass ratios of the four particles in Fig. 2.

Table 1: charge to mass ratios of the four levitated particles

	Particle 1	Particle 2	Particle 3	Particle 4
charge to mass ratio/(C/kg)	$(4.7 \pm 0.3) \times 10^{-3}$	$(4.3 \pm 0.1) \times 10^{-3}$	$(3.8 \pm 0.3) \times 10^{-3}$	$(3.1 \pm 0.4) \times 10^{-3}$

Comment 9:

Concerning Fig.2b: Why do you see only a center of mass mode (CoM) along x ? Why do you observe coupled modes but also an uncoupled mode in y ? I would expect only one of the two scenarios (coupled or uncoupled)? The particles are closer together in z than in y (see Fig 2a). Why is there no coupling along z ? Could the authors please add numbers to the y -axis? Could they authors state the current SNR?

Based on this comment, we have re-defined the coordinate system in our work, and run further simulations to give a more convincing answer, so we greatly thank the reviewer for pushing us on this point.

As is explained in our reply to *Comment 4*, there are two coordinate systems in our experiment, as shown in the picture below. One is the Paul trap system labelled $\{x, y, z\}$, and the other one is camera system labelled as $\{y', z'\}$. The Paul trap coordinates are aligned on the diagonal line of the trapping electrodes, and set the axes of oscillation of the particles, and the image of the particle is projected onto the camera axes as shown in the figure, with $z' = z$.

Hence, the particle's motion along x and y is projected onto the camera's y' -axis and particle's motion along z is parallel to camera z' -axis. So the previously labelled $x_c, y_c, y_B^{(1)}$ and $y_B^{(2)}$ in Fig. 2(b) are actually all along the x -axis, and the four labelled y and z modes are along the y - and z - directions.

From the cross-spectral density (for further details, see Supplementary Materials Section S2), we find four coupled modes along x and no coupling between the modes along the y -axis and the z -axis. Four modes are expected, since n coupled 1D harmonic oscillators have n normal modes. In our experiment, the coupling is caused by the electrostatic forces between the particles. Thus, the coupled modes are along the x -axis, and much weaker in the y, z -axes (we don't measure any) as motion along these two axes are perpendicular to the particles' alignment to each other.

Figure 2: Illustration of our Paul trap coordinate system $\{x, y, z\}$ system and our camera system $\{y', z'\}$. The x and y axes has a 45° projection onto the camera y' -axis and the z' - & z -axes are the same.

In the text we originally write as:

“ x_c is the centre-of-mass mode of all four particles in the x -direction, y_c is the equivalent mode in the y -direction, and $y_B^{(1,2)}$ are two breathing modes in the y -direction. We also see the individual bare modes in the y - and z -directions.”

We have now updated the text as:

“ $x_c^{(1)}$, $x_c^{(2)}$, $x_c^{(3)}$ and $x_c^{(4)}$ are four collective modes of all the four particles in the x -direction. We also see the individual bare modes in the y - and z -directions. (For information on identifying modes see Supplementary Materials S2 & S3.)”

We have also updated this part to Supplementary Materials Section S1 and introduce the two coordinate systems into the text as follows:

“The linear Paul trap defines the coordinate system $\{x, y, z\}$ for the levitated particles, Fig. 1(b). The image on the EBC has a coordinate system $\{y', z'\}$, where the $\{x, y\}$ axes are projected at 45° onto the y' axis, and the z' - and z -axes are parallel, for details see Supplementary Materials S1. This projection allows us to detect all three axes of motion of the levitated particles.”

Numbers are now added to y -axis in Fig. 2(b). The SNRs of the four particles have been added to the Supplementary Materials, and we reproduce them here:

Table 2: Detected SNRs of the four levitated particles

	Particle 1	Particle 2	Particle 3	Particle 4
f_z SNR(dB)	3.9 ± 0.8	6.2 ± 1.1	7.4 ± 1.1	27.9 ± 1.1
f_y SNR(dB)	2.9 ± 1.1	6.5 ± 1.1	6.1 ± 1.1	19.8 ± 1.1
$x_c^{(1)}$ SNR(dB)	4.5 ± 0.8	10.9 ± 0.8	13.3 ± 0.8	15.3 ± 0.8
$x_c^{(2)}$ SNR(dB)	6.0 ± 1.1	5.0 ± 1.3	4.7 ± 1.0	12.1 ± 1.1
$x_c^{(3)}$ SNR(dB)	3.2 ± 0.8	5.1 ± 1.4	0	15.2 ± 0.9
$x_c^{(4)}$ SNR(dB)	0	3.2 ± 1.1	3.7 ± 1.0	15.6 ± 0.9

Comment 10:

I find the following statement misleading.

. . . this method can cause damping or amplification of the particle motion without adding additional noise, . . .

I claim that active feedback always introduces noise, unless one assumes the theoretical case of an ideal detector with zero detector noise, which does not mirror the current situation.

The referee is of course correct. In experiment, active feedback control will always introduce noise to the controlled system. With the sentence, we are explaining that cold damping is a dissipative process without fluctuation (as commonly recognized in the literature, see e.g. [J. Gieseler and J. Millen, *Entropy* 20.5, 326 (2018).]) – in contrast if we immersed our particles

in liquid helium they would thermalize to a cold temperature, but through stochastic collisions with the helium molecules which would cause fluctuation. See also our answer to Comment 13 below. For this reason, we immediately follow with “In reality, input and output noise of the feedback electronics still limits cooling.”

Comment 11:

Concerning Fig3a: Maybe the authors want to remove some data traces to enhance read-ability.

Following the advice, we have kept three data traces with labelled Γ_{fb} of 0.0 Hz, 2.5 Hz and 6.2 Hz respectively and removed other data traces in Fig. 3(a).

Comment 12:

Concerning Fig 3b)-c): If I understood the main text correctly, the two data sets applying the same experimental procedure are fitted with different theoretical equations (Eq. 2 to Fig 3b and Eq. 3 to Fig.3c), where one is considering noise and the other one is not. In my opinion both data sets should be fitted to the same equation, taking the noise into account, meaning Eq. 2 with a phase dependent $\Gamma_{fb}(\theta)$.

We have refreshed the entire theoretical analysis around cooling. We've added the noise terms into equation (1) & (3), and replaced Γ_{fb} with $\Gamma_{fb}(\phi)$ (a function of the feedback phase ϕ) in Eqs.1 & 2 as well. The manuscript is updated and all data in the manuscript is fit with the updated equations.

Comment 13:

Concerning Fig 3d: Which equation is used to fit the data?

For pressures higher than $\sim 10^{-6}$ mbar, the dominant contribution to the stochastic forces is due to air molecules and the damping rate caused by gas molecules is proportional to pressure [J. Millen, *et al.*, *Thermodynamics in the Quantum Regime: Fundamental Aspects and New Directions*. Springer, 853-885 (2019)]. Thus, we use equation (2) as the theoretical model in the manuscript to fit the data derived from measured PSD area in Fig. 3(d).

Comment 14:

Concerning Fig 4d: Maybe the authors can consider removing data points to enhance read-ability?

We have reduced the contrast of the data points to increase readability Fig. 4(d).

Comment 15:

How are the errorbars derived?

We use a χ^2 fitting routine to compare equation 1 to our data, which enables us to calculate standard errors on each of the fitting parameters. When these parameters are used to compare theoretical models to our results (e.g. equation 2), we propagate the uncertainties in the fitting parameters using standard Gaussian error propagation, enabling us to calculate a range of theoretical expectation, as expressed by the shaded regions in Fig. 3 & 4. We have clarified in the caption of Fig. 3: “with the pink shaded region representing the uncertainty in the parameters extracted by fitting the PSDs as in (a)”. The data-points are derived from the area of the PSD (now clarified in the manuscript: “...and compare the temperature measured using the area of the PSD to...”). We take 15 repeats at each set of experimental parameters, and this allows us to calculate a mean area and the standard deviation, which we use as an experimental error bar.

Comment 16:

Concerning Eq.1: I claim T_{com} should be replaced with T_0 the bath temperature. I agree that in the absence of cooling $T_{com} = T_0$ but this is not the case for $\Gamma_{fb} \neq 0$. The following sentence also needs adaptation.

Thank you for this, we agree. We’ve updated equation (1) considering added noise via feedback, in which T_{COM} is replaced by T_0 as well.

Comment 17:

I suggest to modify Eq. 3 and replace it with a clarification in the text that the cold damping feedback is phase dependent $\Gamma_{fb}(\theta) = \Gamma_{fb} \cos(\phi + \phi_0)$. If the authors want to keep Eq. 3, then I suggest that they introduce the noise term in Eq 3, such that Eq 2 and 3 are compatible and describe the same situation.

Again, we appreciate this feedback. Equation (3) is modified with “and data processing, and ϕ is a con-trollable phase delay generated by our feedback electronics.” in the text and the noise term is also introduced in equation (3).

Comment 18:

I would like to raise again that Eq.1 is neglecting the noise contribution. The T_{com} (Eq 2) is directly related to the area of PSD of the displacement noise S_{xx} in Eq 1, therefore I claim that it is not consistent to consider the noise contribution in Eq. 2 but not in Eq.1.

Furthermore, I find that the added sentence is not resolving the issue . . . When Γ_{fb} is large the feedback can introduce extra noise which modifies equation (1) as discussed in [50, 56].. . .

We completely agree, have updated the analysis here and equation (1) is updated with considering noise via feedback.

Comment 19:

I suggest to add reference 40

. . . extra noise which modifies equation (1) as discussed in [50, 56]. . . .

Reference added. To make the text more readable, we have updated the sentence. The text was originally written as: “When Γ_{fb} is large the feedback can introduce extra noise which modifies equation (1) as discussed in [50, 56].”

Now it reads:

“When Γ_{fb} is large the feedback can introduce extra noise and leads to heating, as discussed in [57, 61, 72].”

Comment 20:

Could the authors highlight the units of S_{nn} ? From the text, it is not clear to me if this is voltage noise, displacement noise or other.

The unit of S_{nn} is the same as the unit of $S_{zz}(\omega)$, explicitly $m^2(\text{rad s}^{-1})^{-1}$. The origin is the noise in the feedback electronics, which is fed-back onto the particle (hence why it is multiplied by the feedback gain). This is reflected in the discussion after equation (1): “The terms S_{nn} and S_{dd} are the feedback circuit noise and detector noise respectively, and are modeled as having a constant spectral density”.

Comment 21:

If Γ_{fb} is fitted twice and agree within errors, this could be stated.

From the PSD fitting with equation (1), we get the $\Gamma_{fb} = 0.70 \pm 0.09$ Hz, and via the fitting with equation (3) we obtain $\Gamma_{fb} = 0.82 \pm 0.05$ Hz. Therefore, we've added "The value of Γ_{fb} obtained by fitting the data in Fig. 3(c) with equation (3) agrees with the value obtained at the same feedback gain by fitting the data in Fig. 3(a) with equation (1), ($\Gamma_{fb}/(2\pi) = (0.82 \pm 0.05)$ Hz, (0.70 ± 0.09) Hz respectively)." in the text.

Comment 22:

I find the following statement misleading

... The fitted value of ϕ_0 is 5° , noting that one- or two- periods of phase delay do not significantly effect the cooling for an underdamped oscillator as in our system ...

First of all, $\phi_0 = 5^\circ$ is much less than an oscillation period. Second, only the value of $\phi_0 + \phi$ is important (assuming an underdamped oscillator). And the statement is only true if the mismatch coincides with full oscillation periods. If ϕ_0 equals a fraction of a period while ϕ is constant then cooling can turn into trapping, heating, or anti trapping.

From fitting equation (3), it is hard to determine whether $\phi_0 + \phi$ is 5° , $360^\circ + 5^\circ$ or $720^\circ + 5^\circ$ We characterized the latency of the complete data pipeline by applying an impulse to a levitated particle and monitoring the corresponding response in the data stream recorded at the final stage of the pipeline (after the FPGA). The results, averaged over 40 realizations, are presented below (this is now expanded upon in the Supplementary Materials). The red line indicates that the system responds within approximately 10 milliseconds.

Figure 3: Response measurement of a microparticle to an impulse over 40 realizations.

In our work, particles are levitated with ~ 100 Hz motion oscillation frequencies, for which 10ms corresponds to one period of phase delay so we have updated the manuscript with $\phi_0 = 370^\circ \pm 5^\circ$. We agree that only the value of $\phi_0 + \phi$ is important, and we believe the manuscript in we have now clarified the discussion around equation (3):

“where ϕ_0 is the uncontrollable phase delay caused by electronics and data processing, and ϕ is a controllable phase delay generated by our feedback electronics. Equation (3) is fit to experimental data in Fig. 3(c), with Γ_{fb} and ϕ_0 as free parameters. The fitted value of ϕ_0 is $370^\circ \pm 5^\circ$, noting that one period of phase delay does not significantly effect the cooling for an underdamped oscillator [74]”.

Comment 23:

... We are sensitive to all three degrees-of-freedom due to the angle our imaging system makes to the principal axes of the trap. The geometry of our Paul trap allows the control of all degrees-of-freedom with a single electrode. ...

Why do you only cool two modes then? Given your outlook to employ the method to larger arrays and lower vacuum levels, would cooling in 3D not strengthen your claim?

Following the referee’s advice, we have now conducted 3D cooling of one particle, shown below. However, since receiving the report, our neuromorphic detector broke, and we had to replace it with a different model which exhibited much worse noise characteristics, hence we have not changed the data in the manuscript (otherwise it would include data taken from different models of detector). We are happy to share this data with the referee.

In this figure both the x and z modes reach their noise floor. Due to the geometry of our Paul trap, the feedback force is weaker along the y -axis, hence the cooling along the y -axis does not reach its noise floor. The minimum CoM temperatures along x -, y - and z -axis are decreased by over 15 dB.

Figure 4: Simultaneous cooling of a single particle along x , y and z -axis.

Comment 24:

I must admit that I am not fully convinced by these statements

. . . unfiltered noise from one particle is able to heat the uncooled modes of the other.

At lower pressures this causes particle instability and prevents further cooling. . . .

. . . The issue of imperfect filtering is more pronounced when dealing with more particles, as there are more modes of the system overlapped with the unfiltered noise. . . .

The authors are using only one electrode pair to apply the electrical force onto the particles.

Even without additional noise, the electrical signal contains the signal of all particles/modes meaning that one drives all particles at the same off-resonantly and out of phase time, leading to heating. Why are the authors excluding this effect?

We thank the referee for requesting clarification of this point. If one were to introduce white voltage noise into the experiment, for example by applying a noisy voltage to one electrode, then all of the modes of all of the particles would heat up, since the spectrum of the white noise is flat and overlaps with all of the particle modes.

Since the motion of the different particles is spectrally separated, ideally when performing feedback you would create a signal which only contains on-resonance information for each particle. For example, if you have a 100Hz oscillator, and a 60Hz oscillator, if you take the 100Hz signal and tightly filter it with a bandwidth of 10Hz, the resulting system should not heat the 60Hz oscillator as the feedback signal is far from its response function. However, due to limitations in the lab, we are only able to high- or low-pass signals when dealing with multiple modes. Hence, if we were to only low-pass filter the 100Hz signal, the noise in the resulting feedback signal would spectrally overlap with the 60Hz signal, heating the mode. Additionally, of course filtering is not perfect, and so even with a simple bandpass filter, particularly at these low frequencies, you will get overlap between the feedback signals and the modes of the other particles.

The solution, as mentioned in the manuscript, is to use an extremely narrow filter, for example that produced by a Phase Locked Loop (PLL). We tried this, and include data for the referee below, showing that it's possible to produce a very narrow bandwidth filter significantly suppressing noise even a few Hz from the resonant frequency (we note this method could be optimized):

Figure 5: Example PSD of a 230Hz oscillator filtered using a band-pass filter (blue) and a phase-locked loop (PLL - orange), showing that a PLL can produce a very narrow-bandwidth filter.

We do not have enough devices capable of performing a PLL in the lab, so do not include this in the manuscript.

We hope this answers the referee’s questions. We have not changed the manuscript as we believe this is clear, though are open to suggestions.

Comment 25:

Can the authors specify the type of interaction? (I guess Coulomb interaction)
 . . . we cannot see the type of interparticle interactions . . .

Yes, Coulomb interaction. We have revised it to be “until the Coulomb interaction is weak enough such that there are no coupled modes, see Supplementary Materials S2.” in the text, and also introduce the term “Coulomb interactions” in discussion of Fig1(b) and Fig. 2(b) as “The charged particles form a stable array due to the Coulomb repulsion between them.” and “Coulomb interactions between the particles are evident [31–33, 67], as shown by the collective modes in Fig. 2(b),” .

Comment 26:

. . . Collective modes in particle arrays can also be cooled via sympathetic cooling [28, 29]. . . .

I was under the impression that only center of mass modes (in phase modes) can be cooled and the breathing modes (out-of-phase modes) cannot (assuming the same electric field con-figuration)?

Both centre of mass modes (in phase modes) and the breathing modes (out-of-phase modes) can be cooled via sympathetic cooling. In the cited paper [D. S. Bykov *et al.*, *Optica* 10(4),

438–442 (2023)], the authors used sympathetic cooling to cool six collective oscillation modes (in-phase mode and out-of-phase mode along three axes) of a particle simultaneously.

Comment 27:

. . . Paul traps are stable at low pressures [73], where the motional frequencies of levitated particles have sub-Hz linewidths [74], meaning that the naturally varying charge-to-mass ratio of charged nano- and micro-particles will enable single-particle control and cooling even for large arrays. . . .

In my opinion, this statement is very strong. Given that the authors use the stated platform but do not demonstrate control of only a few particles at very low pressures because of difficulties. Also the problem of uncooled coupled modes seem to be neglected.

It has been demonstrated that particles levitated in ultra-high vacuum have linewidths significantly below micro-Hz. In this way, we are confident that it would always be possible to individually cool particles, whose frequencies would be spectrally separated either through varying charge-to-mass ratio in a Paul trap, *application of a bias voltage* to particles charged in any way, or if using the neuromorphic technique in a different platform, by tuning individual optical trap powers in an optical-tweezer array. Our cooling is limited to not-very-low pressure since we only use simple analogue high-pass filters. Therefore, a lot of unfiltered noise is fed-back into the Paul trap system and heat un-cooled modes. We believe standard phase-locked loops (PLLs) would certainly be able to pick out individual narrow-linewidth oscillations and push the control of particle arrays to a high vacuum, and showed the referee an example above.

We use the flexibility of the Paul trap to turn coupling between the particles on and off. In our two-particle cooling, the separation of the two particles is $170 \pm 10 \mu\text{m}$. The motion frequencies of the two particles along z -axis are $93.88 \pm 0.07 \text{ Hz}$ and $60.55 \pm 0.07 \text{ Hz}$. By computing their cross-spectral density (for further details, see Supplementary Materials Section S2), we find that the two modes are not coupled.

In three-particle cooling work, the separations of two adjacent particles are $320 \pm 20 \mu\text{m}$ and $250 \pm 15 \mu\text{m}$ respectively. The motion frequencies of the three particles along the z -axis are $54.79 \pm 0.01 \text{ Hz}$, $87.64 \pm 0.01 \text{ Hz}$ and $61.72 \pm 0.03 \text{ Hz}$. If we compute their cross-spectral density, we can see the three modes along z -axis are also not coupled.

Therefore, we can say that in our cooling scheme of multiple particles, all modes along z modes are uncoupled. It is true that levitated particles will have coupled modes when we decrease the particle separation within particle arrays (see Supplementary figure 2 in Supplementary Materials Section S1) as we demonstrate in Fig. 2. Coupled modes can also be cooled via cold damping [e.g. V. Liška, *et al.*, *Optica* 10.9, 1203–1209 (2023).] or sympathetic cooling [e.g. TW Penny, A Pontin, and PF Barker. *Physical Review Research* 5 013070 (2023), Dmitry S Bykov *et al.*, *Optica* 10, 438 (2023)].

We believe our work is a proof-of-principle for cooling very large arrays, and that the challenge of implementing a bank of PLLs for each mode is only a technical hurdle, not a conceptual bottleneck.

We have made various changes to the manuscript based on your feedback, including updating the Discussion with “Paul traps are stable at low pressures, where the motional frequencies of levitated particles have sub-Hz linewidths. The naturally varying charge-to-mass ratio of charged microparticles, along with application of electric field gradients, will enable the spectral separation of motional modes, making possible single-particle control and cooling even for large arrays”, while softening our claims to be able to control ~ 100 particles (though detect a lot more, note that for detection the particles do not need to be spectrally separated). We have added Supplementary Materials S2 to show that we can turn interactions between the particles on-and-off.

Reviewer #2 (Remarks to the Author):

I have reviewed the responses to the referees, and my primary observation is that the authors need a stronger understanding of general computing terminology and, more specifically, neuromorphic computing.

It appears the authors are utilizing hardware without fully grasping how it operates. There are numerous neuromorphic hardware platforms and algorithms aimed at improving noise performance and reducing computational complexity. The response regarding computational complexity is particularly misaligned and highlights a lack of perspective on how computation is actually performed.

In short, this implementation doesn't really bring anything noteworthy to the table from the perspective of neuromorphic hardware. I recommend that the authors familiarize themselves with the following references to gain a better understanding of computation:

1. <https://www.nature.com/articles/s41586-018-0180-5>
2. <https://pubs.aip.org/aip/apr/article/7/3/031301/997525/Analog-architectures-for-neural-network>
3. <https://www.nature.com/articles/s41586-024-07902-2>
4. <https://www.nature.com/articles/s41586-019-1677-2>

We thank the reviewer, who is clearly an expert in neuromorphic computing, in suggesting interesting literature for us to read. When embarking in a new area of research, it is always a challenge to identify the most relevant corpus of work, and to align the use of technical knowledge between fields.

We are a little confused as to the suggestion that we are “utilizing [neuromorphic computing] hardware without fully grasping how it operates”. We do not use, and make no claims to be using, neuromorphic computing hardware. We use a neuromorphic vision sensor, whose output is processed by standard computing hardware. The advantage of using the neuromorphic sensor is that the neuromorphic mode of operation produces a very light data-stream containing the information about the many objects within its field of view. Object identification and tracking is done by the hardware using entirely conventional computing techniques.

We are not claiming to have made an advance in any neuromorphic technology. Rather, we claim to have made an advance in control, with application in sensing, by using the neuromorphic sensor to detect many moving objects (which is novel in our field), and to enable feedback control of many objects simultaneously, which is novel generally.

However, there is a lot of relevant information in the references provided, and we sought out even more technical papers on event-based vision based on the feedback from the referee. We have made the following additions based on this report:

- We have added reference 3 and 4 above to our discussion of low power consumption. Before it is written as:
“and high dynamic range (> 120 dB) detection with minimal data output at low power consumption [40].”
Now it reads:
“and high dynamic range (> 120 dB) detection with minimal data output at low power consumption [38, 41, 48].”
- We added all of the references to the motivation statement about what neuromorphic means. Before we write as follows:
“Neuromorphic sensors are highly efficient detectors which mimic neurobiological information gathering [33].”
Now it is revised as:
“Neuromorphic sensors are highly efficient detectors which mimic neurobiological information gathering [37–41].”
- Reference 4 was particularly interesting and useful, we thank the referee for this. We have added it to the discussion about energy consumption, where we define an event, where we mention data-redundancy, we have added “interfacing the sensor directly with FPGA or neuromorphic processing electronics [38, 80]”. We have added the reference [80] J. Furmonas, *et al.*, *Sensors* 22.3, 1201 (2022). to also support these same statements.
- We have also tried to better use technical information from literature we had already cited. Using Z. Ni, *et al.*, *J. Microsc.* 245.3, 236–244 (2012)., we have expanded our discussion of data volume: “The development of custom algorithms has enabled object tracking at 30 kHz by working with the asynchronous data streamed from a DVS using only 4MB of RAM on a standard 2.9 GHz Dual Core CPU [52]”. We have supported discussions around object tracking with Z. Ni, *et al.*, *Neural Comput.* 27.4, 925–953 (2015)., which is written in the text as “detecting changes across a threshold on each pixel in an array asynchronously to produce a stream of events [44] ideally suited for object tracking [45].”.

Reviewer #3 (Remarks to the Author):

On behalf of all of the authors, we would like to sincerely apologise to this reviewer who felt that their concerns were not dealt with properly. Without dwelling on the matter, the lead author of this manuscript was not available during the last round of reviews, leading to a sub-standard response to the referee.

We greatly appreciate the opportunity to respond. We have put significant effort into this new response and version of the manuscript, including brining in additional authors, carrying out new simulations and data analysis. We have added new Supplementary Materials backing up many of our claims, reworked several of the figures, and also include some additional data just for this referee. We believe the manuscript is greatly improved, thanks in no small part to this reviewer.

Comment 1:

“Less-scientific” comments:

- First, the “quality” of the reply was of a very low standard. The authors did not list the sentences that have been changed (apart from one). They seemed to expect that I would scroll the whole text for each of their reply in order to guess which sentence was changed and for what reason.

We appreciate the time the reviewer has spent on this work and apologise for the inconvenience we’ve caused by having to re-review this manuscript. We have ensured that all changes are now highlighted in the manuscript, and for your convenience have compiled a full list of changes at the end of this response.

Comment 2:

For instance, in my comment [3] of the former review, I raised a reasonable concern regarding the nature of the different peaks in the spectrum. This point is not minor at all. As a reply, the authors claimed that they implemented two techniques to confirm the nature of the peaks but without providing any data or figures to back it up. I quote:

“We can say with absolute confidence that the particles are interacting. We hope that the reviewer agrees that we do not need to add extra data to the manuscript [...]”

Maybe adding extra material to the manuscript is much (even though I doubt it). Yet, they should have provided those data to the supplements or the

Methods, and more importantly, I should have access to them. They cannot ask me to believe something that I have not seen.

The referee raises a valid concern, and we now provide a much more thorough analysis, including adding supplementary materials. We carried out new simulations, and have reappraised the multiparticle interaction part of our manuscript. Four harmonic oscillators interacting in 1D should display 4 collective modes, which is indeed what we observe. The four particles we display interact strongly along the x direction, and we label the 4 collective modes as $x_c^{(1-4)}$. For each particle we then see two additional non-interacting modes, y, z . Although the logic of this is clear when looking at the spectra, we furthermore use two experimental methods to verify this assumption.

First, we can identify the modes by resonantly driving them with an electrical signal, and observing their motion on a camera. Here we show two co-levitated micro-particles with $570\mu\text{m}$ separation. We track their motion using our neuromorphic sensor, and calculate the PSDs, as shown in figure. 6. From this we extract six modes in total which are 18.2 Hz, 22.2 Hz, 35.6 Hz, 59.9 Hz, 88.3 Hz and 97.0 Hz.

Figure 6: PSDs derived from the detected motion of two co-levitated particles separated by $570\mu\text{m}$.

We introduce a sinusoidal voltage onto one of our electrodes with a fixed frequency, and then monitor the response of the microparticles using a CMOS camera, shown in figure 7. This allows us to identify that all 6 frequencies correspond to a distinct centre-of-mass mode of one or other of the particles, noting the projection of the Paul trap principal axes onto our camera (discussed in more detail in Supplementary Materials S1).

Figure 7: Camera views after a pair of levitated particles is driven by a sinusoidal voltage at the given frequencies.

Secondly, to identify collective modes, a Cross-Spectral Density (CSD) analysis is employed to pick out only the correlated spectral components. As shown in figure 8, none of the six centre-of-mass motion modes exhibit strong coupling between the two particles.

Figure 8: Cross-spectral density along the z' and y' axes of two co-levitated particles separated by $570\mu\text{m}$. No modes are observed to be coupled.

We now co-levitate another pair of particles, and use the Paul trap voltages to reduce their separation to $150\mu\text{m}$. We again observe 6 resonant frequencies in their PSD, at 33.7 Hz and 58.1 Hz along the x -axis, 85.2 Hz and 150.9 Hz along the y -axis, and 89.4 Hz and 98.2 Hz along the z -axis. This time, a CSD analysis indicates that the x -axis modes at 33.7 Hz and 58.1 Hz, and the z -axis modes at 89.4 Hz and 98.2 Hz, are correlated, as shown in figure. 9, while the y -axis modes at 85.2 Hz and 150.9 Hz are uncoupled.

Figure 9: Cross-spectral density along the z' and y' axes of two co-levitated particles separated by $150\mu\text{m}$. Modes along x -axis and z -axis are observed to be coupled, modes along y -axis are uncoupled.

We presented this analysis for the referee for the case of 2 particles, due to the ease of listing all of the modes. We straightforwardly extend this analysis to the 4 particles presented in the manuscript. Following the advice we have put related data and analysis results in the Supplementary Materials Sections S2 & S3.

Comment 3:

- As a last example, I asked in my comment [7] some questions that were meant to improve the paper and addressing them in the manuscript would have been a strong plus. Here, I would like to quote the first sentence of their answer to my comment:

“We thank the reviewer for these interesting questions! I feel this is for discussion rather than alterations to the manuscript. “

After this introduction sentence, they wrote a few lines and did not make any change to the text. I spent a great deal of time reading and reviewing this paper for free. Thus, a small talk is not what I was looking for.

Here we firstly list the reviewer’s comment [7] in the first round:

“[7] At last, on the neuromorphic approach, can the authors explain if the data transfer rate increase with larger mechanical frequencies (kHz instead of Hz)? Will the data bandwidth be affected? Also, can the neuromorphic approach detect the rotation of the particles? What is the optimal “misalignment” one should impose to get the best detection of translation along x , y and z ?”

We again apologise to the referee for our seemingly brusque response, which we hope to remedy in this revision. As suggested, we have updated the manuscript in the following ways:

“For a fixed frame rate the data volume is fixed regardless of the particle motion frequency. If the accumulation time is decreased, data is transferred more rapidly from the camera, and the data volume and bandwidth increase.”

Optical readout of nano- or micro-particle rotation is commonplace in the field working with levitated particles, and libration motion has been measured in electrical and magnetic traps, see e.g. [J. Delord, et al., Nature 580.7801, 56–59 (2020).], [B. Huillery, et al., Physical Review B 101, 134415 (2020).]. It is possible to design algorithms for computing the orientation of anisotropic objects when working with the data streamed directly from a neuromorphic sensor (e.g. [K. Kim, et al., Optical Trapping and Optical Micromanipulation XIII, Springer, 211–223 (2016).], [J. Apps, et al., arXiv:2505.08126 (2025).]) though it is not a functionality of the generic tracking algorithm we used. We do not have non-spherical particles to test in our Paul trap, and do not yet have the ability to work with the raw sensor data in real-time, though this is an aspiration of ours using an event-based camera with integrated FPGA hardware.

We have updated the manuscript with the above references, and in the following way in the Discussion: “This would require custom tracking algorithms and interfacing the sensor directly with FPGA or neuromorphic processing electronics [38, 80], which would also enable the read-out and control of object alignment and rotation [79,81].”

Regarding the “misalignment”, this was very unclear in the previous version of the manuscript, and we have significantly updated the manuscript in response. The secular oscillation frequency axes of the levitated particles are determined by the geometry of the Paul trap, which we label $\{x, y, z\}$. The Paul trap system is projected onto the coordinate system of our camera, now labelled $\{y', z'\}$. We illustrate these coordinate systems in figure 10.

The particle's motion along the x - and y - axes has a 45° projection onto the camera y' -axis, while its motion along the z -axis is aligned with the camera z' -axis. Therefore, we can detect 3D motion via a single camera. This is the optimal alignment for 3D detection.

Figure 10: Alignment our camera coordinate system relative to the Paul trap system. The x and y axes have a 45° projection onto the camera y' -axis and z is parallel to camera z' -axis.

We have added Supplementary Materials S1, updated the figures in the manuscript, and updated the manuscript:

“The linear Paul trap defines the coordinate system $\{x, y, z\}$ for the levitated particles, Fig. 1(b). The image on the EBC has a coordinate system $\{y', z'\}$, where the x, y – axes are projected at 45° onto the y' – axis, and the z' – and z – axes are parallel, for details see Supplementary Materials S1. This projection allows us to detect all three axes of motion of the levitated particles.”

Comment 4:

[1] The authors claim that they perform detection over the motion of multiple particles and that “cooling is not necessarily required”. The way I understand this point is that tracking is the most important input of this method. And I agree that this is impressive! Yet, in the following answers, they use cooling as a major argument (even as part of their title). Thus, I have the feeling that the role of cooling is downplayed on purpose here to please me here before being put forward further down the line.

Also, I do not have the time to scroll the text to check all the blue sections and try to guess which ones are supposed to have been adapted. Please make a detailed list.

We will do our best to clarify our thinking and motivation regarding this topic. While it is true that there are a host of applications for which cooling is not important, the simple detection of an array of levitated particles is only novel for the community of researchers who study levitated particles. For this community we do believe event based imaging is powerful, since it is a unique way of getting high-speed readout of the motion of each particle in an array, *regardless of whether their motional frequencies are identical or not.*

However, the true novelty of this work is that we use a neuromorphic sensor to generate a real-time signal which is used for real-time feedback control over multiple objects at “high speed” (high speed for the communities which typically work with event based imaging). Cooling is the bridge linking the community of people working with levitated particles, to the community of scientists who may be more generally interested in control via video imaging, where data volume is a huge challenge. We have separated the claims about tracking and control, as given below.

We updated the clear advantages of our methods as follows:

“In our system, with fixed magnification, tracking a single particle at 1 kHz using the entire field-of-view (3.75 mm^2) of the EBC uses $\sim 100 \text{ kB s}^{-1}$ as compared to $64,800 \text{ kB s}^{-1}$ using a standard CMOS camera (Thorlabs CS165MU/M) at the same frame rate and field-of-view. This means that the EBC can track many hundreds of particles before the data volume

becomes comparable to standard camera technology. By not having to restrict the region-of-interest, the EBC can track objects dispersed over several hundred micrometres whilst retaining high spatial resolution [66].”

“The EBC used in this study has a sensor size of 640×480 pixels, with each particle image occupying 25×25 pixels (the coloured boxes in Fig. 1(c)) and having a motional amplitude of 4 pixels. As long as the centre of the particles are separated by approximately 60 pixels, the particles can be individually tracked. Hence, without changing our imaging system we could simultaneously track of order 500 particles with this EBC. Considering the fact that object tracking allows for sub-pixel resolution [53, 79], by changing the magnification of the imaging system this EBC would be capable of simultaneously tracking at least 2000 levitated microparticles, with a correspondingly high data volume.”

“We believe that the particle control method presented in this work could be extended to an array of order 100 microparticles. Multichannel FPGA systems with high-quality digital filters are a common tool in research labs. Paul traps are stable at low pressures [82], where the motional frequencies of levitated particles have sub-Hz linewidths [83]. The naturally varying charge-to-mass ratio of charged microparticles, along with application of electric field gradients, will enable the spectral separation of motional modes, making possible single-particle control and cooling even for large arrays.”

“The neuromorphic detection and cooling presented here is independent of the levitation method, as long as there is optical illumination, meaning it is suitable for small dielectrics in optical traps, charged absorptive materials in Paul traps (such as organic material) [57], and magnetically levitated objects.”

Comment 5:

[2] Here, I don't agree with the different points brought forward by the authors. They claimed that levitated particles can have micro-Hz linewidth and that they can be separated spectrally to be cooled. Just as a rule of thumb, if I assume that I collect a PSD in which two “peaks” are spaced by a few micro meters, the time trace will look like an almost perfect sinus with an extremely low envelope modulation. To acquire a PSD with two peaks one would need seconds of acquisition. To fully measure a micro-Hz linewidth you need 10^6 seconds. You can infer it faster, but recent work like the one from Tracy Northup's group require way more than a few ms to do so. Thus, I have reasonable doubts that using hundreds of particles oscillating at 1 kHz and spaced by even a few Hz would work since everything must happen way faster than a single oscillation period.

The referee makes an excellent point. In principle, one could have an initialization phase, where a long measurement identified all of the frequencies of the particles in an array, and

then lock onto those for future control (since we observe the frequencies to be stable in our experiment), but we accept this is a stretch. However, it is certainly possible to cool particles separated by a few Hz, in Fig. 4(d)) of the manuscript two particles modes are simultaneously cooled while being separated by 7Hz.

There is another technique available to particles in a Paul trap though, and that's to engineer a spatially varying field, which would shift the frequencies of particles in the array by different amounts. In fact, due to the varying charge-to-mass ratios of the levitated particles, even changing the trap stiffness shifts the frequencies of the particles by varying amounts. In this way we do believe it is not such a stretch to spectrally separate modes, especially if working with more highly charged particles with similar linewidths.

We agree that it would be a massive challenge to *control* hundreds or thousands of particles. We have softened the language in the discussion accordingly:

“We believe that the particle control method presented in this work could be extended to an array of order 100 microparticles. Multichannel FPGA systems with high-quality digital filters are a common tool in research labs. Paul traps are stable at low pressures [82], where the motional frequencies of levitated particles have sub-Hz linewidths [83]. The naturally varying charge-to-mass ratio of charged microparticles, along with application of electric field gradients, will enable the spectral separation of motional modes, making possible single-particle control and cooling even for large arrays”

Comment 6:

Also, the argument that you could use a PLL to pick the individual narrow linewidth is also not convincing to me. In order to lock, a PLL needs at least five oscillations (if you are very good). If you have frequencies that are very close in frequency I really doubt that would work or that you could converge fast enough.

We thank the reviewer for challenging us to test what we claim – hopefully the response to Comment 5 goes some way to addressing this point. We have softened our claims, though left them in since we believe it is acceptable to speculate at the state-of-the-art in the discussion section of a manuscript. For the interest of the referee, we temporarily borrowed a lock-in amplifier with a PLL, and applied it to our system, you can see this below in figure 12. We see it is possible to generate a significantly better filtered signal via a PLL, making us optimistic that in future work we will be able to cool nearby modes more straightforwardly.

Figure 12: Comparison of band-pass filter and PLL to particle motion signal.

Comment 7:

At last, I don't get the nuance when changing the claim "to cool particles arrays" into "cool particles in an array". The authors have to define exactly what they mean here.

This change was in response your previous warranted comment. You suggested that "cooling particle arrays" would suggest that we cooled the collective modes which arose due to interparticle interactions. This is not what we do in this manuscript, we only cool the modes of non-interacting particles, hence we "cool particles in an array". We have looked through the manuscript thoroughly, and do not believe that there is any ambiguity on this matter, but if you have further suggestions, we are happy to implement them.

Comment 8:

[3] Here, as discussed above, the authors have to provide data of what they claim. They claim that they have implemented "two methods confirming the modes [they] observe". I am more than ready to believe them but not without data and evidence. And those data have to be incorporated in the Methods or into a Supplementary Material.

Same thing when they claim that they can "switch the interactions on-and-off". I cannot trust something that is not proved.

We completely agree with the referee. We hope that we have adequately answered this question in our reply to *Comment 2*, but have also added Supplementary Materials Sections S2 & S3 to back-up our claims.

Comment 9:

[4] The authors acknowledge that they do have a latency problem but that FPGAs will reduce it. Here, no quantification of how much is to be expected. Also, they claim that a latency in the cooling of a few cycles is harmless by citing a paper that stresses the fact that delays actually change all the thermodynamics of levitated objects [Debiossac et al.] (i.e., substantial changes in the dynamics). Thus, a clarification would be welcome here.

We thank the referee for this comment, which we answered in the previous round in response to a different reviewer. We measured the latency of the entire data pipeline, by exposing a levitated particle to an impulse and then monitoring the response in the data streamed from the last step in our pipeline (after the FPGA). Below, we present the result averaged over 40 realisations. The response is within 10ms, as indicated by the red line:

Figure 13: Response measurement of a microparticle to an impulse at Time = 0ms, averaged over 40 realisations.

Our data-pipeline is sub-optimal, since data is transferred via a PC to the FPGA– in the future we aim to use an FPGA wired directly onto the camera hardware. However, in our current system:

- Latency from the changes in light intensity at each pixel to event output: typical 200 us from the detector manual.
- Data Transfer to Computer: The camera uses USB 3.0 at a maximum data-rate of 4 GBit/s – this will never be the limiting case because it’s much larger than the data transfer at maximum event-rate. USB 3.0 has a latency of about 30 microseconds.
- The EBC software then tracks the objects – this step is very hard to evaluate, since the process is proprietary.
- Python code processes the tracking data for it to be sent to the FPGA, this is simple and can be considered negligible.
- The data is transferred via Ethernet and a network switch to our collection of FPGAs, which has a latency of about 300 microseconds.

- Data Processing on FPGA: Each FPGA systems compute velocity, applies gain and a phase shift, and outputs a feedback signal. Since our FPGA runs at 124 Msps, it is reasonable to say the latency is negligible compared to the data transfer delays.

It is important to emphasize that this data pipeline is highly inefficient and was implemented solely as a proof of principle. The optimal approach would involve on-board data processing directly on the camera, which is entirely feasible with newer and more advanced models. In such cases, the primary limitations would be the sensor latency (200 μ s) and the communication bandwidth between the camera hardware and a multi-channel DAC.

Considering that all of our oscillation frequencies are below 100 Hz, 10ms latency is just under one oscillation period. The work by Debiossac *et al.* showed that a delay of one full period in their feedback protocol made negligible difference to the efficiency of the feedback (their figure 3b), and they were working with an oscillator of similar quality factor (55 in their experiment).

This discussion has been added as a new Supplementary Materials section S4.

Comment 10:

Regarding the ROI vs EBC comparison, I find the edits to the text still biased. I would agree with a statement saying that ROI's scale quadratically with the number of particles while EBC scales linearly. Yet, saying that the data rates are 100 kB/s versus 64 000 kB/s is still misleading.

Thank you for requesting further clarity. We have now revised the related sentence, which was originally written:

“Tracking a single particle at 1 kHz using the entire field-of-view of the EBC uses $\sim 100 \text{ kB s}^{-1}$ as compared to $64,800 \text{ kB s}^{-1}$ using a standard CMOS camera with the same sensor size.”

Now it reads:

“In our system, with fixed magnification, tracking a single particle at 1 kHz using the entire field-of-view (3.75 mm^2) of the EBC uses $\sim 100 \text{ kB s}^{-1}$ as compared to $64,800 \text{ kB s}^{-1}$ using a standard CMOS camera (Thorlabs CS165MU/M) at the same frame rate and field-of-view.”

Comment 11:

[5] Regarding the scalability, as discussed in [2], I do not get how stacking FPGA will help cooling as I doubt that their internal PLL could lock with a resolution low enough.

We hope that our responses to previous comments are adequate to satisfy the referee here, including data where using a PLL produces a very narrow bandwidth signal. We also note that we have generally softened our claims about scalability, especially in comparison to other work and regarding control. In particular, we no longer use the phrasing “truly scalable”, and just use “scalable”.

Comment 12:

[7] As discussed previously, my comments were meant to improve the paper and not for a small talk between myself and the authors. Thus, I would appreciate if they were to take them into account.

Again, we would like to note for the reviewer that the manuscript is greatly improved thanks to their detailed work. We had some critical staffing issues in the months surrounding the previous response, meaning that we were somewhat rushed in providing a reply. This is no excuse, and we hope that the detailed report we have provided this time goes some way towards remedying the situation.

Full list of changes

As for the comment [1], with the referee's comment, we removed any claim that our method is "better" than e.g. ref 27, whilst highlighting the novelty and different areas of applicability.

"Detecting and controlling multiple particles in vacuum has so far involved either single particle control with sympathetic cooling [31, 32, 34] or small arrays of optical traps [35, 36]."

We also included a discussion in the manuscript, whereby we calculate that without changing anything it would still be possible to track 500 particles simultaneously, with a roadmap for going further.

"The EBC used in this study has a sensor size of 640×480 pixels, with each particle image occupying 25×25 pixels (the coloured boxes in Fig. 1(c)) and having a motional amplitude of 4 pixels. As long as the centre of the particles are separated by approximately 60 pixels, the particles can be individually tracked. Hence, without changing our imaging system we could simultaneously track of order 500 particles with this EBC."

Cooling is not necessarily required e.g. for sensing, and we've updated our discussion on multi-particle sensing.

"Arrays of cooled micro-sensors will lead to enhanced signal-to-noise sensing through sensor fusion [58–60], enable force gradient sensing [25], and provide a larger interaction area without increasing the mass of the sensor [11]."

The manuscript has been extensively updated to reflect that we believe our method stands on its own as an important and novel contribution – a general method for tracking arrays of levitated microparticles, cutting-edge multi-particle control and a novel demonstration of real-time control with an event-based camera.

"This means that the EBC can track many hundreds of particles before the data volume becomes comparable to standard camera technology. By not having to restrict the region-of-interest, the EBC can track objects dispersed over several hundred micrometres whilst retaining high spatial resolution [66]."

"There is a latency of 10 ms in this pipeline, see Supplementary Materials S4. EBCs are available with FPGA systems on the camera hardware, which will significantly reduce this latency."

As for the comment [2], we removed the phrase "cross-talk" from the manuscript, as this was poor terminology. We also clarified that PLLs would mitigate the problem in which a large

amount of un-filtered noise is fed-back into the system, heating un-cooled modes in the manuscript.

“Since only high-pass filters are used when cooling in the z-direction (see Methods), unfiltered noise from one particle is able to heat the uncooled modes of the other. At lower pressures this causes particle instability and prevents further cooling.”

“The issue of imperfect filtering is more pronounced when dealing with more particles, as there are more modes of the system overlapped with the unfiltered noise.”

“Filtering can be improved either by separating the particles modes and applying band-pass filtering, or through the use of phase-locked-loops [76, 77].”

As for the comment [3], we have added the following sentences to explain the presence of extra peaks of four levitated particles and more detailed information on Supplementary Materials S2 & S3.

“We observe interactions between particles in the array: $x_c^{(1)}$, $x_c^{(2)}$, $x_c^{(3)}$ and $x_c^{(4)}$ are four collective modes of all the four particles in the x-direction. We also see the individual bare modes in the y- and z-directions. For information on identifying modes see Supplementary Materials S2 & S3.”

As for the comment [4], regarding the first point, we agree and have now softened our comparison to other methods, here is we updated sentences.

“This means that the EBC can track many hundreds of particles before the data volume becomes comparable to standard camera technology. By not having to restrict the region-of-interest, the EBC can track objects dispersed over several hundred micrometers whilst retaining high spatial resolution [66].”

In the manuscript, we have indicated that the sensors can register events at 1 GHz, and also the challenges following this, in the following way:

“The development of custom algorithms has enabled object tracking at 30 kHz by working with the asynchronous data streamed from a DVS using only 4MB of RAM on a standard 2.9 GHz Dual Core CPU [52].”

“By pushing above 100 kHz, neuromorphic sensors would be suitable for feedback cooling optically levitated particles to the quantum ground state of motion [17], considering the shot-noise limited potential of object tracking [75, 78]. This would require customizing an EBC via tracking algorithm development, and interfacing the sensor directly with FPGA or neuromorphic processing electronics [38, 80], which would also enable the read-out and control of object alignment and rotation [79, 81].”

We also note that the power consumption of the EBC is far lower than a CMOS (less than 30mW per detected particle compared to >1W for a standard CMOS and >10W for a high-speed camera)

“When combined with the low power-consumption of neuromorphic imaging technology (less than 30 mW per tracked particle, see Supplementary Materials S5), and great progress in chip-scale particle levitation [61], integrated devices containing arrays of quantum sensors are closer to being a reality.”

As for the comment [5], we totally agree that we were not making a reasonable comparison to other techniques, and we have softened our comparison to other methods. We introduce our method as an important and novel method for tracking arrays of levitated microparticles and real-time control with an event-based camera. We hope that the reviewer prefers our updated manuscript.

As for the comment [6], we have added the references as suggested.

“The control of levitated particles allows the exploration of a wide range of fundamental science [4, 5], and the demonstration of cooling to the ground state of an optical potential [15–20] opens the door to macroscopic quantum physics [21–24].”

Reference [20] is the work of F. Marin: <https://doi.org/10.1103/PhysRevResearch.4.033051>

“Detecting and controlling multiple particles in vacuum has so far involved either single particle control with sympathetic cooling [31, 32, 34] or small arrays of optical traps [35, 36].”

Reference [36] is the work of S. Rotter: <https://doi.org/10.1103/PhysRevLett.130.083203>

As for the comment [7], a slightly more thorough description of opportunities for higher speeds is in the manuscript.

“By pushing above 100 kHz, neuromorphic sensors would be suitable for feedback cooling optically levitated particles to the quantum ground state of motion [17], considering the shot-noise limited potential of object tracking [75, 78]. This would require customizing an EBC via tracking algorithm development, and interfacing the sensor directly with FPGA or neuromorphic processing electronics [38, 80], which would also enable the read-out and control of object alignment and rotation [79, 81].”

Review response III: *Neuromorphic detection and cooling of microparticles in arrays*, Y. Ren *et al.*

The authors would like to express sincere gratitude to the referees for their efforts behind these constructive feedback and suggestions. We have revised the manuscript accordingly to the comments, and we hope these changes are clear, accurate and satisfactory to the reviewers. We feel that both reviewers believe that this work is suitable for publication in Nature Communications upon satisfactorily acting upon their queries.

In this response letter, the reviewers' comments are addressed individually. For clarity, the reviewers' comments are coloured in red, our replies in black, and our changes to the manuscript in blue. All changes in the manuscript are also coloured in blue, excepting some typographical or language-clarity points.

Reviewer #1 (Remarks to the Author):

The authors Y. Ren et al. of the manuscript Neuromorphic detection and cooling of microparticle arrays addressed all of my questions and answered most of them sufficiently. My comments are mainly suggestions, that I hope the authors will find helpful. Exceptions to this are comments 12, 14, III and IV which I believe needs addressing.

We sincerely thank the reviewer for the careful reading of our manuscript and for the constructive suggestions, and we have considered the reviewer's comments and made changes accordingly. We believe these changes have improved both the clarity and quality of the manuscript, and we are grateful for the reviewer's insights.

Comment 3:

I appreciate the authors effort. Unfortunately the end cap electrodes are covered by larger electrodes. I would like to make the not required suggestion to adapt the figure again.

Thanks for pointing it out. We have changed the view angle of the sub-figure, while still enabling us to define the coordinate axes, and now the endcap electrodes are more visible.

Comment 5:

To prevent misunderstandings of the reader, I would suggest to define the origin of the camera and particle coordinate system to be the trap center (see Fig.2a).

Following the reviewer's suggestion, we have updated Fig. 2(a), and highlighted this in the figure caption as follows:

"Particles 2 and 4 are closest to and furthest from the Paul trap centre (the centre of the coordinate axes) respectively."

Comment 8:

A unrequired suggestion is to state the charge to mass ratio related to Fig.2.

We have added the charge-to-mass ratios to the Supplementary Materials (S3) and signposted this from the text as follows:

"Each particle has a different charge-to-mass ratio (see Supplementary Materials S3) and is levitated in a different part of the confining field"

Table 1: Charge to mass ratios of the four levitated particles.

	Particle 1	Particle 2	Particle 3	Particle 4
q/m(C/kg)	$(470 \pm 30) \times 10^{-5}$	$(430 \pm 10) \times 10^{-5}$	$(380 \pm 30) \times 10^{-5}$	$(310 \pm 10) \times 10^{-5}$

Comment 9:

Thanks for illustrating this point. I found the illustration added to the answer of this comment to be more useful than Fig. 1 and Fig. 2. Consider adding it to the manuscript/replacing part of Fig.1 with it. Nevertheless, I still find the explanation of the absence of coupled modes in the y- and z-direction in the manuscript insufficient. Could the authors add a sentence highlighting the fact that the coupling is increased along x due to the motion along the array (in contrast to y and z)??

We thank the referee for this useful request for clarification.

We have updated the manuscript in the following way:

“The four particles are aligned along the x -axis. Motion of the charged particles in this direction leads to coupling between them via the Coulomb interaction [31–33, 67]. This is evident via the collective modes x_C seen in Fig.2(b). By controlling the separation between the particles, we can control the coupling strength, see Supplementary Materials S2 for further details. For the multi-particle cooling presented below, we work in a regime where the coupling between the modes is too small to measure, and perform cooling along the z -direction.”

Comment 10:

In the given reference J. Gieseler and J. Millen, Entropy 20.5, 326 (2018) they state . . . Note that this simplified picture assumes that the feedback signal is perfect and that it does not feedback any noise, which in general is not true. . . In my opinion the author’s phrasing . . . this method can cause damping or amplification of the particle motion without adding additional noise, hence the terminology “cold damping”. In reality, input and output noise of the feedback electronics still limits cooling. . . is causing unnecessary confusion without adding important information to the topic of the manuscript. The wording can cause, in contrast to assumes, suggests a real possibility which is incorrect.?

Thank you for highlighting this potential ambiguity. The “*can cause*” term refers to heating or cooling caused by the method, rather than the noise. We’ve clarified in the following way by removing the description of the word “cold” in cold damping, to avoid any confusion:

“Depending on the phase of the feedback force relative to the motion, this method damps (cools) or amplifies (heats) the oscillator. When cooling, input and output noise of the feedback electronics limits the ultimate temperature.”

Comment 12:

If I understand correctly, the detector/camera is an in-loop detector meaning that it is used to detect and at the same time to generate the feedback signal. If this is the case, I disagree with Eq. 1 where I claim S_{dd} should not be present as an independent noise term. All noise (in the feedback loop S_{nn} and in the detection S_{dd}) is noise that is being feedback and therefore heats the motion governed by the susceptibility of the oscillator (see M.Poggio PRL 99, 017201 (2007)).?

We thank the reviewer pointing out that our system is in fact an in-loop detector, as assessment which we now agree with. As a result, we have updated not only Eq. 1 but also Eq. 2 and Eq. 3 to reflect the in-loop nature of our detector. In accordance with this, we have also re-processed the data presented in fig. 3 and fig. 4 using the in-loop expressions. This causes some small changes in presented results (e.g., the minimum noise-floor temperature changes to 6.8K in figure Fig. 3(d) due to fitting with the new expression). However, it does not have a significant impact to the main conclusions of the work.

Note, as with comment 17 we have merged equations 2 and 3.

We updated the following expressions

Replace Eq. 1:
$$\frac{2k_B T_0 \Gamma_0 / m}{(\omega^2 - \omega_z^2)^2 + \Gamma_t^2 \omega^2} + \frac{\Gamma_{fb}(\phi)^2 \omega^2}{(\omega_0^2 - \omega^2)^2 + \Gamma_t^2 \omega^2} S_{nn} + S_{dd}$$

with the in-loop expression:
$$\frac{2k_B T_0 \Gamma_0 / m}{(\omega^2 - \omega_z^2)^2 + \Gamma_t^2 \omega^2} + \frac{\Gamma_0 \omega^2 + (\omega_0^2 - \omega^2)}{(\omega_0^2 - \omega^2)^2 + \Gamma_t^2 \omega^2} S_{nn}$$

Delete Eq. 2:
$$T_{COM} = T_0 \frac{\Gamma_0}{\Gamma_t} + \frac{\frac{1}{2} m \omega_z^2}{k_B} \frac{\Gamma_{fb}(\phi)^2}{\Gamma_t} S_{nn}$$

and keep the Eq.3:
$$T_{COM} = \frac{T_0 \Gamma_0}{\Gamma_0 + \Gamma_{fb} \cos(\phi + \phi_0)} + \frac{m \omega_z^2}{2 k_B} \frac{\Gamma_{fb}^2 \cos^2(\phi + \phi_0)}{\Gamma_0 + \Gamma_{fb} \cos(\phi + \phi_0)} S_{nn}$$

In the text, we also modified related descriptions about data fitting, which were originally written as:

“We cool a single microsphere to sub-Kelvin temperatures and single degrees-of-freedom of multiple particles.”

“Experimentally, the temperature of each mode T_{CoM} can be extracted from the integral of the measured PSD over the corresponding resonance peak [57].”

“In Fig. 3(b) we show the effect of increasing Γ_{fb} on the temperature of a single mode of a single particle, and compare the temperature measured using the integral of the PSD to the model in equation (2), with S_{nn} as a free parameter and the other parameters extracted from the PSDs in Fig. 3(a) using equation (1).”

Now they write as follows:

““We cool a single microsphere to a temperature of a few Kelvin and single-degrees-of-freedom of multiple particles.”

“Experimentally, the temperature of each mode T_{CoM} can be extracted from equation (2) after fitting the measured PSD over the corresponding resonance peak with equation (1) [57]. In Fig. 3(b) we show the effect of increasing Γ_{fb} on the temperature of a single mode of a single particle”

Comment 14:

Instead of increasing the readability of Fig. 4d, I can barely see the data now which is the basis of the displayed fit. Reconsider decreasing the number of data points instead of making them less visible as suggested previously.?

We have done as the referee suggests, binned our data and made the points more visible. We also noted that Fig. 3(a) has the same visibility issue and have applied the same data processing.

Comment 15:

This information should be added somewhere . . . We take 15 repeats at each set of experimental parameters, and this allows us to calculate a mean area and the standard deviation, which we use as an experimental error bar. . . .?

The following phrase has been added to the captions of figures 3 & 4:

“All experimental error-bars in the figure are derived by taking 15 repeat experiments at each set of parameters to calculate a mean and standard deviation.”

Comment 17:

Eq. 2 and 3 are the same now. Consider merging them.

That's done. We have updated the text following equation (2) to reflect the merger.

Comment 24:

I agree with the authors that they can filter the detection signals. But my concern was related to cooling, so feeding back the signal. Let's assume perfect, noise free detection of oscillators at frequencies f_1 , f_2 , and f_3 . To control the modes with a single electrode means the authors add the three detection signals and apply it to a single electrode. This means that I will drive all oscillators with all three frequencies of which only one frequency is required for cooling. Hence even in the absence of noise, the authors will heat up to a certain extent all the modes. So additionally to unfiltered noise (which is discussed by the authors), this will be an additional problem (which is not mentioned).

The reviewer is correct that there could be cross-talk between modes and between particles. In experiments, the motion frequency of particles are between 60Hz and 150 Hz, where the gas damping rate is usually below 1 Hz at the pressure $P < 10^{-2}$ mbar. In Figure 1, we conduct a simulation to observe the trend of particle's PSD frequency response when a driving signal is off the particle's resonance frequency at different pressures. Take $\Gamma_0 = 1$ Hz as an example, the frequency response drops by over 25 dB with a 10 Hz frequency offset (from the centre frequency at 100 Hz). As the gas pressure decreases, the gas damping rate Γ_0 also decreases. A frequency offset of the same magnitude will result in a smaller response in the particle's PSD. Therefore, we can say that the cross-talk between modes and between particles can be very small especially when entering high vacuum.

Figure 1: The frequency response of a driving signal off the particle's resonance frequency at different pressures. The set particle resonance frequency is 100 Hz.

Cross-talk between orthogonal modes can be minimized by better electrode design, as in *Gosling, H. et al. arXiv:2506.17172*. We do see some suggestion of minimal cross talk between parallel modes of different particles from the CSD analysis in the supplementary material S3. We note that this is an issue which becomes less prominent the colder the particles get, since the size of the feedback signal also will decrease. We have updated the manuscript in the following way:

“Cross-talk between particle modes would also limit the ultimate cooling temperature, which could be resolved by better design of the feedback electrodes [76].”

Comment 26:

I disagree with the author. In Fig. 3 of D. S. Bykov et al., *Optica* 10(4), 16 438–442 (2023) they show that they can only cool efficiently the detected particle. Collective modes can only be cooled in the weak cooling regime where the feedback damping is smaller than the coupling strength of the modes.

We agree with the referee, and have added “in the limit that the feedback damping rate is smaller than the coupling strength between the modes [31, 32].” to the manuscript.

Comment I:

Strictly speaking I disagree with the following statement. . . The particles are not coupled in the z-direction, which is the mode we use for multiparticle cooling. . . . If the particles are charged, then they are always coupled, only their coupling in this case is negligible. I suggest to change the wording.

We take the point, and updated the manuscript as follows:

“By controlling the separation between the particles, we can control the coupling strength, see Supplementary Materials S2 for further details. For the multi-particle cooling presented below, we work in a regime where the coupling between the modes is too small to measure, and perform cooling along the z-direction.”

Comment II:

The main novelty of this article is the simultaneous detection of many particles. Hence, I would suggest to state typical SNR achieved here in the context of the following sentence: . . . The signal-to-noise for the different particles varies due to non-uniform illumination and varying coupling to electronic noise.. . . since this is the main general limitation for particle cooling.?

We have added the following to the manuscript:

“The signal-to-noise for the different particles varies due to non-uniform illumination and varying coupling to electronic noise. It ranges from 3-30, with exact values given in Supplementary Materials S3.”

Comment III:

I claim Fig.3 c plots the relative temperature versus $\phi + \phi_0$ instead of ϕ . If not then, why does the relative temperature maximize at $\phi = \pi$?

The axis on the figure is correct, it just happens that our fixed phase delay is 370 degrees, so close to 2π that the temperature still maximises at close to π , within the uncertainty we've provided.

Comment IV:

... Figure 3(a) shows the PSD of a particle's motion along the z-axis as it is cooled via cold damping. The shape of the position PSD S_{zz} is given by [57]: ...?

I claim that Fig 3a is showing the detected position signal which is not identical with the real position signal, which due to detection noise is undetectable directly. I suggest to clarify that Fig 3a shows the measured detection signal. Eq. 1 on the other hand is the real position (neglecting S_{dd}). The measured data should be fitted with the power spectral density of the detected signal and the real position inferred from there. I am aware that differences will be small (possibly negligible) but wording should be corrected.

The reviewer is correct that the PSDs are based on the particle's detected position. We have updated the PSD equation in equation 1 (see comment 12). For this comment, we corrected the wording in the caption of Figure 3(a) and the text description as follows:

“Single particle PSDs of the measured motion along the z-axis” and “Figure 3(a) shows the PSD of a particle's measured motion along the z-axis”.

Comment V:

I suggest to change the statement ... a constant spectral density. . . to “a constant power spectral density”. More common would be to use the term white noise as described in Millen, J., Gieseler, J. (2019). Single particle thermodynamics with levitated nanoparticles. In *Thermodynamics in the Quantum Regime: Fundamental Aspects and New Directions* (pp. 853-885).

As suggested, we now say that they are modelled as “white noise” instead of “a constant spectral density”. The sentence is revised as follows:

“are modeled as white noise.”

Comment VI:

. . The parameters T_0 , Γ_0 and $\Gamma_{fb}(\phi)$ can be extracted from a measured PSD by fitting equation (1) . . .

Given that T_0 can vary significantly for Fig 3 and Fig 4, it would be helpful to state it somewhere. E.g.in Fig. 4c, the particles would be cooled only close to room temperature if $T_0= 1500K$, which I believe is important to know. Moreover, the bath temperature T_0 might be not well defined if the electrical noise is not white which I expect it not to be. Maybe the underlying assumptions should be mentioned as well.

We thank the reviewer for the suggestion. We have added this information the Supplementary Materials S6, and reproduce it as follows:

“... Here we model the electrical noise as a white noise bath and assume the particles are trapped in a quadratic potential. Under the assumption that the equipartition theorem holds, the different bath temperatures used in the main text figures are listed in Supplementary Table 3. In Fig. 4(c), P2 is located closest to the trap center and therefore experiences the least electric field noise, resulting in the lowest bath temperature. In contrast, P1 and P3 are farther from the trap center, and their bath temperatures are approximately an order of magnitude higher than that of P2. A similar trend can be observed in Fig. 4(b), where P1, being farther from the trap center, also exhibits a higher temperature than P2.”

Table 2: Different equilibrium temperature T_0 values in the main text figures.

	Fig3(b)	Fig3(c)	Fig3(d)	Fig4(a)	Fig4(b)	Fig4(c)
T_0 (K)	900 ± 300	1500 ± 100	400 ± 50	500 ± 70	P1: 2300 ± 300 P2: 600 ± 100	P1: 3300 ± 600 P2: 400 ± 50 P3: 4300 ± 800

Comment VII:

I claim the authors cannot distinguish between $\phi_0 = 370$ degrees, 10 degrees or 730 degrees. Accordingly, the statement . . . noting that one period of phase delay does not significantly effect cooling . . . should be adapted in a way that it could be many oscillations delay, not just one.

We are confident in our assessment of the latency in our system, which we have been able to measure, as given in Supplementary Material S4. Because our frequencies are low, one full period of system latency is easy to distinguish from another.

Comment VIII:

. . . The value of Γ_{fb} obtained by fitting the data in Fig. 3(c) with equation (3) agrees with the value obtained at the same feedback gain by fitting the data in Fig. 3(a)

I claim that Fig. 3a has no value that is stated afterwards, only $\Gamma_{fb} = 0, 2.5, 6.0$ Hz. with equation (1), . . .

Thanks for mentioning this point. We removed many of our plots (with different values of Γ_{fb}) from Fig 3a to improve the readability, at your suggestion. But it will not affect the reading, the value of 0.7, for example, is clearly visible on Fig. 3(b). To avoid confusion, we have added more explanation in the caption to Fig. 3 as follows (underlines not in the manuscript, but to highlight the relevant changes to the reviewer):

“A selection of single particle PSDs of the measured motion along the z –axis with different feedback gains $\Gamma_{fb}/2\pi$. fit with equation (1). (b) Extracted T_{CoM} relative to initial temperature T_0 over a wider range of feedback gains than presented in (a)”

Comment IX:

. . . Finally, we combine the optimized Γ_{fb} and ϕ to push cooling to the limit by reducing. . . How is Γ_{fb} optimized? The displayed data does not show a temperature study versus the feedback strength Γ_{fb} where heating was observed. What is the optimized value of Γ_{fb} ? According to Fig 3a and b it should be >6 Hz..

We thank the reviewer for the question. In Fig. 3(d), our Γ_{fb} and ϕ are actually fixed with the optimized values at pressure 10^{-3} mbar, and we show how the temperature is affected by reducing the pressure. We have adjusted the following wording to correctly address this figure.

In the Fig. 3 caption:

“Variation in single particle temperature with fixed feedback gain and phase, as the background gas pressure decreases. When the pressure reaches 10^{-3} mbar we cool to the noise floor of our system, as indicated by the grey shaded region, corresponding to $T_{CoM} = (6.8 \pm 0.7)$ K”

And in the last paragraph of the section **Single particle cold damping using neuromorphic imaging**:

“Finally, in Fig.3d) we show the variation in temperature with Γ_0 by reducing the gas pressure, with Γ_{fb} and ϕ fixed at around 6 Hz and 0° , respectively. At a pressure of 10^{-3} mbar, we reach the noise floor of our system, indicated by the grey region, at a temperature corresponding to $T_{CoM} = (6.8 \pm 0.4)$ K, representing 17 dB of cooling. The optimal Γ_{fb} can be derived from equation (2), see Supplementary Materials S7.”

To answer the reviewer’s question: We did not present data on heating at high feedback gain, as we did not think it would be of interest to the reader. We now include this analysis in the Supplementary Material S7, which shows in theory that, when Γ_{fb} increases, T_{CoM} first decreases, then gradually stabilizes at a certain value, and finally increases again. This behaviour arises because the increasing amplitude of the feedback signal also amplifies the noise fed back to the particle, preventing from further cooling and eventually cause heating. From Eq. 2 in our text, we can obtain the partial differential equation of T_{CoM} with respect to Γ_{fb} . Via solving $\partial T_{CoM} / \partial \Gamma_{fb} = 0$, the optimal Γ_{fb} is obtained [B. Melo, et al., *Nature Nanotechnology* 19.9, 1270–1276 (2024).]. In our case of Fig. 3(a) and Fig.3(b) where the pressure is 10^{-2} mbar, the optimized $\Gamma_{fb} = 18$ Hz which we didn’t reach. In the case of Fig. 3(d), we take $\Gamma_{fb} = 6$ Hz. At the pressure at 10^{-3} mbar we hit our detection noise floor.

Comment X:

Filtering can be improved either by separating the particle modes and applying bandpass filtering, or through the use of phase locked loop . . .

The fact that the authors are driving/cooling all particles with the same signal is also not ideal. See earlier comment.

We acknowledge that driving all particles with the same signal can limit the performance of cooling due to cross-talk among modes of the system. We have added this information to the manuscript, as stated in comment 24.

Reviewer #3 (Remarks to the Author):

I read the response from the authors with scrutiny and I acknowledge the efforts that have been made. If I agree with many of the replies to my comments, I would like the authors to make a few clarifications before accepting the publication of their work.

We extend our sincere gratitude to the reviewer for careful and constructive suggestions, which have been instrumental in improving the quality of our manuscript. We have reviewed each point raised and have made corresponding changes.

Comment 2:

I am convinced (and I was already) that the particles can indeed couple. The new data are convincing in that regard and improve the quality of the paper without a doubt. Yet, as the authors stress it at some point in the text, the multiparticles modes that are cooled need to be uncoupled. Thus, I would appreciate if this important point is mentioned early in the text. For instance, can the author modify the sentence in the abstract: “[...] control by implementing real-time feedback to cool the motion of three objects simultaneously [...]” into “[...] control by implementing real-time feedback to cool the motion of three uncoupled objects simultaneously [...]”

We thank the reviewer for the valuable advice. We agree that the term "uncoupled" is crucial for clarity. We have revised the abstract sentence to reflect this important point, which were originally written:

“We present a scalable method for arbitrary multiparticle tracking and control by implementing real-time feedback to cool the motion of three objects simultaneously,”

“to demonstrate real-time simultaneous feedback control of multiple particles in an array.”

Now they read as follows:

“We present a scalable method for arbitrary multiparticle tracking and control by implementing real-time feedback to **simultaneously cool the motion of three uncoupled objects,**”

“to demonstrate real-time simultaneous feedback control of multiple **uncoupled** particles in an array.”

Comment 5:

I believe there is a misunderstanding between myself and the authors. Their technique (at least at this stage) requires the modes to be uncoupled (see

above). Thus, they should not overlap spectrally. I believe that we all agree that, if we note Δf the linewidth of the resonances, assuming that the authors can assemble resonances over the total spectral range B , the max number of modes that can be cooled scales as $B / \Delta f$. Thus, I do understand that taking narrower resonances increases the number of modes that can be cooled.

My point relates to the fact that Δf cannot be arbitrarily small. Let's assume that they have two modes that are ultra-narrow and very close in frequency. As suggested in the reply, I can admit (assuming no experimental drift) that they start with a routine that measure extremely precisely the resonance frequencies of those modes. The cooling of these two modes will be achieved using an electronic signal that encapsulate the carrier frequencies of both modes. Yet, this signal has a certain time duration, meaning that it will possess a non-negligible spectral width that limits the spectral resolution of the technique. Therefore, it seems to me that there should be a limit in the number of modes that can be cooled (like in any technique). In that regard, is there a way to estimate (by a rule of thumbs) of many modes can be addressed? This would for me a straightforward way to back up the claim below: "We believe that the particle control method presented in this work could be extended to an array of order 100 microparticles [...]" , which at this stage remains speculative to me.

Indeed, considering the measurement time duration for signal demodulation there is a bandwidth limit for filters to pick out certain frequency signal which will set a limit to Δf . But PLLs have very narrow filter bandwidth, for example a Zurich Instruments HF2LI has minimum 83 μHz bandwidth and can address 6 peaks at once with this filter. Therefore, with a PLL it is possible to identify feedback signals at mHz resolution.

On the other hand, since we use cold damping to cool the particles, the particles inevitably experience an additional damping, which broadens the linewidth and makes the modes harder to distinguish. According to Eq. (2) in the manuscript, by solving $\partial T_{CoM} / \partial \Gamma_{fb} = 0$, we can obtain an optimal Γ_{fb} at a given pressure (See Supplementary Materials S7). This optimal Γ_{fb} would be the dominant factor in determining the scaling of how many particles can be cooled. The optimal Γ_{fb} reduces as pressure goes down.

If we take the above argument as a rule of thumb, we can estimate the number of modes as follows. Suppose we start cooling at 10^{-4} mbar in experiment, at which the particle's linewidth is about $\Gamma_0 \approx 0.002$ Hz, the optimal feedback rate is $\Gamma_{fb} \approx 1.8$ Hz. This limits the capacity of our setup to be around 270 modes. We are far more cautious in our claim of number of particles which can be simultaneously cooled then detected.

We made the following changes to the main text guiding the reader to a discussion in Supplementary Materials S8.

“We believe that the particle control method presented in this work could be extended to an array of order 100 microparticles (see Supplementary Materials S8).”

Reviewer #1 Comments (Attachment 1)

Comments to the authors concerning NCOMMS-24-58065-T

The authors *Y. Ren et al.* of the manuscript **Neuromorphic detection and cooling of microparticle arrays** describe a novel technique to detect multiple particles simultaneously by using a single detector. This single detector is an event based camera that allows for scalable detection of many particles.

The authors employ this new technique to apply cold damping on several modes of one particle and single modes of several particles. The cooling itself does not match the state of the art, both in pressure regime and noise performance. The main result of the manuscript is the fact that several particles can be cooled, which is the first demonstration of a scalable technique that could allow the motion control of 100s of particles in high vacuum conditions.

The levitation field is currently moving towards many body physics in e.g. optical lattices or coulomb crystals. While the levitation of particle arrays by itself is already demanding, their motion control in vacuum relies on multiparticle detection and up to today no scalable technique is available. This makes the work of *Y. Ren et al.* very timely.

While the authors spend a lot of effort to discuss their cooling results, the discussion of the limitations and possibilities of this event based camera, their main contribution to the field, is shallow. In my opinion, the authors should answer some key questions concentrating on the detection aspect:

1. What is the detection efficiency or signal-to-noise ratio in the current experiments? What is the main limitation?
2. What kind of improvements are needed to reach ground state cooling with event based cameras?
3. With the employed event-based camera, how many particles can be controlled? Is there a trade-off between detection quality and particle number?
4. Ultimately, 3D detection will be needed. What are the possibilities and their consequences to implement 3D detection with the current setup?

Nevertheless, in my view this approach is promising and innovative and with some revision I believe the manuscript could merit publication in Nature Communication. Below the authors will find general and specific comments/suggestions, that I hope the authors will find helpful.

Quality of the data: The presented data, theory and conclusions on cooling is, except of a few points highlighted later, valid and robust. Nevertheless, data supporting the performance and limits of the new detection technique is not very abundant and would be valuable to add. This could also raise the significance of this work.

Appropriate referencing: The referencing can partially improved. A few times more original and more suitable references are not cited in favour of more current ones, replicating the original ones. Furthermore, references are sometimes not complete e.g. ground state cooling or not covering the referred topic e.g. quantum enhanced sensing. More precise comments can be found in the specific comments.

General comments:

- Some sentences are disconnected and causalities between statements are sometimes not super clear, e.g.

Detecting and controlling multiple particles in vacuum has so far involved single particle control with sympathetic cooling [23, 24, 26], and arrays of optical traps [27]. It will be a challenge to levitate and individually control arrays of tens, or even thousands of objects [9].

Why is it a challenge to control more particles with the referenced methods? This comment applies more strongly to the first part of the manuscript.

- The authors trap charged particles in a Paul trap that are coupled via Coulomb interaction. How can the effects of sympathetic mode cooling be excluded?
- The authors mention that they apply additional filters to their feedback signal. It is not clear to me what the goal or figure of merit of this filtering is. Could the authors add this information to the manuscript?
- The authors refer to arrays of tens of particles. It is not clear to me if they mean an array of e.g. 3x3 or 10x10. Could the authors be more specific in the manuscript by giving the dimension of the hinted arrays?
- The detection method is inherently 2D and allows for 3D detection only at the price of misalignment. Can the authors discuss the limitations if their method is used for the ultimately required 3D detection.
- Can the authors discuss what the minimal distance on the detector sensor is before cross leakage between different particles occur and what this means for the maximum particle number? The signal of one particle is spread over how many pixels in this work?
- The detection bandwidth is high enough for Paul traps with lower mechanical eigenfrequencies. What about optical traps? Can the authors state a value for the bandwidth in the manuscript?
- I believe there is not a single Paul trap reference. Could this be added?
- Suggestion: All equations are in terms of ω and plots are in terms of f . In my opinion it helps preventing misunderstandings if this is adjusted to one of the two cases. It is also not quite clear if quantitative values for Γ_{fb} are now $\Gamma_{fb} = 4\text{Hz}$ or $\Gamma_{fb} = 2\pi \times 4\text{Hz}$.
- A few typos can be found: Camers, Proprietary

Specific comments:

- In the abstract the authors claim that an array of particles increases the sensitivity. Why is that?
- I disagree with the following statement

... The performance of a mechanical sensor is inversely proportional to its volume due to dissipation through thermal contacts and surface strain ...

I claim that the acceleration or force sensitivity of a thermally limited harmonic oscillator is given by $S_{aa} \propto \sqrt{\frac{k_B T \Gamma}{m}}$ or $S_{FF} \propto \sqrt{k_B T \Gamma m}$ which is related to the mass and not the volume. Maybe the authors could clarify their statement to avoid misunderstanding.

- I claim the following statement is only true at ultrahigh vacuum

... By levitating nano- or micro-particles in optical, electrical or magnetic fields, one creates a mechanical oscillator with remarkably low dissipation ...

Could the authors clarify this?

- There exist older proposals that suggested macroscopic quantum physics with levitated particles than [16]

... opens the door to macroscopic quantum physics [16]. ...

as for example [Chang, Darrick E., et al. Proceedings of the National Academy of Sciences 107.3 (2010): 1005-1010] and [Oriol Romero-Isart et al 2010 New J. Phys. 12 033015].

- Are dynamics vision sensors and neuromorphic sensors the same?
- A word is missing

... Therefore, neuromorphic detection is highly suited to high-speed and real-time applications requiring low-power in environments with uncontrolled such ...

- In my opinion is the chosen reference for active feedback via cold damping unsuitable.

... We implement cold damping feedback to cool the motion of the levitated particles [40], ...

Instead the following should be cited [Tebbenjohanns, Felix, et al. "Cold damping of an optically levitated nanoparticle to microkelvin temperatures." Physical review letters 122.22 (2019): 223601], [Conangla, Gerard P., et al. "Optimal feedback cooling of a charged levitated nanoparticle with adaptive control." Physical review letters 122.22 (2019): 223602] and [M. Poggio, C. L. Degen, H. J. Mamin, and D. Rugar Phys. Rev. Lett. 99, 017201].

- I suggest to add the word of L. Magrini et al too.

... technique with demonstrated ground state cooling capabilities [41]. ...

- Could the authors add more quantitative information related to the following statement?

... This single-device method for cooling and controlling particle arrays is readily scalable due to the low data output of neuromorphic detection. ...

More precisely, how many particles can be tracked? And what is the limitation e.g. the sensor size or data flow?

- Could the authors elaborate on the following statement?

... Arrays of cooled microsensors will lead to enhanced sensitivity [42], ...

In my opinion, several coupled oscillators allow for the implementation of the mode localization method, but this method is not increasing the sensitivity of a thermally driven harmonic oscillator as such. Could the authors clarify their statement?

- In Fig 1a), what is the black electrode? The particles are depicted in a circle. Shouldn't the particles form a 1D chain?
- I find the relative panel size of Fig 1 unlucky, given that panel b) is the most important panel. Some boxes in Fig1b) show "two" particles. Why is that and what is the consequence?
- It is not clear of which variable the PSD is calculated:

... In Fig. 1(c) we generate the power spectral density (PSD) ...

- The relative position of the particles is unclear.

... for each particle in an array of four ...

Is this a 1x4 or 2x2 array?

- Could the authors elaborate more on this?

... Interactions between the particles are evident, as shown by the unlabelled peaks in Fig. 1(c). ...

I believe some peaks could also be higher harmonics of a certain mechanical eigenmode. I claim that interactions should lead to coupled oscillators with peaks at the sum and difference of the peaks, which I can not identify. If the authors have identified the origin of additional peaks, it would be helpful to add additional labels.

- Is the Paul trap symmetric in x and y ? If so, could the authors elaborate why all x -modes are equal in frequency but the y -modes are different?
- I disagree with this statement:

... By reducing the position fluctuations interactions between the particles can be minimized. ...

I claim that the interactions can be stabilized due to fixed positions but this does not imply that interaction strengths are decreased.

- The authors should clarify that they mean active feedback methods.

... Reduction of the particle energy to the ground-state of the levitating potential [14, 46] ...

Otherwise, citations using cavity cooling are missing.

- References incomplete or misleading:

... opens up a toolbox of quantum control [47] and sensitivity enhancement [6]. ...

Depending on what the authors mean with quantum control, the references should be more towards the manipulation of mechanical oscillators in the quantum regime; or if the authors only refer to cooling, then the reference list should include all 1D cooling experiments too.

Concerning [6], I claim that the reference does not cover quantum enhanced sensitivity but instead is a classical force sensing experiment. An example of quantum enhanced sensing would be the exploitation of squeezed mechanical states.

- Original references on cold damping are missing:

... Cold damping is a feedback method ...

E.g. [Tebbenjohanns, Felix, et al. "Cold damping of an optically levitated nanoparticle to microkelvin temperatures." Physical review letters 122.22 (2019): 223601], [Conangla, Gerard P., et al. "Optimal feedback cooling of a charged levitated nanoparticle with adaptive control." Physical review letters 122.22 (2019): 223602] and [M. Poggio, C. L. Degen, H. J. Mamin, and D. Rugar Phys. Rev. Lett. 99, 017201].

- I believe in Eq. (1) it should be at least hinted that there is a noise term being neglected, that heats the particle motion for large gains (a regime that is explored here too). For more details see Eq. (4) in Conangla et al., PRL 122, 223602 (2019). Especially, because the noise term appears then suddenly in Eq. (2).
- In my opinion there are earlier references for Equation (2), namely Conangla et al., PRL 122, 223602 (2019).
- ... In Fig. 2(b) we show the effect of increasing feedback gain on the temperature of a single mode of a single particle, and compare the data to the model in equation (2), with only S_{nn} as a free parameter. ...

I disagree with this statement that S_{nn} is the only free parameter since the same data has previously been used to extract T_{com} and Γ_t . It would be also interesting if the fitted value for S_{nn} coincides with the noise extracted from the detection with the EBC and how this compares to standard photodiodes.

- The authors are fitting Γ_{fb} twice.

... This model is fit to experimental data, Fig. 2(c), with Γ_{fb} and ϕ_0 as free parameters. ...

Do the values coincide?

- In Fig 3, why are two particles colder than 3 particles if the particle detection is independent?

- It is not quite clear to me what imperfect filtering shall mean. Is this due to the fact that the detection with the camera does not deliver independent detection signals, or because the filters are too steep such that unwanted phase shifts are occurring or is it because only one single electrode is used, which leads to cross talk in the electric fields?

- Can the authors elaborate more on this statement?

... Different particles can have different noise floors, since their coupling to voltage noise depends on the particle charge. ...

In my opinion, the noise floor is particle independent and is the noise in absence of the particle motion. While I agree that the amplitude of the motion might depend on the particle charge and therefore the SNR, I disagree that the noise floor should depend on the particle charge. Can the authors explain this in more detail?

- ... The issue of cross-talk is more pronounced when there are more modes of the system, yet we still achieve better than -7 dB of cooling ...

Can the authors specify at which point this cross-talk happens, e.g. in the detection, in the summation of the individual feedback signals, or elsewhere.

- Phase lock loops have been in use for a long time. Maybe reference [50] is not the most suitable one.

- If I understand correctly, the independent detection of many particles with a single device is the main result of the paper, which is independent of the levitation technique. I find the following sentence therefore misleading since the charge is not needed for detection:

... It's also worth mentioning that our method is suitable for any object which can carry a charge, unlike optical trapping with intense beam detection which is limited to low-absorption dielectrics. ...

- In Fig. 3d), which particle is which in Fig 3c)? Does the final temperature match the expectation from the SNRs in Fig. 3d)?

Reviewer #1 Comments (Attachment 2)

Comments to the authors concerning NCOMMS-24-58065-T

The authors *Y. Ren et al.* of the manuscript **Neuromorphic detection and cooling of microparticle arrays** addressed all of my questions and answered most of them sufficiently. I agree with the authors that limitations and possibilities are hard to quantify, but I also believe that this is required to make this work a high-impact contribution to the field and to underline the discussed prospects of high frequency detection and cooling to low phonon states, despite the setup being unable to do so.

I stand with my view that this approach is promising and innovative but unfortunately I also believe that more revision is required before the manuscript could merit publication in Nature Communication. As you will see my comments concern mainly the cooling via cold damping and not the detection method itself. Below the authors will find general and specific comments/suggestions, that I hope the authors will find helpful.

General comments:

- As a matter of personal liking, I still believe that the readability of the manuscript could be improved and a more concise writing style could be applied.

Specific comments:

- The following statement references only ground-state cooling with a cavity.

...and the demonstration of cooling to the ground state of an optical potential [15–17] opens...

As generally as the sentence is phrased, I find the list of references incomplete and, given that the authors are applying measurement-based feedback, also misleading. I would suggest the authors either cite also experiments for ground state cooling with measurement based feedback, or concentrate solely on measurement based feedback and phrase the sentence accordingly.

- ... a truly scalable method ...

Unfortunately, I do not understand what "truly scalable" means in comparison to scalable. Could the authors please clarify this to me?

- Concerning Fig 1a: The smaller inset of the Paul trap is identical to the larger illustration in the same figure. Instead of showing the same structure twice, I would appreciate if the 3D aspect of the electrode layout were displayed more clearly. It is not clear to me that black and blue electrodes are end-cap electrodes.

- Concerning Fig.2a: Why are the particles placed on a diagonal line and not the central line (along z)? In my understanding, the particle furthest away from the Paul trap center should experience the strongest noise and strongest micromotion. In Fig 2a, the particles' amplitudes suggest the opposite? Could the authors elaborate on this? Which particle is thermalized to 900K? Fig 2a suggests that the particles thermalize to different temperatures since they display different amplitudes or can this be attributed to different charges? Have the authors measured charge to mass ratios of the particles?
- Concerning Fig.2b: Why do you see only a center of mass mode (CoM) along x ? Why do you observe coupled modes but also an uncoupled mode in y ? I would expect only one of the two scenarios (coupled or uncoupled)? The particles are closer together in z than in y (see Fig 2a). Why is there no coupling along z ? Could the authors please add numbers to the y -axis? Could they authors state the current SNR?
- I find the following statement misleading.

... this method can cause damping or amplification of the particle motion without adding additional noise, ...

I claim that active feedback always introduces noise, unless one assumes the theoretical case of an ideal detector with zero detector noise, which does not mirror the current situation.

- Concerning Fig 3a: Maybe the authors want to remove some data traces to enhance readability.
- Concerning Fig 3b)-c): If I understood the main text correctly, the two data sets applying the same experimental procedure are fitted with different theoretical equations (Eq. 2 to Fig 3b and Eq. 3 to Fig.3c), where one is considering noise and the other one is not. In my opinion both data sets should be fitted to the same equation, taking the noise into account, meaning Eq. 2 with a phase dependent $\Gamma_{fb}(\theta)$.
- Concerning Fig 3d: Which equation is used to fit the data?
- Concerning Fig 4d: Maybe the authors can consider removing data points to enhance readability?
- How are the errorbars derived?
- Concerning Eq.1: I claim T_{com} should be replaced with T_0 the bath temperature. I agree that in the absence of cooling $T_{com} = T_0$ but this is not the case for $\Gamma_{fb} \neq 0$. The following sentence also needs adaptation.
- I suggest to modify Eq. 3 and replace it with a clarification in the text that the cold damping feedback is phase dependent $\Gamma_{fb}(\theta) = \Gamma_{fb} \cos(\phi + \phi_0)$. If the authors want to keep Eq. 3, then I suggest that they introduce the noise term in Eq 3, such that Eq 2 and 3 are compatible and describe the same situation.
- I would like to raise again that Eq.1 is neglecting the noise contribution. The T_{com} (Eq 2) is directly related to the area of PSD of the displacement noise S_{xx} in Eq 1, therefore I claim that it is not consistent to consider the noise contribution in Eq. 2 but not in Eq.1. Furthermore, I find that the added sentence is not resolving the issue

... When Γ_{fb} is large the feedback can introduce extra noise which modifies equation (1) as discussed in [50, 56]...

- I suggest to add reference 40

...extra noise which modifies equation (1) as discussed in [50, 56]. ...

- Could the authors highlight the units of S_{nn} ? From the text, it is not clear to me if this is voltage noise, displacement noise or other.
- If Γ_{fb} is fitted twice and agree within errors, this could be stated.
- I find the following statement misleading

...The fitted value of ϕ_0 is 5° , noting that one- or two- periods of phase delay do not significantly effect the cooling for an underdamped oscillator as in our system ...

First of all, $\phi_0 = 5^\circ$ is much less than an oscillation period. Second, only the value of $\phi_0 + \phi$ is important (assuming an underdamped oscillator). And the statement is only true if the mismatch coincides with full oscillation periods. If ϕ_0 equals a fraction of a period while ϕ is constant then cooling can turn into trapping, heating, or anti trapping.

- ... We are sensitive to all three degrees-of-freedom due to the angle our imaging system makes to the principal axes of the trap. The geometry of our Paul trap allows the control of all degrees-of-freedom with a single electrode. ...

Why do you only cool two modes then? Given your outlook to employ the method to larger arrays and lower vacuum levels, would cooling in 3D not strengthen your claim?

- I must admit that I am not fully convinced by these statements

... unfiltered noise from one particle is able to heat the uncooled modes of the other. At lower pressures this causes particle instability and prevents further cooling...

...The issue of imperfect filtering is more pronounced when dealing with more particles, as there are more modes of the system overlapped with the unfiltered noise. ...

The authors are using only one electrode pair to apply the electrical force onto the particles. Even without additional noise, the electrical signal contains the signal of all particles/modes meaning that one drives all particles at the same off-resonantly and out of phase time, leading to heating. Why are the authors excluding this effect?

- Can the authors specify the type of interaction? (I guess Coulomb interaction)

... we cannot see the type of interparticle interactions ...

- ... Collective modes in particle arrays can also be cooled via sympathetic cooling [28, 29]...

I was under the impression that only center of mass modes (in phase modes) can be cooled and the breathing modes (out-of-phase modes) cannot (assuming the same electric field configuration)?

- ... Paul traps are stable at low pressures [73], where the motional frequencies of levitated particles have sub-Hz linewidths [74], meaning that the naturally varying charge-to mass ratio of charged nano- and micro-particles will enable single-particle control and cooling even for large arrays. ...

In my opinion, this statement is very strong. Given that the authors use the stated platform but do not demonstrate control of only a few particles at very low pressures because of difficulties. Also the problem of uncooled coupled modes seem to be neglected.

Reviewer #1 Comments (Attachment 3)

Comments to the authors concerning NCOMMS-24-58065-T

The authors *Y. Ren et al.* of the manuscript **Neuromorphic detection and cooling of microparticle arrays** addressed all of my questions and answered most of them sufficiently. My comments are mainly suggestions, that I hope the authors will find helpful. Exceptions to this are comments 12, 14, III and IV which I believe needs addressing.

- **Comment 3:** I appreciate the authors effort. Unfortunately the end cap electrodes are covered by larger electrodes. I would like to make the not required suggestion to adapt the figure again.
- **Comment 5:** To prevent misunderstandings of the reader, I would suggest to define the origin of the camera and particle coordinate system to be the trap center (see Fig.2a).
- **Comment 8:** A unrequired suggestion is to state the charge to mass ratio related to Fig.2.
- **Comment 9:** Thanks for illustrating this point. I found the illustration added to the answer of this comment to be more useful then Fig. 1 and Fig. 2. Consider adding it to the manuscript/replacing part of Fig.1 with it. Nevertheless, I still find the explanation of the absence of coupled modes in the y- and z-direction in the manuscript insufficient. Could the authors add a sentence highlighting the fact that the coupling is increased along x due to the motion along the array (in contrast to y and z)?
- **Comment 10:** In the given reference J. Gieseler and J. Millen, Entropy 20.5, 326 (2018) they state

... Note that this **simplified picture assumes** that the feedback signal is perfect and that it does not feedback any noise, which in general is not true...

In my opinion the author's phrasing

... this method **can cause** damping or amplification of the particle motion without adding additional noise, hence the terminology "cold damping". In reality, input and output noise of the feedback electronics still limits cooling...

is causing unnecessary confusion without adding important information to the topic of the manuscript. The wording **can cause**, in contrast to **assumes**, suggests a real possibility which is incorrect.

- **Comment 12:** If I understand correctly, the detector/camera is an in-loop detector meaning that it is used to detect and at the same time to generate the feedback signal. If this is the case, I disagree with Eq. 1 where I claim S_{dd} should not be present as an independent noise term. All noise (in the feedback loop S_{nn} and in the detection S_{dd}) is noise that is being feedback and therefore heats the motion governed by the susceptibility of the oscillator (see M.Poggio PRL 99, 017201 (2007)).

- **Comment 14:** Instead of increasing the readability of Fig. 4d,, I can barely see the data now which is the basis of the displayed fit. Reconsider decreasing the number of data points instead of making them less visible as suggested previously.
- **Comment 15:** This information should be added somewhere

... We take 15 repeats at each set of experimental parameters, and this allows us to calculate a mean area and the standard deviation, which we use as an experimental error bar. ...
- **Comment 17:** Eq. 2 and 3 are the same now. Consider merging them.
- **Comment 24:** I agree with the authors that they can filter the detection signals. But my concern was related to cooling, so feeding back the signal. Let's assume perfect, noise free detection of oscillators at frequencies f_1, f_2 , and f_3 , To control the modes with a single electrode means the authors add the three detection signals and apply it to a single electrode. This means that I will drive all oscillators with all three frequencies of which only one frequency is required for cooling. Hence even in the absence of noise, the authors will heat up to a certain extent all the modes. So additionally to unfiltered noise (which is discussed by the authors), this will be an additional problem (which is not mentioned).
- **Comment 26:** I disagree with the author. In Fig. 3 of D. S. Bykov et al., Optica 10(4), 16 438–442 (2023) they show that they can only cool efficiently the detected particle. Collective modes can only be cooled in the weak cooling regime where the feedback damping is smaller than the coupling strength of the modes.

New comments based on manuscript

- **Comment I:** Strictly speaking I disagree with the following statement

... The particles are not coupled in the z-direction, which is the mode we use for multiparticle cooling. ...

If the particles are charged, then they are always coupled, only their coupling in this case is negligible. I suggest to change the wording.
- **Comment II:** The main novelty of this article is the simultaneous detection of many particles. Hence, I would suggest to state typical SNR achieved here in the context of the following sentence:

... The signal-to-noise for the different particles varies due to non-uniform illumination and varying coupling to electronic noise...

since this is the main general limitation for particle cooling.
- **Comment III:** I claim Fig.3 c plots the relative temperature versus $\phi + \phi_0$ instead of ϕ . If not then, why does the relative temperature maximize at $\phi = \pi$?
- **Comment IV:**

... Figure 3(a) shows the PSD of a particle's motion along the z-axis as it is cooled via cold damping. The shape of the position PSD S_{zz} is given by [57]: ...

I claim that Fig 3a is showing the detected position signal which is not identical with the real position signal, which due to detection noise is undetectable directly. I suggest to clarify that Fig 3a shows the measured detection signal. Eq. 1 on the other hand is the real position (neglecting S_{dd}). The measured data should be fitted with the power spectral density of the detected signal and the real position inferred from there. I am aware that differences will be small (possibly negligible) but wording should be corrected.

- **Comment V:** I suggest to change the statement

... a constant spectral density...

to "a constant power spectral density". More common would be to use the term white noise as described in Millen, J., Gieseler, J. (2019). Single particle thermodynamics with levitated nanoparticles. In Thermodynamics in the Quantum Regime: Fundamental Aspects and New Directions (pp. 853-885).

- **Comment VI:**

... The parameters T_0 , Γ_0 and $\Gamma_{fb}(\phi)$ can be extracted from a measured PSD by fitting equation (1) ...

Given that T_0 can vary significantly for Fig 3 and Fig 4, it would be helpful to state it somewhere. E.g. in Fig. 4c, the particles would be cooled only close to room temperature if $T_0 = 1500\text{K}$, which I believe is important to know. Moreover, the bath temperature T_0 might be not well defined if the electrical noise is not white which I expect it not to be. Maybe the underlying assumptions should be mentioned as well.

- **Comment VII:** I claim the authors cannot distinguish between $\phi_0 = 370$ degrees, 10 degrees or 730 degrees. Accordingly, the statement

... noting that one period of phase delay does not significantly effect cooling ...

should be adapted in a way that it could be many oscillations delay, not just one.

- **Comment VIII:**

... The value of Γ_{fb} obtained by fitting the data in Fig. 3(c) with equation (3) agrees with the value obtained at the same feedback gain by fitting the data in Fig. 3(a) with equation (1), ...

I claim that Fig. 3a has no value that is stated afterwards, only $\Gamma_{fb} = 0, 2.5, 6.0$ Hz.

- **Comment IX:**

... Finally, we combine the optimized Γ_{fb} and ϕ to push cooling to the limit by reducing...

How is Γ_{fb} optimized? The displayed data does not show a temperature study versus the feedback strength Γ_{fb} where heating was observed. What is the optimized value of Γ_{fb} ? According to Fig 3a and b it should be $>6\text{Hz}$.

Comment X:

Filtering can be improved either by separating the particle modes and applying band-pass filtering, or through the use of phase locked loop . . .

The fact that the authors are driving/cooling all particles with the same signal is also not ideal. See earlier comment.